# Effective Neural Approximations for Geometric Optimization Problems

**Samantha Chen**
Computer Science and Engineering
University of California, San Diego
sac003@ucsd.edu

**Oren Ciolli**
Computer Science and Engineering
University of California, San Diego
ociolli@ucsd.edu

**Anastasios Sidiropolous**
Computer Science and Engineering
University of Illinois, Chicago
sidiropo@uic.edu

**Yusu Wang**
Halıcıoğlu Data Science Institute
University of California, San Diego
yusuwang@ucsd.edu

## Abstract

Neural networks offer a promising data-driven approach to tackle computationally challenging optimization problems. In this work, we introduce neural approximation frameworks for a family of geometric "extent measure" problems, including shape-fitting descriptors (e.g. minimum enclosing ball or annulus). Central to our approach is the *alignment* of our neural model with a new variant of the classical $\varepsilon$-kernel technique from computational geometry. In particular, we develop a new relaxed-$\varepsilon$-kernel theory that maintains the approximation guarantees of the classical $\varepsilon$-kernels but with the crucial benefit that it can be implemented with *bounded model complexity* (i.e, constant number of parameters) by the simple SumFormer neural network. This leads to a simple neural model that approximate quantities such as the directional width of any input point set and empirically shows good out-of-distribution generalization. Many geometric extent measures, such as the minimum enclosing spherical shell, cannot be directly captured by $\varepsilon$-kernels. To this end, we show that an encode-process-decode framework with our kernel-approximating NN used as the "process" module can approximate such extent measures, again, with bounded model complexity where parameters scale only with the approximation error $\varepsilon$ and not the size of the input set. Empirical results on diverse point-cloud datasets demonstrate the practical performance of our models.

## 1 Introduction

Using neural networks to solve optimization problems in a data-driven manner has received great attention in recent years [9, 45, 19, 30]. However, one challenge in using neural networks to solve optimization problems is that their architectures often grow with problem size [23]. The key question we consider is the design of practical neural models that both have the capacity to approximate solutions to specific optimization problems with *bounded model complexity* (i.e. number of parameters is independent of the size of input), and perform well in practice. A direction that has shown promise both theoretically and practically is *neural algorithmic alignment* [37, 38, 44, 36], which intuitively informs architecture design via loosely imitating ("alignment" with) certain algorithmic frameworks to achieve improved accuracy and out-of-distribution (OOD) generalization.

Our overall motivation is to design efficient and practical neural models to solve problems in a data-driven manner. In this paper, we study this problem in the context of a particular family of

39th Conference on Neural Information Processing Systems (NeurIPS 2025).

*geometric optimization* problems and demonstrate that biasing the model architecture towards some algorithmic flow yields more efficient and accurate approximations. In particular, we aim to compute a variety of point cloud descriptors, such as convex hulls, bounding volumes (e.g. minimum enclosing ball or ellipsoids), or best shapes fitting an input point set (e.g, the minimum-width slabs or spherical shells that cover an input set of points). These problems are called *extent measure problems* and are well-studied in computational geometry; e.g, [10, 2, 1, 46, 21, 16, 6, 11]. We show how biasing our model architectures towards a particularly well-known approximation technique known as the $\varepsilon$-kernel [3, 27, 2, 11] results in neural networks which can approximate the aforementioned point cloud descriptor with *bounded model complexity*, or a constant number of parameters which is independent to input size.

**New work.** In the high level, the type of geometric problems we consider have the following form: Given an arbitrary set of points $P$ in $\mathbb{R}^d$, the goal is to compute a certain geometric descriptor $\mu(P)$ for $P$. We want to train a suitable neural network model that can approximate $\mu(P)$ efficiently for *any* input point set $P$. We draw inspiration from an elegant line of work in computational geometry that approximately solves these problems using the $\varepsilon$-coreset framework via $\varepsilon$-kernels; see Section 2 for detailed descriptions. Our contributions can be summarized as follows.

**Neural approximation of $\varepsilon$-kernels.** An $\varepsilon$-kernel $Q$ of a set of points $P$ is a subset of $P$ that approximately preserves the projection of $P$ along any direction in $\mathbb{R}^d$. Once $Q$ is given, one can approximate a family of geometric descriptors (e.g, the convex hull of $P$) by computing them using $Q$ in time dependent only on $|Q|$ and not on $|P|$. However, computing the $\varepsilon$-kernel is not naturally aligned with common neural models for point clouds. We instead introduce an alternative theory of *relaxed-$\varepsilon$-kernels*, which retain the same approximation guarantees while aligning naturally with the simple Sumformer [5] (a variant of DeepSets [47]). This leads to an efficient neural model that approximates relaxed $\varepsilon$-kernels with *bounded model complexity* independent of size of input point set $P$.

**Neural approximations of extent measures.** Among the so-called *unfaithful extent measures* which cannot be directly approximated by $\varepsilon$-kernels, a family of them can nevertheless be approximated via the *linearization* technique [4]. In Section 4, we show that our relaxed-$\varepsilon$-kernel can still be combined with the linearization technique to approximate such extent measures. We further employ an *encode-process-decode* framework (introduced by [37] to solve algorithmic tasks) where our relaxed $\varepsilon$-kernel neural network will be used as the processor module. See Figure 1. This framework can approximate both faithful and a family of unfaithful extent measures using only bounded model complexity.

**Empirical performance.** Importantly, we empirically demonstrate the accuracy (both in and out of distribution) and efficiency of the proposed neural models both for approximating $\varepsilon$-kernels and several (faithful and unfaithful) extent measures (e.g, minimum enclosing balls / annulus) in comparison to both natural neural baselines and algorithmic approaches. Our Sumformer instantiated neural model consistently exhibits much stronger OOD generalization than Transformer-instantiated neural models for these tasks, which we think is partially due to the alignment of the Sumformer model with our relaxed-$\varepsilon$-kernel algorithm.

In summary, the encode-process-decode framework instantiated with our relaxed $\varepsilon$-kernel NN as the processor network is both practical and theoretically sound, and can solve a family of geometric optimization problems accurately. Importantly, our models offer the advantage of being differentiable, enabling them to be used in downstream ML pipelines. Our neural approach also provides greater flexibility in comparison to classical algorithmic techniques: our framework can tackle tasks where the objective is hidden and only implicitly given via labeled examples. In contrast to classical algorithms which can not be used without knowing the target objective, our model can still efficiently learn the objective in a data-driven manner, as long as the underlying goal corresponds to a variant of the shape-fitting problem (even when unknown).

**Related work.** This paper discusses efficient neural approximations of a certain family of geometric optimization problems on point clouds. As such, we will discuss relevant work related to neural point cloud processing and neural network approximations of general optimization problems. In particular, deep learning on geometric data – especially point clouds – has advanced rapidly. Two foundational permutation-equivariant models are DeepSets [47] and PointNet [28, 29]. Both DeepSets and PointNet have universal approximation guarantees; however, general universal approximation of permutation

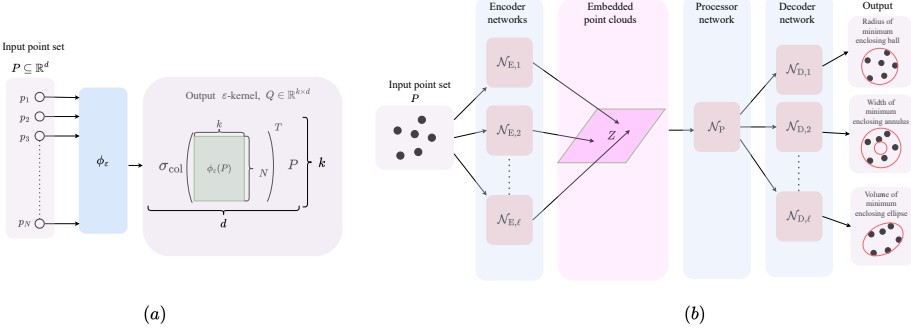

Figure 1: Illustration of (a) our kernel-approx-NN, and (b) our encode-decode-process framework for approximating various extent measures. The processor module $\mathcal{N}_P$ can be frozen as a pre-trained kernel-approx-NN, and only encoder / decoder needs to be trained in a task-dependent manner.

invariant functions requires the internal latent dimension *depend on the size of the input point set* [40, 12]. In geometric deep learning, it is also common to use geometry-enhanced graph neural networks to process point clouds; see the proto-book [8] for a comprehensive discussion. Recent work has seen a shift to Transformer based architectures for point cloud processing [48, 41, 42, 24, 14].

In recent years there have been great interest in using ML to help tackle combinatorial optimization problems; e.g, [39, 19, 22, 32, 13, 31], and surveys [7, 9]. Theoretically, there are also several works on the best possible approximation guarantees one can get from these neural network approaches [23, 31, 45, 30, 17]. However, bounded model complexity usually is not guaranteed, other than [18, 25], which use hybrid neural-algorithmic models. Our work uses neural algorithmic alignment ideas [37, 38, 44] to develop practical neural networks with bounded model complexity.

## 2 Preliminaries

**Transformers.** A standard Transformer is the composition of multiple Transformer blocks, each of which models a sequence-to-sequence permutation equivariant function using a specific attention mechanism. For simplicity, below we introduce the formulation using just a single attention head: Given a sequence $X \in \mathbb{R}^{n \times d}$ (a length-$n$ sequence of vectors in $\mathbb{R}^d$), the output of a single attention block $\mathcal{T}$ is another length-$n$ sequence $\mathcal{T}(X) \in \mathbb{R}^{n \times d}$ defined by

$$\mathcal{T}(X) = \text{MLP}(X + \text{Attn}(X)XW_V), \quad \text{where} \quad \text{Attn}(X) = \text{softmax}(XW_Q(XW_K)^T).$$

Here, $W_Q, W_K, W_V \in \mathbb{R}^{d \times d}$ are matrices and $\text{MLP} : \mathbb{R}^d \to \mathbb{R}^d$ operates element-wise on each of the $n$ elements. (We fix the internal dimension to be $d$ for simplicity of presentation.) Sequences for language tasks typically use positional encodings, e.g rotary PEs [33], to encode ordering information. However, since we are concerned with processing point clouds, each point in an input point cloud $P$ has its coordinates in $\mathbb{R}^d$; hence we omit positional encodings. For a Transformer with *bounded complexity*, i.e, total number of parameters is a constant independent of $n$, it still takes $O(n^2)$ time per forward pass when processing an input point set of size $n$.

**Sumformers.** A Sumformer [5] is a sequence-to-sequence permutation equivariant model formed by composition of blocks defined as follows:

**Definition 2.1** (Sumformer Block). *A Sumformer block is a sequence-to-sequence function $\mathcal{S} : \mathcal{X}^n \to \mathcal{Y}^n$, such that for any length-$n$ sequence $X = [x_1, \dots, x_n] \in \mathcal{X}^n$, the output is another length-$n$ sequence in $\mathcal{Y}^n$ defined by $\mathcal{S}(X) = \mathcal{S}([x_1, \dots, x_n]) = [\psi(x_1, Z), \dots, \psi(x_n, Z)]$, where $Z := \sum_{i=1}^n \phi(x_i)$, $\phi : \mathcal{X} \to \mathbb{R}^{d'}$, and $\psi : \mathcal{X} \times \mathbb{R}^{d'} \to \mathcal{Y}$. Usually, both the input and output feature spaces are Euclidean: $\mathcal{X} = \mathbb{R}^{d_1}$ and $\mathcal{Y} = \mathbb{R}^{d_2}$.*

Intuitively, at each layer, a Sumformer block first computes a global representation of the input sequence via sum-aggregation (the quantity $Z$ above), and then computes the individual representation of each input token from its initial encoding and the global representation $Z$ via the function $\psi$ (i.e, $\mathcal{S}(X)[i] = \psi(x_i, Z)$). Conceptually, it is similar to the DeepSet architecture [47], but each layer is not necessarily a linear equivariant layer. Unlike the Transformer, the SumFormer takes linear

time per forward pass as computing $Z$ takes $\Theta(n)$ time while computing each $\psi(x_i, Z)$ takes only constant time, assuming $d'$ and the dimensions of $\mathcal{X}$ and $\mathcal{Y}$ are constant.

**$\varepsilon$-kernels and coresets.** A classical technique in developing approximation algorithms for geometric problems is via the extraction of a certain small representative subset of the input object, called $\varepsilon$-coresets, and solving the problem only on the small subset. A popular coreset construction is via the $\varepsilon$-kernels. First, given a set of points $P$ and a direction $u \in \mathbf{S}^{d-1}$ with $\mathbf{S}^{d-1}$ denoting the unit-sphere in $\mathbb{R}^d$, the **directional interval** $\overrightarrow{\mathsf{w}}_u(P)$ and **directional width** $\mathsf{w}_u(P)$ w.r.t. $u$ are defined as:

$$\overrightarrow{\mathsf{w}}_u(P) = \left[ \min_{p \in P} \langle u, p \rangle, \max_{p \in P} \langle u, p \rangle \right]; \text{ and } \mathsf{w}_u(P) = \max_{p \in P} \langle u, p \rangle - \min_{p \in P} \langle u, p \rangle = \text{length of } \overrightarrow{\mathsf{w}}_u(P).$$

**Definition 2.2** ($\varepsilon$-kernel). *Given a set of point $P \subseteq \mathbb{R}^d$, an $\varepsilon$-kernel for $P$ is **a subset of points** $Q \subseteq P$ such that for any $u \in \mathbf{S}^{d-1}$, $\mathsf{w}_u(Q)$ is an $\varepsilon$-approximation of $\mathsf{w}_u(P)$; i.e., $(1 - \varepsilon)\mathsf{w}_u(P) \le \mathsf{w}_u(Q)$.*

Hence, an $\varepsilon$-kernel $Q$ of $P$ approximates the width of $P$ along any direction $u$. In this paper, we assume that any input point set $P$ is contained within a bounded compact set in $\mathbb{R}^d$; in fact, w.l.o.g., assume each $P$ is contained in the $d$-D hypercube $\mathbb{C}_d = [-1, 1]^d$. Let $\text{CH}(P)$ denote the *convex hull* of $P$ consisting of all convex combinations of points in $P$. Our theoretical results in this paper will assume that each point set $P$ is not too "skinny", in the sense that it is "fat": A point set $P \subseteq \mathbb{R}^d$ is $\alpha$-**fat** if there exist $0 < \alpha \le 1$ and $\vec{o} \in \mathbb{R}^d$ such that $\vec{o} + \alpha\mathbb{C}_d \subseteq \text{CH}(P) \subseteq \vec{o} + \mathbb{C}_d$.

We now explain the utility of $\varepsilon$-kernels for extent measures. An **extent measure** $\mu$ is a function that maps any point set $P \subset \mathbb{R}^d$ to a real value such that $\mu(P') \le \mu(P)$ for all $P' \subset P$. Examples include the diameter of point set $P$, the minimum radius of any ball covering $P$ (i.e, the radius of the minimum enclosing ball of $P$), the minimum width of any slab (parallel hyperplane) containing $P$, and the minimum width of any spherical or cylindrical shells containing $P$.

**Definition 2.3** (Coresets and faithful extent measures). *$Q \subseteq P$ is an $\varepsilon$-**coreset for** $P$ **w.r.t.** $\mu$ if $\mu(P) \le (1 + \varepsilon)\mu(Q)$. An extent measure $\mu$ is **faithful** if there exists $c > 0$ such that every $\varepsilon$-kernel $Q$ of $P$ is a $c\varepsilon$-coreset for $\mu$.*

Note that $\varepsilon$-kernel is defined w.r.t. directional width; while $\varepsilon$-coreset is defined for a target extent measure. Intuitively, $\mu$ is a faithful measure if any $\varepsilon$-kernel of $P$ serves as a valid $\varepsilon$-coreset for $P$ w.r.t. this measure $\mu$. Examples of faithful measures include diameter, smallest enclosing-ball radius, min-width of any enclosing slab, and bounding-box volume. For a faithful measure $\mu$, once an $\varepsilon$-kernel $Q$ is computed for $P$, evaluating $\mu(Q)$ yields a $\varepsilon$-approximation to $\mu(P)$ in time *depending only on* $|Q|$. In short, computing $\varepsilon$-kernels leads to approximations of faithful extent measures. In the next Section 4, we will discuss approximations of *unfaithful extent measures*.

**Motivating example.** Before introducing our neural architecture, we provide a motivating example: the minimum enclosing ball. Given a set of points $P \subseteq \mathbb{R}^d$, the minimum enclosing ball is the smallest-radius ball that contains all of $P$. In Figure 2, we compare two natural end-to-end neural network approaches, one using a *Sumformer* and the other a *Transformer*, to directly predict the radius of the minimum enclosing ball from the input point set. We also compare with the algorithmically aligned encoder-processor-decoder network that we propose later in the paper. Each model is trained with point clouds of size 100 sampled uniformly from $[-5, 5]^2$ and tested on point clouds of increasing sizes and bounding boxes (see Appendix for details).

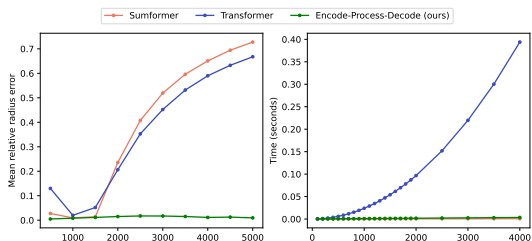

Figure 2: Comparison of error (left) and inference time (right) of the Transformer, Sumformer, and encode-process-decode approaches for estimating the radius of the minimum enclosing ball.

The error is shown on the left side of Figure 2 and we see that *neither* end-to-end approach generalize across sizes since the error grows rapidly as the size of the point cloud increases. In contrast, our encode-process-decode approach maintains low error across sizes as its error rate remains flat. On the right side of Figure 2, we see that our new encode-process-decode approach is as efficient as Sumformer (time complexity grows linearly), while Transformer grows quadratically. In what follows, we show how the encode-process-decode aligns with the $\varepsilon$-kernel and can therefore approximate extent measures with bounded model complexity.

# 3 Neural approximations of relaxed-$\varepsilon$-kernels

The previous section states that for faithful measures, computing $\varepsilon$-kernels leads to their approximation for any input point-set $P$. Hence intuitively, we wish to have a neural architecture to compute $\varepsilon$-kernels. It turns out that existing algorithms to construct $\varepsilon$-kernels all require being able to identify the exact extreme point(s) in various projects, which does not easily align with Transformer nor SumFormer. Instead, we will propose a variant of the classical $\varepsilon$-kernels, called *the relaxed-$\varepsilon$-kernel*, which shares similar theoretical guarantees as $\varepsilon$-kernels. We then develop a simple Sumformer-based neural model that can efficiently approximate a relaxed-$\varepsilon$-kernel with *bounded model complexity*.

---

**Algorithm 1** Relaxed-$\varepsilon$-kernels

**Require:** $P = \{p_1, \ldots, p_n\} \subseteq \mathbb{R}^d$, $k$
1: $\Omega \leftarrow$ set of $k$ directions from $\mathbf{S}^{d-1}$
2: $Q \leftarrow \{\}$
3: **for** $u \in \Omega$ **do**
4:     $A_u \leftarrow \{\langle u, p \rangle : p \in P\}$
5:     $w(A_u) \leftarrow \max A_u - \min A_u$
6:     Compute $\rho_\varepsilon(A_u)$
7:     $q_u \leftarrow (\sigma(\rho_\varepsilon(A_u)))^T P$
8:     $Q.\text{append}(q_u)$
9: **end for**
10: **return** $Q$

Set $M_{A_u} := \max(A_u) = \max_{p \in P} \langle u, p \rangle$

$$\rho_\varepsilon(A_u) = \begin{pmatrix} \text{ReLU}(\langle u, p_1 \rangle + \varepsilon w(A_u) - M_{A_u}) \\ \text{ReLU}(\langle u, p_2 \rangle + \varepsilon w(A_u) - M_{A_u}) \\ \vdots \\ \text{ReLU}(\langle u, p_n \rangle + \varepsilon w(A_u) - M_{A_u}) \end{pmatrix}$$

Let $\sigma : \mathbb{R}^n \to \mathbb{R}^n$ such that

$$\sigma(x) = \left[ \frac{e^{x_1} - 1}{\sum_i^n (e^{x_i} - 1)} \cdots \frac{e^{x_n} - 1}{\sum_i^n (e^{x_i} - 1)} \right]^T$$

Figure 3: An algorithm for computing relaxed $\varepsilon$-kernels. The functions $\rho_\varepsilon$ and $\sigma$ are defined on the right. Notice that $\sigma$ is a modified softmax function.

## 3.1 Relaxed $\varepsilon$-kernels

**Definition 3.1** (Relaxed-$\varepsilon$-kernel)**.** *Given a set of points $P \subseteq \mathbb{R}^d$, a relaxed $\varepsilon$-kernel is a set of points $Q \subseteq CH(P)$ such that for any $u \in \mathbf{S}^{d-1}$, we have $(1 - \varepsilon)w_u(P) \leq w_u(Q)$.*

Unlike a standard $\varepsilon$-kernel, where the output points must be a subset of $P$, the relaxed-$\varepsilon$-kernel allows each output to be *convex combinations* of input points (as guaranteed by $Q \subset CH(P)$). See Figure 4 for a visualization. The key is to show the relaxed-$\varepsilon$-kernels can replace $\varepsilon$-kernels in approximating faithful extent measures. Indeed, this is guaranteed by the following theorem (through the relaxed $\varepsilon$-coreset). The proofs are in Appendix A.3.1. In addition to the relaxed-$\varepsilon$-kernel, we define an analogous relaxed-$\varepsilon$-coreset and show that it has the same approximation guarantees for faithful measures $\mu$. Specifically, a **relaxed $\varepsilon$-coreset** of $P$ w.r.t $\mu$ is a set of points $Q \subseteq \mathbf{CH}(P)$ (versus $Q \subset P$) such that $(1-\varepsilon)\mu(Q) \leq \mu(P) \leq (1 + \varepsilon)\mu(Q)$.

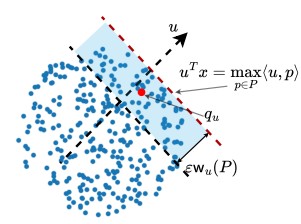

Figure 4: $q_u$ is the convex combination of points in the shaded region.

**Theorem 3.2.** *Let $\mu$ be a faithful measure with constant $c$. Let $0 < \varepsilon < \frac{1}{6c}$. There exists $c'$ such that for any $P$, any relaxed $\varepsilon$-kernel $Q$ for $P$ is a relaxed $c'\varepsilon$-coreset for $P$.*

**A NN-aligned relaxed-$\varepsilon$-kernel construction.** A relaxed-$\varepsilon$-kernel for $P$ can be computed by Algorithm 1. Intuitively, we pick a fixed set of directions $\Omega$ and for each $u \in \Omega$, we output a point $q_u \in \mathbb{R}^d$ which is a convex combination of points in $P$ whose projections onto $u$ are within $\varepsilon w_u(P)$ of the maximum. $\rho_\varepsilon(A_u)$ computes an indicator vector for such points, and $(\sigma(\rho_\varepsilon(A_u)))^T P$ then computes $q_u$, where $\sigma$ is the modified softmax function described in Algorithm 1 and $q_u$ is a convex combination of the points in $P$.

Theorem 3.3 (proof in Appendix A.3.1) below shows that the quality of the output of Algorithm 1 depends on the "fatness" of the input point set $P$. We also need to choose an appropriate set of

directions $\Omega$: in particular, $\Omega$ is a $\frac{\alpha\varepsilon}{4\sqrt{d}}$-net for all possible unit directions $\mathbf{S}^{d-1}$. Note that a set $A$ is a $\delta$-net of a metric space $(X, d_X)$ if $A \subset X$ and for any point $x \in X$, there exists $a \in A$ such that $d_X(x, a) \leq \delta$. It is easy to construct a $\delta$-net of $\mathbf{S}^{d-1}$ of size $O(1/\delta^{d-1})$ where the distance between two directions is measured as the angle between them.

**Theorem 3.3.** *Let $P \subset [-1, 1]^d$ be a $\alpha$-fat point set and let $0 < \varepsilon < \frac{1}{3}$. Suppose $\Omega^+$ is an $\frac{\alpha\varepsilon}{4\sqrt{d}}$-net for $\mathbf{S}^{d-1}$ s.t. $|\Omega| = O\left(\frac{1}{(\frac{\alpha\varepsilon}{4\sqrt{d}})^{d-1}}\right)$. Let $\Omega^- = \{-u : u \in \Omega^+\}$ and let $\Omega = \Omega^- \cup \Omega^+$ so for all $u \in \Omega$, $-u \in \Omega$. Then the set of points $Q$ constructed via Algorithm 1 is a relaxed-$3\varepsilon$-kernel of $P$.*

### 3.2 Relaxed-$\varepsilon$-kernel approximating NN architecture

Given an arbitrary point set $P \subseteq \mathbb{C}_d$, we will represent $P \in \mathbb{R}^{N \times d}$ as a sequence of $d$-vectors where $N = |P|$. Given $X \in \mathbb{R}^{N \times k}$, let $\sigma_{\text{col}}$ denote column-wise application of the modified softmax normalization described in Algorithm 1 (i.e. we apply $\sigma$ to each column in $X$). First, we introduce a general neural framework of a specific form: given $P \in \mathbb{R}^{N \times d}$, it will output $k$ points in $\mathbb{R}^d$; that is, $\mathcal{N}_{\phi_\varepsilon}(P) \in \mathbb{R}^{k \times d}$, as follows:

$$\mathcal{N}_{\phi_\varepsilon}(P) = \sigma_{\text{col}}(\phi_\varepsilon(P))^T P, \tag{1}$$

where $\phi_\varepsilon$ is a sequence-to-sequence permutation equivariant model that maps a sequence of points in $\mathbb{R}^d$ to a sequence of points in $\mathbb{R}^k$. Note that regardless of the length of the input sequence $P$ (i.e., $N$), the output $\mathcal{N}_{\phi_\varepsilon}(P)$ is always in $\mathbb{R}^{k \times d}$, viewed as $k$ points in $\mathbb{R}^d$. The model $\phi_\varepsilon$ can be instantiated with any permutation equivariant neural network: such as a Sumformer $\mathcal{S}_\varepsilon$ or a Transformer $\mathcal{T}_\varepsilon$. The resulting final model $\mathcal{N}_{\phi_\varepsilon}$ is then denoted by $\mathcal{N}_{\mathcal{S}_\varepsilon}(P) = \sigma_{\text{col}}(\mathcal{S}_\varepsilon(P))^T P$ for the SumFormer instantiation or $\mathcal{N}_{\mathcal{T}_\varepsilon}(P) = \sigma_{\text{col}}(\mathcal{T}_\varepsilon(P))^T P$ for the Transformer instantiation. See Figure 1 (a). We will use the Sumformer implementation $\mathcal{N}_{\mathcal{S}_\varepsilon}$, as the following result shows that this simple model is sufficient to align with the computation of a relaxed-$\varepsilon$-kernel.

**Theorem 3.4.** *Given any $\varepsilon, \alpha > 0$, there exists a SumFormer $\mathcal{S}_\varepsilon$ with only a single block, and bounded model complexity (depending only on $\varepsilon$, $\alpha$, and the input dimension $d$) such that, for any $\alpha$-fat point clouds $P \subset \mathbb{R}^d$ of arbitrary size, $\mathcal{N}_{\mathcal{S}_\varepsilon}(P)$ is a relaxed-$\varepsilon$-kernel for $P$.*

Intuitively, the SumFormer $\mathcal{S}_\varepsilon(P) \in \mathbb{R}^{N \times k}$ can easily simulate the computation of the collection of $\rho_\varepsilon(A_u)$'s, for all $k$ number of $u \in \Omega$, as in Algorithm 1. Line 7 of Algorithm 1 is then computed by the outer columnwise-softmax normalization as in Eqn. (1), thereby outputting $k$ points in $\mathbb{R}^d$ which form a relaxed-$\varepsilon$-kernel for input point set $P$. We refer to the neural network model $\mathcal{N}_{\mathcal{S}_\varepsilon}$ in Theorem 3.4 as our *relaxed-$\varepsilon$-kernel-approximating-neural network*, abbrev. *kernel-approx-NN*.

With our kernel-approx-NN, it is now easy to construct a bounded-complexity neural network to approximate *faithful extent measures*. Consider any faithful extent measure $\mu$: For any input point set $P$, once we are given a relaxed-$\varepsilon$-kernel $Q$ of $P$, we can simply compute $\mu(Q)$ as an approximation of $\mu(P)$. By Theorem 3.4, the $k$ points $\mathcal{N}_{\phi_\varepsilon}(P) \in \mathbb{R}^{k \times d}$ form a relaxed-$\varepsilon$-kernel of $P$, for a suitable $\varepsilon$ depending on the output size $k$. By Definition 2.3, we can compute $\mu(\mathcal{N}_{\phi_\varepsilon}(P))$ as an approximation of $\mu(P)$. Note that $\mu$ can be viewed as a **permutation-invariant function** as its output does not change as the ordering of input points change. Hence we can use a DeepSet architecture [47] to approximate $\mu$: in particular, we set our *faithful-measure-approximating NN* as $\mathcal{N}_{\text{faithful}}(P) = \mathcal{N}_{\text{deepset}}(\mathcal{N}_{\phi_\varepsilon}(P))$. Deepset models are universal approximators for permutation-invariant functions where the model complexity depends on the maximum input size (e.g, [40, 34]) – in our case, the input is of size $k \times d$ ($k$ points in $\mathbb{R}^d$ as computed by our kernel-approx-NN $\mathcal{N}_{\mathcal{S}_\varepsilon}$). Hence the model complexity of $\mathcal{N}_{\text{deepset}}$ is also bounded, depending only $k$, thus only on $\varepsilon, \alpha$ and $d$, and not on the size of original input point set $P$. Indeed, the following corollary implies that we have a NN with bounded complexity that can approximate extent measures such as diameter, radius of minimum-enclosing ball, and width of minimum-width slab, for input point set of arbitrary size.

**Corollary 3.5.** *For any $\varepsilon > 0, \alpha > 0$ and a faithful extent measure $\mu$ with constant $c$ as in Definition 2.3, there exists a bounded-complexity NN of the form $\mathcal{N}_{\text{faithful}}(P) = \mathcal{N}_{\text{deepset}}(\mathcal{N}_{\phi_\varepsilon}(P))$, such that for any $\alpha$-fat point set $P$, the output $\mathcal{N}_{\text{faithful}}(P)$ satisfies that:*

$$(1 - 3c\varepsilon)\mu(P) - \varepsilon \leq \mathcal{N}_{\text{faithful}}(P) \leq (1 + 3c\varepsilon)\mu(P) + \varepsilon$$

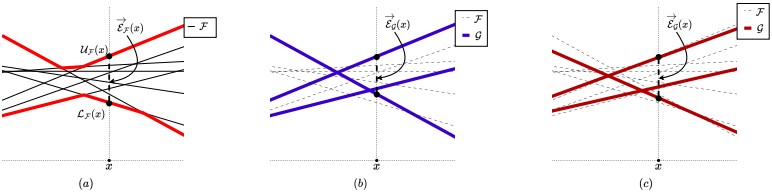

Figure 5: (a) Upper / lower envelopes and the extent vector for a family of linear functions. Thick curves in (b) form an $\varepsilon$-kernel $\mathcal{G}$ of $\mathcal{F}$ and those thick curves in (c) is a relaxed-$\varepsilon$-kernel. Dashed lines are those in $\mathcal{F}$. Note that, unlike (b), a relaxed-$\varepsilon$-kernel does not need to be a subset of $\mathcal{F}$.

# 4    Neural approximations of unfaithful extent measures

The $\varepsilon$-kernel framework can also be used to derive efficient approximations for a certain family of unfaithful measures via the so-called "linearization" trick [2]. We briefly describe the high-level idea, which follows the same framework as [2]. We will show that our relaxed-$\varepsilon$-kernel can replace the role of $\varepsilon$-kernel in this approach. First, we need to define (relaxed-)$\varepsilon$-kernels for *sets of functions* (instead of points). The way to handle those unfaithful extent measures is to first convert them to higher-dimensional **linear functions** via the linearization technique. Specifically, given a set of real-valued functions $\mathcal{F} = \{f_1, \ldots, f_n : \mathbb{R}^{d-1} \to \mathbb{R}\}$, its *upper envelope* is the function $U_\mathcal{F} : \mathbb{R}^{d-1} \to \mathbb{R}$ where $U_\mathcal{F}(x) = \max_{f \in \mathcal{F}} f(x)$, while its *lower envelope* is defined by $L_\mathcal{F}(x) = \min_{f \in \mathcal{F}} f(x)$.

**Definition 4.1.** *Given a set of functions $\mathcal{F} = \{f_1, \ldots, f_n\}$, where $f_i : \mathbb{R}^{d-1} \to \mathbb{R}$, its **extent** is $\mathcal{E}_\mathcal{F}(x) = U_\mathcal{F}(x) - L_\mathcal{F}(x)$ and its **extent-interval** is $\overrightarrow{\mathcal{E}}_\mathcal{F}(x) = [L_\mathcal{F}(x), U_\mathcal{F}(x)]$. An $\varepsilon$-**kernel** for $\mathcal{F}$ is a set $\mathcal{G} \subseteq \mathcal{F}$ such that $\mathcal{E}_\mathcal{G} \geq (1 - \varepsilon)\mathcal{E}_\mathcal{F}$.*

See Figure 5. We define analogous concepts for relaxed-$\varepsilon$-kernels. Given two intervals $I_1 = [a, b]$ and $I_2 = [c, d]$, the *Hausdorff distance* between them equals $\max\{|a - c|, |b - d|\}$.

**Definition 4.2** (Relaxed-$\varepsilon$-kernels for sets of functions)**.** *A relaxed $\varepsilon$-kernel for $\mathcal{F}$ is a **set** $\mathcal{G}$ of functions (not necessarily subset of $\mathcal{F}$) such that for any $x \in \mathbb{R}^{d-1}$, we have that (i) $\overrightarrow{\mathcal{E}}_\mathcal{G}(x) \subseteq \overrightarrow{\mathcal{E}}_\mathcal{F}(x)$ and (ii) the Hausdorff distance between $\overrightarrow{\mathcal{E}}_\mathcal{G}(x)$ and $\overrightarrow{\mathcal{E}}_\mathcal{F}(x)$ is at most $\varepsilon \cdot \mathcal{E}_\mathcal{F}(x)$.*

Now consider a linear function $f(x) = a_1 x_1 + \cdots + a_{d-1} x_{d-1} + a_d$, where $x = [x_1, \ldots, x_{d-1}]^T \in \mathbb{R}^{d-1}$: This linear function represents a hyperplane in $\mathbb{R}^d$. Its dual is the point $f^* := [a_1, \ldots, a_d]^T \in \mathbb{R}^d$. Symmetrically, given a point $p = [p_1, \ldots, p_d]^T \in \mathbb{R}^d$, its dual hyperplane is the linear function $p^*(x) = p_1 x_1 + \ldots + p_{d-1} x_{d-1} + p_d$. Given a set of **linear functions** $\mathcal{F} = \{f_1, \ldots, f_n\}$, let $\mathcal{F}^* = \{f_1^*, \ldots, f_n^*\} \subset \mathbb{R}^d$ denote its dual point set. Note that $(f^*)^* = f$ and $(\mathcal{F}^*)^* = \mathcal{F}$.

**Theorem 4.3.** *$\mathcal{G}$ is a relaxed $\varepsilon$-kernel for the set of linear functions $\mathcal{F}$ if and only if $\mathcal{G}^*$ is a relaxed $\varepsilon$-kernel for the point set $\mathcal{F}^*$.*

Often we want to find the extent or $\varepsilon$-kernels for sets of non-linear functions. Depending on the functions, it is possible to *linearize* them as follows [2]: Suppose $\mathcal{F}$ is a set of $(d + p)$-variate polynomials $\mathcal{F} = \{f_i(x) = f(x, a_i) : i \in [N], x \in \mathbb{R}^d, a_i \in \mathbb{R}^p\}$. We say that $\mathcal{F}$ *admits a linearization of dimension $m$* if there exist $p$-variate polynomials $\psi_0, \ldots, \psi_m : \mathbb{R}^p \to \mathbb{R}$ and $d$-variant polynomials $\varphi_1, \ldots, \varphi_m : \mathbb{R}^d \to \mathbb{R}$ such that for each $f_i$, we can write it as

$$f_i(x) = f(x, a_i) = \psi_0(a_i) + \psi_1(a_i)\varphi_1(x) + \cdots + \psi_k(a_i)\varphi_m(x).$$

Now define functions $\Psi : \mathbb{R}^p \to \mathbb{R}^{m+1}$ and $\Phi : \mathbb{R}^d \to \mathbb{R}^{m+1}$ as $\Psi(a) = [\psi_0(a), \ldots, \psi_m(a)]^T$ for any $a \in \mathbb{R}^p$ and $\Phi(x) = [1, \varphi_1(x), \ldots, \varphi_m(x)]^T$ for any $x \in \mathbb{R}^d$. Then we can view $f_i$ as a new **linearized form** $f(x, a_i) = \Psi(a_i)^T \Phi(x)$ in space $\mathbb{R}^{m+1}$. For standard $\varepsilon$-kernels, finding the $\varepsilon$-kernel for the dual of the set of linearized functions gives an $\varepsilon$-kernel for the set of original functions. In Theorem 4.4 we prove that we can also construct a relaxed-$\varepsilon$-kernel for a set of functions $\mathcal{F}$ from an $\varepsilon$-kernel of the dual set after linearization.

**Theorem 4.4.** *Suppose $\mathcal{F}$ is a set of $N$ number of $(d + p)$-variate polynomials which admit a linearization of dimension $m$; i.e. there is a set of $A = \{a_1, \ldots, a_N\} \subseteq \mathbb{R}^p$, $\Psi : \mathbb{R}^p \to \mathbb{R}^{m+1}$ and $\Phi : \mathbb{R}^d \to \mathbb{R}^{m+1}$, such that $\mathcal{F} = \{f_i(x) = \Psi(a_i)^T \Phi(x) : a_i \in A, i \in [N]\}$, and the dual $\mathcal{F}^* = \{\Psi(a_i) : i \in [N]\}$. Let $\mathcal{Q}_\varepsilon^* \subseteq \mathbb{R}^{m+1}$ be a relaxed-$\varepsilon$-kernel for $\mathcal{F}^*$ of size $k$ computed via Alg. 1. Suppose $\mathcal{Q}_\varepsilon^* = \{\alpha_1^j \Psi(a_1) + \cdots + \alpha_N^j \Psi(a_N) : j \in [k], \sum \alpha_\ell^j = 1, \alpha_\ell^j \geq 0\}$. Then the set of functions $\mathcal{Q}_\varepsilon := \{q^T \Phi(x) : q \in \mathcal{Q}_\varepsilon^*\}$ is a relaxed-$\varepsilon$-kernel for the original set of functions $\mathcal{F}$.*

Finally, as shown in [2], a family of unfaithful extent measures can be transformed to finding the *smallest extent* $\mathcal{E}_{\mathcal{F}}^* = min_x \mathcal{E}_{\mathcal{F}}(x)$ of a set of $N$ functions $\mathcal{F}$ (constructed from the input pointset $P$ with $N = |P|$) that admit a linearization, and thereby can be approximated by computing the smallest extent of the relaxed-$\varepsilon$-kernel $\mathcal{Q}_\varepsilon$ as computed in Theorem 4.4. Once $\mathcal{Q}_\varepsilon$ is computed, finding the smallest extent $\mathcal{E}_{\mathcal{Q}_\varepsilon}^*$ takes time only depending on $|\mathcal{Q}_\varepsilon|$ and indepent to the original pointset size $P$. We provide an example in Appendix A.2 of how relaxed-$\varepsilon$-kernels can be used to approximate the width of the minimum enclosing annulus for a point cloud.

**Neural framework for approximating unfaithful extent measures.** In order to approximate those unfaithful extent measures that can be solved via linearization, we make use of the encode-process-decode framework proposed in [15]. Given $\mathcal{X}$ which is the space of point clouds, a point cloud $P \in \mathcal{X}$, and an extent measure $\mu : \mathcal{X} \to \mathbb{R}^d$, we define

$$\mathcal{N}_{\text{extent}}(P) = \mathcal{N}_D(\mathcal{N}_P(\mathcal{N}_E(P))) \tag{2}$$

Intuitively, given a size-$N$ pointset $P \in \mathbb{R}^{N \times d}$, the encoder $\widehat{P} = \mathcal{N}_E(P) \in \mathbb{R}^{N \times (m+1)}$ computes the linearization which maps $P$ to a size-$N$ (dual) point set in $\mathbb{R}^{m+1}$. The process-network $\mathcal{N}_P$ then outputs a size-$k$ relaxed-$\varepsilon$-kernel, and the decoder $\mathcal{N}_D$ network computes the final approximation of the extent measure. Importantly, for the process-network $\mathcal{N}_P$, we use our $\varepsilon$-kernel approximating neural network $\mathcal{N}_{\phi_\varepsilon}$ defined in Equation (1), and given $\widehat{P} = \mathcal{N}_E(P)$, the output of processor $\mathcal{N}_P(\widehat{P}) \in \mathbb{R}^{k \times (m+1)}$ is a size-$k$ relaxed-$\varepsilon$-kernel for $\widehat{P}$. See a visualization of this architecture in Figure 1 and note that this is exactly the encode-process-decode framework from our motivating example in Figure 2. Note that the encoder $\mathcal{N}_E$ can be any **permutation equivariant** architecture. In our experiments, we simply apply an MLP to each element of $P$ so $\mathcal{N}_E([p_1, \ldots, p_N]) = [\text{MLP}(p_1), \ldots, \text{MLP}(p_N)]$. The decoder $\mathcal{N}_D$ can be implemented by any universal **permutation invariant** architecture. In our experiments, we use DeepSet as the final decoder, $\mathcal{N}_D$, to compute the final approximation of the extent architecture. Note that we can also use $\mathcal{N}_{\text{extent}}$ to approximate faithful extent measures as $\mathcal{N}_E$ can be set to the identity function. In our experiments, we use this encoder-process-decoder $\mathcal{N}_{\text{extent}}$ to approximate both faithful and unfaithful extent measures.

## 5 Experiments

We evaluate both our kernel-approx-NN (cf. Eqn. (1)) to approximate relaxed-$\varepsilon$-kernels as well as our encode-process-decode framework (cf. Eqn. (2)) for extent measure approximation tasks (faithful and unfaithful). Throughout this section, we report the best results after a hyperparameter search. All details regarding training and datasets are given in Appendix A.4.

Our experiments use (i) two synthetic datasets 'Gaussian Mixture' (point clouds from 2 to 5 random Gaussian clusters), and 'Mixed Synthetic' (a mixture of synthetically generated point clouds, including 'Ellipse', 'Gaussian Mixture' and more), and (ii) two real datasets, SQUID [26] (2D point clouds generated from boundary contours of fish) and the 3D ModelNet [43]. In the main text, we present average evaluation metrics over all test data; the Appendix provides complete results, including standard deviations, and further experiments demonstrating that fixed-size models continue to perform well on OOD point sets (across varied sampling schemes and sizes).

**Relaxed-$\varepsilon$-kernel approximation.** We examine the performance of the kernel-approx-NN (cf. Equation (1)) using both Sumformers and Transformers as $\phi_\varepsilon$ in $\mathcal{N}_{\phi_\varepsilon}$; and we denote these different instantiations as $\mathcal{S}_\varepsilon$ and $T_\varepsilon$, respectively. We also compare against a direct neural baseline that directly outputs a coarsened point cloud aggregating encoded point features and passing the result through an MLP to produce the final output point cloud. The Sumformer and Transformer implementations of this baseline are denoted $\mathcal{S}_{\varepsilon,\text{B}}$ and $\mathcal{T}_{\varepsilon,\text{B}}$, respectively. We also include a comparison to the classical $\varepsilon$-kernel and our relaxed $\varepsilon$-kernel algorithms, both with $\varepsilon = 0.1$, denoted $\mathcal{B}_{\varepsilon,E}$ and $\mathcal{B}_{\varepsilon,R}$, respectively. For performance, we use the *directional width error* to measure the error caused by our approximation: $\mathcal{E}_{\text{dir}}(P, \tilde{Q}) = \max_{u \in \Omega} \frac{|w_u(P) - w_u(\tilde{Q})|}{w_u(P,u)}$ where $P$ is the input point set, $\tilde{Q}$ is the output of our kernel-approx-NN, and $\Omega$ is a set of 1000 random directions. We report results for datasets in $\mathbb{R}^2$ and in $\mathbb{R}^3$ in Table 1. Additional results for point clouds in $\mathbb{R}^5$, as well as many other experiments including effects of hyperparameters (e.g. width of latent dimension), are given in the Appendix. Overall, using Sumformer instantiation $\mathcal{S}_\varepsilon$ as $\mathcal{N}_{\phi_\varepsilon}$ consistently yields lower error on OOD data than the Transformer-instantiation $\mathcal{T}_\varepsilon$. This gap is most striking on real world datasets: when trained on the synthetic datasets ('Mixed Synthetic or 'Gaussian Mixture'), the $\mathcal{S}_\varepsilon$ has significantly lower error

Table 1: $\mathcal{E}_{\text{dir}}$ for predicted relaxed-$\varepsilon$-kernels of size 16 and 64 respectively, for point clouds in $\mathbb{R}^2$ (left) and $\mathbb{R}^3$ (right) across three different test sets for each train set. 'In-dist.' refers to in-distribution test sets. "Mixed Synthetic" has no "OOD, Synthetic" because the train set contains all types of synthetic data. Similarly, the models trained on SQUID and ModelNet are not OOD on those test sets (so unreported). Since algorithmic baselines do not require training, they have no 'In-dist' error. Boldface indicates the lower error (better accuracy).

| | | Test Sets | | | | | Test Sets | | |
| | | | OOD | | | | | OOD | |
| **Train Set** | **Method** | In-dist. | Synthetic | SQUID | **Train Set** | **Method** | In-dist. | Synthetic | ModelNet |
|---|---|---|---|---|---|---|---|---|---|
| Gaussian Mixture | $\mathcal{S}_\varepsilon$ | **0.014** | **0.043** | **0.028** | Gaussian Mixture | $\mathcal{S}_\varepsilon$ | **0.047** | **0.039** | **0.064** |
| | $\mathcal{T}_\varepsilon$ | 0.027 | 0.097 | 0.218 | | $\mathcal{T}_\varepsilon$ | 0.054 | 0.066 | 0.250 |
| | $\mathcal{S}_{\varepsilon,\text{B}}$ | 0.146 | 0.146 | 0.962 | | $\mathcal{S}_{\varepsilon,\text{B}}$ | 0.274 | 0.223 | 1.949 |
| | $\mathcal{T}_{\varepsilon,\text{B}}$ | 0.190 | 0.383 | 0.94 | | $\mathcal{T}_{\varepsilon,\text{B}}$ | 0.357 | 0.803 | 5.514 |
| SQUID | $\mathcal{S}_\varepsilon$ | **0.028** | 0.504 | - | ModelNet | $\mathcal{S}_\varepsilon$ | **0.041** | **0.099** | - |
| | $\mathcal{T}_\varepsilon$ | 0.061 | **0.400** | - | | $\mathcal{T}_\varepsilon$ | 0.049 | 0.265 | - |
| | $\mathcal{S}_{\varepsilon,\text{B}}$ | 0.421 | 7.137 | - | | $\mathcal{S}_{\varepsilon,\text{B}}$ | 0.379 | 0.528 | - |
| | $\mathcal{T}_{\varepsilon,\text{B}}$ | 0.228 | 14.561 | - | | $\mathcal{T}_{\varepsilon,\text{B}}$ | 0.542 | 0.581 | - |
| Mixed Synthetic | $\mathcal{S}_\varepsilon$ | **0.027** | – | **0.065** | Mixed Synthetic | $\mathcal{S}_\varepsilon$ | **0.035** | – | **0.055** |
| | $\mathcal{T}_\varepsilon$ | 0.032 | – | 0.296 | | $\mathcal{T}_\varepsilon$ | 0.042 | – | 0.117 |
| | $\mathcal{S}_{\varepsilon,\text{B}}$ | 0.094 | - | 0.97 | | $\mathcal{S}_{\varepsilon,\text{B}}$ | 0.174 | - | 2.288 |
| | $\mathcal{T}_{\varepsilon,\text{B}}$ | 0.203 | - | 0.939 | | $\mathcal{T}_{\varepsilon,\text{B}}$ | 0.289 | - | 4.356 |
| Algorithmic Baselines | $\mathcal{B}_{\varepsilon,R}$ | - | 0.180 | 0.228 | Algorithmic Baselines | $\mathcal{B}_{\varepsilon,R}$ | - | 1.016 | 2.176 |
| | $\mathcal{B}_{\varepsilon,E}$ | - | 0.040 | 0.062 | | $\mathcal{B}_{\varepsilon,E}$ | - | 0.062 | 0.050 |

than $\mathcal{T}_\varepsilon$ when tested on OOD samples from SQUID (in $\mathbb{R}^2$) and ModelNet (in $\mathbb{R}^3$). While $\mathcal{T}_\varepsilon$ also has the capacity to implement Algorithm 1, we believe that the simple and more direct alignment of $\mathcal{S}_\varepsilon$ with this algorithm leads to faster training (optimization) and better (OOD) generalization.

**Approximating extent measures.** We train the encode–process–decode model (Eq. 2) to predict three extent measures: radius of minimum enclosing ball (MEB), radii (and therefore area) of minimum enclosing ellipse (MEE), and width of minimum-width annulus (MEA) covering input points. The first two are faithful extent measures, while the last is unfaithful. We compare two instantiations of the general model $\mathcal{N}_{\text{extent}}$: (i) $\mathcal{S}_{\text{extent}}$ when using the Sumformer within the processor model $\mathcal{N}_{\text{P}}$, and (ii) $\mathcal{T}_{\text{extent}}$ when using the Transformer.

To examine the contribution of the relaxed-$\epsilon$-kernel module when used as $\mathcal{N}_{\phi_\varepsilon}$, we compare two training protocols: (1) full end-to-end optimization of the entire pipeline (denoted as 'E2E') and (2) a regime in which the inner kernel-approx-NN is fixed after being pre-trained to output a relaxed-$\varepsilon$-kernel (denoted as 'Frozen'), and then only the encoder and decoder are trained based on the specific task. To understand the effect of mapping input point to a relaxed-$\varepsilon$-kernel in the processor module, we also construct a baseline where we do not use Equation (1) to compute a coreset. Instead, we use the output $\phi_\varepsilon(\widehat{P})$ (where $\widehat{P} = \mathcal{N}_{\text{E}}(P)$) and feed it to the decoder $\mathcal{N}_{\text{D}}$. $\mathcal{S}_{\text{Baseline}}$ and $\mathcal{T}_{\text{Baseline}}$ to denote the Sumformer and Transformer instantiations of this baseline model. Additionally, we also compare against the proposed direct neural approximations described in Section 2, with Sumformer and Transformer implementations denoted as $\mathcal{S}_{\text{Direct}}$ and $\mathcal{T}_{\text{Direct}}$. An extended comparison to classical algorithmic baselines in terms of both accuracy and runtime is also included in Appendix A.4.

The results for all three extent measures are shown in Table 2 and we evaluate the performance as follows: (1) For MEB, we measure the relative error $\mathcal{E}_{\text{r}}$ of predicted minimum-radius over the true optimal. (2) For MEE, we use the relative errors of both predicted major and minor radii (denoted as $\mathcal{E}_{\text{r},maj}$ for the major radius and $\mathcal{E}_{\text{r},min}$ for the minor radius). (3) For the MEA, we use the relative error for the predicted smallest width $\mathcal{E}_{\text{w}}$. To measure the quality of our returned geometric shape, we also compute $\mathcal{E}_{\text{p}}$, the percentage of input points not covered by these approximations. These results are in the Appendix and usually this portion is very small (around 1-2%).

All results in Table 2 are for in-distribution test sets: e.g, if they are trained on SQUID, then the test sets are also from SQUID. The term "Synthetic" in Table 2 refers to different datasets for different tasks (see Appendix A.4 for details). From Table 2, we see that in almost all cases, "$\mathcal{S}_{\text{extent}}$ (Frozen)" achieves the best result (lowest). In general Sumformer versions outperform corresponding Transformer versions, which is suggestive of the benefit of the alignment (resemblance) of Sumformer-instantiated kernel-approx-NN with Algorithm 1. In the case of Sumformer versions of neural models,

Table 2: Error metrics across MEB, MEE, and MEA (standard deviations in Appendix). $\mathcal{E}_r$: relative radius error of MEB $\mathcal{E}_{r,\min}$ / $\mathcal{E}_{r,\mathrm{maj}}$: minor/major radius errors of MEEs $\mathcal{E}_w$: error in MEA width. Boldface marks the best (lowest) value in each column.

| Train Set | Method | MEB $\mathcal{E}_r$ | MEE $\mathcal{E}_{r,\min}$ | $\mathcal{E}_{r,\mathrm{maj}}$ | MEA $\mathcal{E}_w$ | Train Set | Method | MEB $\mathcal{E}_r$ | MEE $\mathcal{E}_{r,\min}$ | $\mathcal{E}_{r,\mathrm{maj}}$ | MEA $\mathcal{E}_w$ |
|---|---|---|---|---|---|---|---|---|---|---|---|
| Synthetic | $\mathcal{S}_{\text{extent}}$ (Frozen) | **0.009** | **0.037** | **0.022** | 0.050 | SQUID | $\mathcal{S}_{\text{extent}}$ (Frozen) | 0.017 | **0.056** | 0.047 | 0.073 |
| | $\mathcal{S}_{\text{extent}}$ (E2E) | 0.023 | 0.056 | 0.047 | **0.038** | | $\mathcal{S}_{\text{extent}}$ (E2E) | **0.010** | 0.078 | 0.050 | 0.087 |
| | $\mathcal{T}_{\text{extent}}$ (Frozen) | 0.099 | 0.041 | 0.027 | 0.083 | | $\mathcal{T}_{\text{extent}}$ (Frozen) | 0.068 | 0.058 | **0.032** | 0.064 |
| | $\mathcal{T}_{\text{extent}}$ (E2E) | 0.024 | 0.038 | 0.022 | 0.077 | | $\mathcal{T}_{\text{extent}}$ (E2E) | 0.190 | 0.093 | 0.045 | 0.104 |
| | $\mathcal{S}_{\text{Baseline}}$ | 0.048 | 0.378 | 0.426 | 0.674 | | $\mathcal{S}_{\text{Baseline}}$ | 0.027 | 0.322 | 0.250 | 0.191 |
| | $\mathcal{T}_{\text{Baseline}}$ | 0.030 | 0.071 | 0.047 | 0.417 | | $\mathcal{T}_{\text{Baseline}}$ | 0.039 | 0.357 | 0.049 | 0.118 |
| | $\mathcal{S}_{\text{Direct}}$ | 0.126 | 0.033 | 0.038 | 0.141 | | $\mathcal{S}_{\text{Direct}}$ | 0.050 | 0.68 | 0.05 | **0.041** |
| | $\mathcal{T}_{\text{Direct}}$ | 0.178 | 0.932 | 0.887 | 0.079 | | $\mathcal{T}_{\text{Direct}}$ | 0.685 | 1.269 | 0.188 | 0.065 |

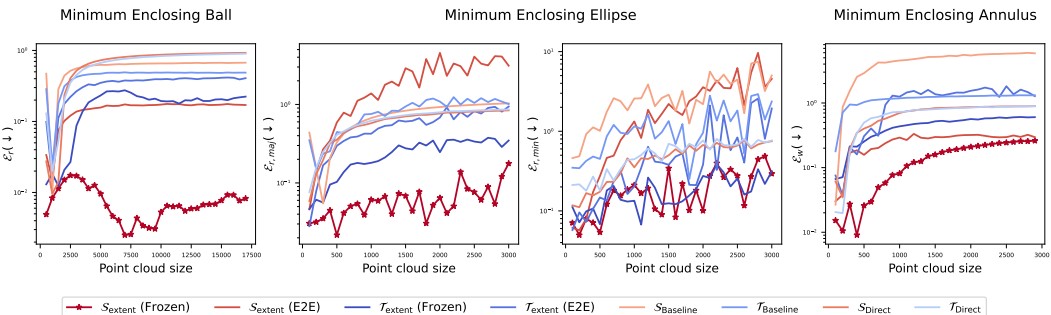

Figure 6: Size generalization across all extent measure tasks. We record the error for each model on extent measure tasks as the size and bounding box of the input point cloud increases. Our frozen $\mathcal{S}_{\text{extent}}$ model consistently outperforms all other neural baselines across every task.

the "Frozen" version (where the processor module is pre-trained then frozen, and only the encoder and decoder will be trained in a task driven manner) is usually much better than the "E2E" version. This advantage, however, is less clear for the Transformer based models. Finally, while the direct models outperform our models in terms of $\mathcal{E}_{\mathrm{w}}$ for MEA on the SQUID dataset, refer to Table 15 in Appendix A.4 to see that our models achieve more geometrically faithful predictions, reducing the proportion of excluded points by approximately one third.

Additionally, we examine how each model's error changes as the size of the input point cloud increases (i.e., its size generalization capability) across all extent-measure problems in Figure 6. Each model in this experiment is trained on synthetic data generated from within the bounding box $[-5, 5]^2$. For the test data, we simultaneously scale both the number of input points and the size of the bounding box from which each point cloud is sampled. Extended details for the experiment set-up are in Appendix A.4. The results in Figure 6 highlight the advantage of the Sumformer-instantiated kernel-approx-NN as the "$\mathcal{S}_{\text{extent}}$ (Frozen)" has much better generalization as compared to all other neural models. Visualizations of the solutions in Figure 14 (Appendix A.4) for in-distribution and out-of-distribution data for each problem even more suggestive of how aligning the processor network explicitly with relaxed-$\varepsilon$-kernel can better adjust to OOD data.

## 6 Conclusion and Limitations

We introduce an alternative theory of relaxed-$\varepsilon$-kernels which we show naturally aligns with a SumFormer-based architecture to approximate a family of faithful and unfaithful extent measure problems using bounded model complexity. Empirically, we demonstrate the benefits of *alignment* of our SumFormer instantiated models and the effectiveness of our (encoder-process-decoder) architecture, especially in better OOD generalization performance. Some limitations include: The model complexity still depends on the dimension (but not the size) of the input point sets. Also our theoretical results depend on the "fatness" of the input point sets. Nevertheless, our work shows the benefits of neural-algorithmic alignments and opens the door to further explorations of how the interplay between task structure, classical algorithmic approaches, and network design yields better performance, both theoretically and empirically.

# 7 Acknowledgements

This work is partially supported by NSF (National Science Foundation) by grants CCF-2112665 as well as a NAIRR Pilot Project NAIRR250087.

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

# A Technical Appendices and Supplementary Material

## A.1 Additional details from Section 2

Here, we provide more details regarding the computation of $\varepsilon$-kernels. One simple algorithm to compute an $\varepsilon$-kernel is given in Algorithm 2. More sophisticated methods can be found in [10] and [46] but we highlight Algorithm 2 as it is most similar to the computation of the relaxed-$\varepsilon$-kernel in Section 3.2. In summary, one may compute an $\varepsilon$-kernel by carefully choosing a set of directions $\Omega$ and choose points according to the max-projection in that direction. We show the correctness of this algorithm in Theorem A.3. However, we first provide some auxiliary lemmas which we will use in the proof of Theorem A.3 as well as subsequent proofs.

**Lemma A.1.** *If $P$ is an $\alpha$-fat point set, then for any $u \in \mathbf{S}^{d-1}$, $\mathsf{w}_u(P) \geq 2\alpha$.*

*Proof.* Suppose $P$ is an $\alpha$-fat point set so $\alpha\mathbb{C} \subseteq \mathrm{CH}(P)$. For any $u \in \mathbf{S}^{d-1}$ and $p \in \mathrm{CH}(P)$, $\langle u, p \rangle \geq \|u\|\|p\|$. Since $\alpha\mathbb{C} \subseteq \mathrm{CH}(P)$, $\|p\| \geq \alpha$ so $\langle u, p \rangle \geq \alpha$ and $\max_{p \in P}\langle u, p \rangle \geq \alpha$. Furthermore, $\min_{p \in P}\langle u, p \rangle = -\max_{p \in P} -\langle u, p \rangle = -\max_{p \in P}\langle -u, p \rangle \leq -\alpha$. Then

$$\mathsf{w}_u(P) = \max_{p \in P}\langle u, p \rangle - \min_{p \in P}\langle u, p \rangle \geq 2\alpha$$

$\square$

**Lemma A.2.** *Let $S \subseteq \mathbb{C}_d$. Given two unit vectors $u, u' \in$ such that $\|u - u'\| \leq \lambda$, $\mathsf{w}_u(S) \geq \mathsf{w}_{u'}(S) - 2\lambda\sqrt{d}$.*

*Proof.* Given any $x \in S$,

$$|\langle x, u \rangle - \langle x, u' \rangle| = |\langle u - u', x \rangle| \leq \|\|u - u'\|\|x\|\| \leq \lambda\sqrt{d}.$$

where the final $\sqrt{d}$ is because $x \in [-1, 1]^d$. Then

$$
\begin{aligned}
\mathsf{w}_{u'}(S) &= \max_{s \in S}\langle u', s \rangle - \min_{s, \in S}\langle u', s \rangle \\
&\geq \langle u', \mathrm{argmax}_{s \in S}\langle u, s \rangle \rangle - \min_{s \in S}\langle u', s \rangle \\
&\geq \langle u, \mathrm{argmax}_{s \in S}\langle u, s \rangle \rangle - \lambda\sqrt{d} - \min_{s \in S}\langle u', s \rangle \\
&\geq \langle u, \mathrm{argmax}_{s \in S}\langle u, s \rangle \rangle - \lambda\sqrt{d} - \langle u', \mathrm{argmin}_{s \in S}\langle u, s \rangle \rangle \\
&\geq \langle u, \mathrm{argmax}_{s \in S}\langle u, s \rangle \rangle - \lambda\sqrt{d} - (\langle u, \mathrm{argmin}_{s \in S}\langle u, s \rangle \rangle + \lambda\sqrt{d}) \\
&\geq \mathsf{w}_u(S) - 2\lambda\sqrt{d}.
\end{aligned}
$$

$\square$

**Theorem A.3.** *Let $P \subseteq \mathbb{C} = [-1, 1]^d$ be an $\alpha$-fat point set and let $0 < \varepsilon < \frac{1}{3}$. Suppose $\Omega$ is an $\frac{\alpha\varepsilon}{4\sqrt{d}}$-net for $\mathbf{S}^{d-1}$ so $|\Omega| = O\left(\frac{1}{(\frac{\alpha\varepsilon}{4\sqrt{d}})^{d-1}}\right)$, $\forall u, u' \in \Omega$, $\|u - u'\| \leq \frac{1}{(\frac{\alpha\varepsilon}{4\sqrt{d}})^{d-1}}$, and for all $u \in \Omega$, $-u \in \Omega$. Suppose that $\forall u \in \Omega$, $-u \in \Omega$. Then the set of points $Q$ constructed via Algorithm 2 is an $\varepsilon$-kernel for $P$.*

---

**Algorithm 2** Computation of $\epsilon$-kernels

---

**Require:** $P \subseteq \mathbb{R}^d$, $k$
    $\Omega$ is a set of $k$ directions
    Initialize $Q = \{\}$
    **for** $u \in \Omega$ **do**
        $Q$.append($\mathrm{argmax}_{p \in P}\langle u, p \rangle$)
    **end for**
    **return** $Q = \{q_u : u \in \Omega\}$

---

**Proof of Theorem A.3.** Let $u \in \mathbf{S}^{d-1}$. Let $\lambda = \frac{\alpha \varepsilon}{4\sqrt{d}}$. Since $\Omega$ is an $\lambda$-net for $\mathbf{S}^{d-1}$, there is a $u' \in \Omega$ such that $\|u - u'\| \le \lambda$. Therefore, by Lemma A.2,

$$\mathsf{w}_u(Q) \ge \mathsf{w}_{u'}(Q) - 2\lambda\sqrt{d} \ge \mathsf{w}_{u'}(P) - 2\lambda\sqrt{d} \ge \mathsf{w}_u(P) - 4\lambda\sqrt{d}.$$

Therefore,

$$\frac{\mathsf{w}_u(Q)}{\mathsf{w}_u(P)} \ge 1 - \frac{4\lambda\sqrt{d}}{\mathsf{w}_u(P)} \ge 1 - \frac{4\lambda\sqrt{d}}{\alpha} = 1 - \varepsilon$$

where the second inequality comes from Lemma A.1. Thus, $\mathsf{w}_u(Q) \ge (1 - \varepsilon)\mathsf{w}_u(P)$.

## A.2 Additional details from Section 4

We provide an example of how relaxed-$\varepsilon$-kernels can be used to approximate minimum width enclosing annulus. However, we will first need show in Theorem A.5 how Theorem 4.4 can be used to compute relaxed-$\varepsilon$-kernels for fractional powers of polynomials. Our proof of Theorem A.5 builds on the following lemma, originally proved by [2].

**Lemma A.4.** *Let $0 < \varepsilon < 1$ be a parameter, $r \ge 2$ and let $\delta = (\varepsilon/2(r-1))^r$. If we have $0 \le a \le A \le B \le b$ and $B - A \ge (1 - \delta)(b - a)$, then*

$$B^{1/r} - A^{1/r} \ge (1 - \varepsilon)(b^{1/r} - a^{1/r})$$

**Theorem A.5.** *Let $\mathcal{F}$ be a family of $(d + p)$ variate polynomials which admit a linearization of dimension $m$ as in Theorem 4.4. Additionally, suppose for every $f_i$, $f_i(x) \ge 0$ for all $x \in \mathbb{R}^d$. Let $r \ge 2$, $\delta = (\varepsilon/2(r-1))^r$, and let $\mathcal{G} = \{g_i\}$ be a relaxed-$\delta$-kernel of $\mathcal{F}$. Then $\mathcal{G}_{1/\mathrm{r}} = \{g_i^{1/r}\}$ is a relaxed-$\varepsilon$-kernel of $\mathcal{F}_{1/\mathrm{r}} = \{f_i^{1/\mathrm{r}}\}$.*

*Proof.* Let $\delta = (\varepsilon/2(r-1))^r$ where $r \ge 2$ is an integer. Let $\mathcal{G}$ be a relaxed-$\delta$-kernel of $\mathcal{F}$ so $\overrightarrow{\mathcal{E}}_{\mathcal{G}}(x) \subseteq \overrightarrow{\mathcal{E}}_{\mathcal{F}}(x)$ and $\overrightarrow{\mathcal{E}}_{\mathcal{G}}(x)$ is within $\varepsilon \cdot \mathcal{E}_{\mathcal{F}}(x)$-Hausdorff distance of $\overrightarrow{\mathcal{E}}_{\mathcal{F}}(x)$. Since $f \in \mathcal{F}$ is positive, for any $x \in \mathbb{R}^d$, we know that

$$0 \le \min_{f \in \mathcal{F}}(x) \le \min_{g \in \mathcal{G}}(x) \le \max_{g \in \mathcal{G}}(x) \le \max_{f \in \mathcal{F}}(x).$$

Additionally, by the definition of relaxed-$\varepsilon$-kernels, $\max_{g \in \mathcal{G}} g(x) - \min_{g \in \mathcal{G}} g(x) \ge (1 - \varepsilon)(\max_{f \in \mathcal{F}}(x) - \min_{f \in \mathcal{F}}(x))$. Then, we apply Lemma A.4 to get

$$\max_{g \in \mathcal{G}_{1/r}} g(x)^{1/r} - \min g \in \mathcal{G}_{1/r} g(x)^{1/r} \ge (1 - \varepsilon)(\max_{g \in \mathcal{F}_{1/r}} f(x)^{1/r} - \min_{f \in \mathcal{F}_{1/r}} f(x)^{1/r}).$$

$\square$

Now, we are prepared to provide an example of Theorem A.5 can be used to approximate minimum enclosing annulus for points in $\mathbb{R}^2$. Given $P = \{p_1, \ldots, p_N\} \subseteq \mathbb{R}^2$ and any $x \in \mathbb{R}^2$, finding the width of the a minimum enclosing spherical annulus centered at $x$ is

$$w(x, P) = \max_{p \in P} \|x - p\| - \min_{p \in P} \|x - p\|.$$

We define the set of functions $\mathcal{F} = \{f_p(x) = \|x - p\| : p \in P\}$. Notice then that $\mathcal{E}_{\mathcal{F}}(x) = w(x, P)$ and the width of the minimum enclosing spherical shell is exactly $\min_{x \in \mathbb{R}^d} \mathcal{E}_{\mathcal{F}}(x)$. Similarly, the optimal center for the minimum enclosing spherical shell is $\mathrm{argmin}_{x \in \mathbb{R}^d} \mathcal{E}_{\mathcal{F}}(x)$. By Theorem A.5, a relaxed-$\varepsilon$-kernel for $\mathcal{F}' = \{\|x - p\|^2 : p \in P\}$ translates to relaxed-$\varepsilon$-kernel for $\mathcal{F}$.

Suppose $x = (x_1, x_2) \in \mathbb{R}^2$ and $p_i = (p_{i,1}, p_{i,2}) \in \mathbb{R}^2$ where $p_i \in P$. Given $f_i \in \mathcal{F}'$, $f_i = x_1^2 + x_2^2 - 2p_{i,1}x_1 - 2p_{i,2}x_2 + p_{i,1}^2 + p_{i,2}^2$. $f_i$ admits a linearization of dimension 3 as follows:

$$\psi_0(p_i) = p_{i,1}^2 + p_{i,2}^2 \quad \psi_1(p_i) = -2p_{i,1} \quad \psi_2(p_i) = -2p_{i,2} \quad \psi_3(p_i) = 1$$
$$\varphi_1(x) = x_1 \quad \varphi_2(x) = x_2 \quad \varphi_3(x) = x_1^2 + x_2^2$$

By Theorem 4.4, we can compute a relaxed-$\varepsilon$-kernel for the dual space via Algorithm 1 in $\mathbb{R}^4$ and map back to relaxed-$\varepsilon$-kernel for $\mathcal{F}'$, $\mathcal{Q}_\varepsilon$. Then $\mathcal{Q}_{\varepsilon,1/r}$ is a relaxed-$\varepsilon$-kernel for $\mathcal{F}$ by Theorem A.5. Computing $\min_{x \in \mathbb{R}^2} \mathcal{E}_{\mathcal{Q}_{\varepsilon,1/r}}(x)$ as well as $\mathrm{argmin}_{x \in \mathbb{R}^2} \mathcal{E}_{\mathcal{Q}_{\varepsilon,1/r}}(x)$ outputs the width and center of the minimum enclosing annulus, respectively.

## A.3 Proofs details

In this section, we will provide the missing proofs from the main text.

### A.3.1 Proofs from Section 3

The following lemma will be important to show that relaxed $\varepsilon$-kernels can approximate well-behaved measurements of the original point set $P$.

**Lemma A.6.** *Suppose $Q$ is a relaxed-$\varepsilon$-kernel of $P$. Let $\hat{P} = P \cup Q$. Then (a) $Q$ is a $\varepsilon$-approximation of $\hat{P}$ and (b) $P$ is an $\varepsilon$-approximation of $\hat{P}$.*

**Proof of Lemma A.6**  Let $u \in \mathbf{S}^{d-1}$. To show that $Q$ is an $\varepsilon$-approximation of $\hat{P}$, we must show that

$$\max_{q \in Q}\langle q, u \rangle - \varepsilon \mathsf{w}_u(Q) \leq \max_{\hat{p} \in \hat{P}}\langle \hat{p}, u \rangle \leq \max_{q \in Q}\langle q, u \rangle + \varepsilon \mathsf{w}_u(Q)$$

and

$$\min_{q \in Q}\langle q, u \rangle - \varepsilon \mathsf{w}_u(Q) \leq \min_{\hat{p} \in \hat{P}}\langle \hat{p}, u \rangle \leq \min_{q \in Q}\langle q, u \rangle + \varepsilon \mathsf{w}_u(Q)$$

We know that $\max_{\hat{p} \in \hat{P}}\langle \hat{p}, u \rangle = \max\{\max_{p \in P}\langle p, u \rangle, \max_{q \in Q}\langle q, u \rangle\}$. We know that $\max_{q \in Q}\langle q, u \rangle - \varepsilon \mathsf{w}_u(Q) \leq \max_{p \in P}\langle u, p \rangle \leq \max_{q \in Q}\langle u, q \rangle + \varepsilon \mathsf{w}_u(Q)$ because $Q$ is an $\varepsilon$-relaxed-kernel of $P$. Clearly, $\max_{q \in Q}\langle q, u \rangle - \varepsilon \mathsf{w}_u(Q) \leq \max_{q \in Q}\langle q, u \rangle \leq \max_{q \in Q}\langle q, u \rangle + \varepsilon \mathsf{w}_u(Q)$ so

$$\max_{q \in Q}\langle q, u \rangle - \varepsilon \mathsf{w}_u(Q) \leq \max\{\max_{p \in P}\langle p, u \rangle, \max_{q \in Q}\langle q, u \rangle\} = \max_{\hat{p} \in \hat{P}}\langle \hat{p}, u \rangle \leq \max_{q \in Q}\langle q, u \rangle + \varepsilon \mathsf{w}_u(Q).$$

We can do something similar for $\min_{q \in Q}\langle q, u \rangle$ and $\min_{\hat{p} \in \hat{P}}\langle \hat{p}, u \rangle$.

The same argument will hold for (b).

**Proof of Theorem 3.2.**  Let $\hat{P} = P \cup Q$. Let $\mu$ be a faithful measure and let $c$ be the constant in for the faithful measure, $\mu$. By Lemma A.6 (a), $Q$ is $\varepsilon$-kernel of $\hat{P}$, meaning $Q$ is a $c\varepsilon$-coreset of $\hat{P}$. By Lemma A.6 (b), $P$ is $\varepsilon$-kernel of $\hat{P}$, meaning $P$ is a $c\varepsilon$-coreset of $\hat{P}$. It then follows that

$$\frac{1 - c\varepsilon}{1 + c\varepsilon}\mu(Q) \leq \mu(P) \leq \frac{1 + c\varepsilon}{1 - c\varepsilon}\mu(Q).$$

For $c\varepsilon < 1/3$, it is easy to verify that this implies

$$(1 - 3c\varepsilon)\mu(Q) \leq \mu(P) \leq (1 + 3c\varepsilon)\mu(Q),$$

hence the theorem holds with $c' = 3c$.

**Proof of Theorem 3.3**  First, we make two important observations regarding the points in the output relaxed-$\varepsilon$-kernels, $Q$, computed by Algorithm 1 and their relationship to the set of chosen directions $\Omega$. First, suppose we are given $\varepsilon > 0$ and $A = \{a_1, \ldots, a_N\} \subseteq \mathbb{R}$ such that $a_i = \langle u, p_i \rangle$, let $\mathbf{a} = [a_1, \ldots, a_N]^T$. Recall from Algorithm 1, $\rho_\varepsilon$ is defined as

$$\rho_\varepsilon(\mathbf{a}) = \begin{pmatrix} \mathrm{ReLU}(a_1 + \varepsilon \cdot w(A) - M_A) \\ \mathrm{ReLU}(a_2 + \varepsilon \cdot w(A) - M_A) \\ \vdots \\ \mathrm{ReLU}(a_n + \varepsilon \cdot w(A) - M_A) \end{pmatrix}$$

where $M_A = \max(A)$ and $w(A) = \max(A) - \min(A)$. Note $\mathrm{ReLU}(a_i + \varepsilon \cdot w(A) - M_A) > 0$ only when $a_i$ is within $\varepsilon \cdot w(A)$ of the maximum value, $M_A$. Recall our modified softmax normalization from Algorithm 1 given $x \in \mathbb{R}^n$:

$$\sigma(x) = \left[ \frac{e^{x_1} - 1}{\sum_{i=1}^n (e^{x_i} - 1)} \cdots \frac{e^{x_n} - 1}{\sum_{i=1}^n (e^{x_i} - 1)} \right].$$

Notice that $\sigma(x)_i > 0$ only when $x_i > 0$. Hence, we can see $\sigma(\rho_\varepsilon(\mathbf{a}))$ as a probability vector over $\mathbf{a}$ which only has non-zero values corresponding to those $a_i$ that are within $\varepsilon \cdot w(A)$ of $M_A$. If we take $\sigma(\rho_\varepsilon(\mathbf{a}))^T\mathbf{a}$, we observe that $\sigma(\rho_\varepsilon(\mathbf{a}))^T\mathbf{a}$ is a convex combination of the points in $A$.

Thus, given $u \in \Omega$ and $q_u \in Q$, we make the following important observations about $q_u$.

**Observation A.7.** *Given $u \in \Omega$, $q_u \in Q$ is the convex combination of a subset of points $p_{I_1}, \ldots, p_{I_{r_u}}$. For each such point $p_{I_i}$, $\langle u, p_{I_i} \rangle \in [\max_{p \in P} \langle u, p \rangle - \varepsilon \mathsf{w}_u(P), \max_{p \in P} \langle u, p \rangle]$.*

**Observation A.8.** *Given $u \in \Omega$, $\langle u, q_u \rangle \in [\max_{p \in P} \langle u, p \rangle - \varepsilon \mathsf{w}_u(P), \max_{p \in P} \langle u, p \rangle]$.*

Our goal is to show that for any $u \in \mathbf{S}^{d-1}$, the $u$-projection $\overrightarrow{\mathsf{w}}_u(Q)$ is an $\varepsilon$-approximation of the interval ($u$-projection) $\overrightarrow{\mathsf{w}}_u(P)$. First, note that by construction, each point in $Q$ is a convex combination of a subset of points in $P$ so $Q \subseteq \mathrm{CH}(P)$. This implies that $\overrightarrow{\mathsf{w}}_u(Q) \subseteq \overrightarrow{\mathsf{w}}_u(P)$. What remains is to show that $\mathsf{w}_u(Q) \geq (1 - 3\varepsilon)\mathsf{w}_u(P)$. If $u \in \Omega$, then this holds as

$$
\begin{aligned}
\max_{q \in Q}\langle u, q \rangle - \min_{q \in Q}\langle u, q \rangle &\geq \langle u, q_u \rangle - \langle -u, q_{-u} \rangle \\
&\geq \max_{p \in P}\langle u, p \rangle - \varepsilon \mathsf{w}_u(P) - \min_{p \in P}\langle u, p \rangle - \varepsilon \mathsf{w}_u(P) \quad \text{due to Observation 2} \\
&\geq \mathsf{w}_u(P) - 2\varepsilon \mathsf{w}_u(P) \\
&\geq (1 - 3\varepsilon)\mathsf{w}_u(P)
\end{aligned}
$$

Consider when $u \notin \Omega$ and let $\lambda = \frac{\alpha \varepsilon}{4\sqrt{d}}$. Then there is a $u' \in \Omega$ such that $\|u - u'\| \leq \lambda$ due to the construction of $\Omega$ and $-u' \in \Omega$. Then

$$
\begin{aligned}
\mathsf{w}_{u'}(Q) &= \max_{q \in Q}\langle u', q \rangle - \min_{q \in Q}\langle u', q \rangle \\
&\geq \langle u', q_{u'} \rangle - \min_{q \in Q}\langle u', q \rangle \\
&\geq \langle u', \mathrm{argmax}_{p \in P}\langle u', p \rangle \rangle - \varepsilon \mathsf{w}_{u'}(P) - \min_{q \in Q}\langle u', q \rangle \quad \text{(Observation 2)} \\
&= \langle u', \mathrm{argmax}_{p \in P}\langle u', p \rangle \rangle - \varepsilon \mathsf{w}_{u'}(P) - \max_{q \in Q}\langle -u', q \rangle \\
&= \max_{p \in P}\langle u', p \rangle - \varepsilon \mathsf{w}_{u'}(P) - \langle -u', q_{-u'} \rangle \\
&\geq \max_{p \in P}\langle u', p \rangle - \varepsilon \mathsf{w}_{u'}(P) - (\langle -u', \mathrm{argmax}_{p \in P}\langle -u', p \rangle \rangle - \varepsilon \mathsf{w}_{-u'}(P)) \\
&= \max_{p \in P}\langle u', p \rangle - \min_{p \in P}\langle u', p \rangle - 2\varepsilon \mathsf{w}_{u'}(P) = (1 - 2\varepsilon)\mathsf{w}_{u'}(P)
\end{aligned}
$$

By Lemma A.2,

$$
\begin{aligned}
\mathsf{w}_u(Q) &\geq \mathsf{w}_{u'}(Q) - 2\lambda\sqrt{d} \\
&\geq (1 - 2\varepsilon)\mathsf{w}_{u'}(P) \\
&\geq (1 - 2\varepsilon)(\mathsf{w}_u(P) - 2\lambda\sqrt{d}) - 2\lambda\sqrt{d} \\
&= (1 - 2\varepsilon)\mathsf{w}_u(P) - 2\lambda\sqrt{d}(1 - 2\varepsilon + 1) \\
&= (1 - 2\varepsilon)\mathsf{w}_u(P) - 4\lambda\sqrt{d}(1 - \varepsilon)
\end{aligned}
$$

Therefore, by Lemma A.1,

$$
\begin{aligned}
\frac{\mathsf{w}_u(Q)}{\mathsf{w}_u(P)} &\geq 1 - 2\varepsilon - \frac{4\lambda\sqrt{d}(1 - \varepsilon)}{\mathsf{w}_u(P)} \\
&\geq 1 - 2\varepsilon - \frac{4\lambda\sqrt{d}(1 - \varepsilon)}{\alpha} \quad \text{due to } \alpha\text{-fatness of } P \\
&\geq 1 - 2\varepsilon - \varepsilon(1 - \varepsilon) \\
&\geq 1 - 3\varepsilon
\end{aligned}
$$

and we get our desired result i.e. $\mathsf{w}_u(Q) \geq (1 - 3\varepsilon)\mathsf{w}_u(P)$

**Proof of Theorem 3.4.** Let $k(\varepsilon) \geq O\left(\frac{1}{(\frac{\alpha \varepsilon / 3}{4\sqrt{d}})^{d-1}}\right)$ and let $\Omega$ be a fixed set of directions such that $|\Omega| = k(\varepsilon)$. Notice that here, since $k(\varepsilon) \geq O\left(\frac{1}{(\frac{\alpha \varepsilon / 3}{4\sqrt{d}})^{d-1}}\right)$, by Theorem 3.3, the output of

Algorithm 1 would be a relaxed-$\varepsilon$-kernel. Let $F_{\text{inner}} : \mathcal{X} \to \mathbb{R}^{k(\varepsilon)}$ such that

$$F_{\text{inner}}(S) = \begin{pmatrix} \max_{x \in S} \langle u_1, x \rangle \\ \vdots \\ \max_{s \in S} \langle u_{k(\varepsilon)}, x \rangle \end{pmatrix}$$

Additionally, let $F_{\text{outer},1} : \mathbb{R}^d \times \mathbb{R}^d \to \mathbb{R}^{k(\varepsilon)}$ such

$$F_{\text{outer},1}(x) = \begin{pmatrix} \langle x, u_1 \rangle \\ \langle x, u_2 \rangle \\ \vdots \\ \langle x, u_{k(\varepsilon)} \rangle \end{pmatrix}$$

and $F_{\text{outer},2} : \mathbb{R}^{k(\varepsilon)} \times \mathbb{R}^{k(\varepsilon)} \to \mathbb{R}^{k(\varepsilon)}$ be defined as $F_{\text{outer},2}(x, y) = \text{ReLU}(x + y)$. Given $x \in \mathbb{R}^d$ and $y \in \mathbb{R}^{k(\varepsilon)}$, let $F_{\text{outer}}(x, y) = F_{\text{outer},2}(F_{\text{outer},1}(x) + y)$.

First, we note that $\ell_p$-norms converge to the $\ell_\infty$-norm so there is a $p$ such that for any $x \in \mathbb{R}^N$, $|\|x\|_p - \|x\|_\infty| < \frac{\varepsilon\alpha}{2}$. $F_{\text{inner}}$ can be approximated to within $\frac{\varepsilon\alpha}{2}$ via a sum-decomposition by first mapping each $s \in S$ to $\phi_1(s) = [\langle u_1, s \rangle^p, \ldots, \langle u_{k(\varepsilon)}, s \rangle^p]^T \in \mathbb{R}^{k(\varepsilon)}$ and then taking $\sum(\phi_1(s))^{1/p}$ where the $1/p$-power is taken elementwise. Thus, by the universal approximation property of MLPs, there are multilayer perceptrons $\text{MLP}_1$ and $\text{MLP}_2$ such that

$$\left\| F_{\text{inner}}(S) - \text{MLP}_1 \left( \sum_{s \in S} \text{MLP}_2(s) \right) \right\| < \varepsilon\alpha.$$

This means that $F_{\text{inner}}(S)_i$ is within $\varepsilon\alpha$ for $\max_{s \in S} \langle u_i, s \rangle$ for any $S \in \mathcal{X}_\alpha$. Because each set in $\mathcal{X}_\alpha$ is $\alpha$-fat, $\forall u \in S^{d-1}$, $F_{\text{inner}}(S)_i$ is within $\varepsilon\mathsf{w}_u(S)$ of the maximum value. In other words,

$$\left| \text{MLP}_1 \left( \sum_{s \in S} \text{MLP}_2(s) \right)_i - F_{\text{inner}}(S)_i \right| \le \varepsilon \cdot \alpha \le \varepsilon \cdot \mathsf{w}_{u_i}(S).$$

We can implement $F_{\text{outer},1}$ with $\text{MLP}_3$ i.e. $F_{\text{outer},1}(x) = \text{MLP}_3(x)$ where $\text{MLP}_3$ can realize $F_{\text{outer},1}$ by choosing its weight matrix to be diagonal with entries $[u_1, \ldots, u_{k(\varepsilon)}]$ along the diagonal. Similarly, $F_{\text{outer},2}$ can be exactly represented by

$$F_{\text{outer},2}(x, y) = \text{MLP}_4(x + y)$$

where $\text{MLP}_4$ represents the identity function followed by a ReLU activation. $F_{\text{outer}}$ can be represented by a neural network $\psi$ such that $\psi(x, y) = \text{MLP}_4(\text{MLP}_3(x) + y)$. Given $P \in \mathcal{X}_\alpha$ and $p_i \in P$,

$$\psi \left( p_i + \text{MLP}_1 \left( \sum_{s \in S} \text{MLP}_2(s) \right) \right) = \begin{pmatrix} \text{ReLU}(\langle u_1, p_i \rangle + \varepsilon\mathsf{w}_{u_i}(P) - \max_{p \in P} \langle u_1, p \rangle) \\ \vdots \\ \text{ReLU}(\langle u_{k(\varepsilon)}, p_i \rangle + \varepsilon\mathsf{w}_{u_i}(P) - \max_{p \in P} \langle u_{k(\varepsilon)}, p \rangle) \end{pmatrix}$$

This is exactly the point-wise update computed in a SumFormer architecture so if we instantiate a SumFormer $\mathcal{S}_\varepsilon$ with the MLPs described above and each MLP in $\mathcal{S}_\varepsilon$ mapping to an intermediate dimension of $k(\varepsilon)$, $Q_{\mathcal{S}_\varepsilon} = \sigma_{\text{col}}(\mathcal{S}_\varepsilon(P))^T P$ is a relaxed-$\varepsilon$-kernel by Theorem 3.3. Additionally, note that each MLP used to approximate $Q_{\mathcal{S}_\varepsilon}$ maps to $\mathbb{R}^k(\varepsilon)$ where $k$ is *independent of the size of the input set and only dependent on the input dimension d, the fatness of the point set $\alpha$, and the desired approximation error $\varepsilon$.*

**Proof of Corollary 3.5**   From Theorem 3.4, we know that $Q_\varepsilon = L_{1,\text{col}}(\mathcal{S}_\varepsilon(P))^T P$ is a relaxed $\varepsilon$-kernel for $P$. By Theorem 3.2, since $Q_\varepsilon$ is a relaxed $\varepsilon$-kernel, it is $3c\varepsilon$-coreset for $P$. Then $(1 - 3c\varepsilon)\mu(P) \le \mu(Q_\varepsilon) \le (1 + 3c\varepsilon)\mu(P)$. By the universality of DeepSets, $\mathcal{N}_{\text{deepset}}$ can approximate $\mu$ to arbitrary accuracy for any point set in $\mathcal{X}_\alpha$. Thus, there is a $\mathcal{N}_{\text{deepset}}$ architecture such that

$$|\mathcal{N}_{\text{deepset}}(Q_\varepsilon) - \mu(Q_\varepsilon)| < \varepsilon$$

and we get the desired result.

### A.3.2 Proofs from Section 4

**Proof of Theorem 4.3**  Note that the non-relaxed version is also given in [2]. For completeness, we prove this theorem for the relaxed case which essentially just follows from the definitions of duality.

First for any point $x \in \mathbb{R}^{d-1}$, let $\hat{x} = [x, 1]$ denote the point in $\mathbb{R}^d$ with the last coordinate being 1. Let $\phi(x) := \hat{x}/\|\hat{x}\| \in \mathbf{S}^{d-1}$ be the unit vector in direction $x$. Let $\mathbf{S}_+^{d-1}$ be the positive semi-sphere. It is easy to see that $\phi$ is bijective from $\mathbb{R}^{d-1}$ to $\mathbf{S}_+^{d-1}$. Note that for any $x \in \mathbb{R}^{d-1}$, $\langle \hat{x}, f^* \rangle$ and for any set of functions $\mathcal{H}$,

$$L_\mathcal{H}(x) = \min_{h \in \mathcal{H}} h(x) = \|x\| \min_{h \in \mathcal{H}} \langle \phi(x), h^* \rangle \quad \text{and}$$

$$U_\mathcal{H}(x) = \max_{h \in \mathcal{H}} h(x) = \|x\| \max_{h \in \mathcal{H}} \langle \phi(x), h^* \rangle.$$

Then

$$\mathcal{E}_\mathcal{H}(x) = \|x\| \cdot \mathsf{w}_{\phi(x)}(\mathcal{H}^*)$$

where $\mathcal{H}^*$ is the dual point set of $\mathcal{H}$.

If $\mathcal{G} = \{g_1, \ldots, g_k\}$ is a relaxed $\varepsilon$-kernel for $\mathcal{F}$. Then for any $x \in \mathbb{R}^d$, we have that $L_\mathcal{F}(x) \geq L_\mathcal{G}(x) - \varepsilon \mathcal{E}_\mathcal{F}(x)$ and $U_\mathcal{F}(x) \leq U_\mathcal{G}(x) + \varepsilon \mathcal{E}_\mathcal{F}(x)$. It then follows that for any $x \in \mathbb{R}^{d-1}$,

$$\min_{f^* \in \mathcal{F}^*} \langle \phi(x), f^* \rangle \geq \min_{g^* \in \mathcal{G}^*} \langle \phi(x), g^* \rangle - \varepsilon \mathsf{w}_{\phi(x)}(\mathcal{F}^*) \quad \text{and}$$

$$\max_{f^* \in \mathcal{F}^*} \langle \phi(x), f^* \rangle \leq \max_{g^* \in \mathcal{G}^*} \langle \phi(x), g^* \rangle + \varepsilon \mathsf{w}_{\phi(x)}(\mathcal{F}^*)$$

That is, for any $u \in \mathbf{S}_+^{d-1}$, $\overrightarrow{\mathsf{w}}_u(\mathcal{G}^*)$ $\varepsilon$-approximates $\overrightarrow{\mathsf{w}}_u(\mathcal{F}^*)$. Given that $\langle -u, p \rangle = -\langle u, p \rangle$ for any point $p$, this means that $\overrightarrow{\mathsf{w}}_u(\mathcal{G}^*)$ $\varepsilon$-approximates $\overrightarrow{\mathsf{w}}_u(\mathcal{F}^*)$ holds for any $u \in \mathbf{S}_+^{d-1} \cup \mathbf{S}_-^{d-1}$. Hence the only directions left are $u \in \mathbf{S}^{d-1} \cap \{x \in \mathbb{R}^d, x_d = 0\}$. However, given that each term above is continuous, this holds due to continuity.

The other direction follows easily from the fact that $\phi$ being bijective.

**Proof of Theorem 4.4**  Suppose $\mathcal{F}$ admits a linearization of dimension $m$. For any $x \in \mathbb{R}^d$, we will show first that $\overrightarrow{\mathcal{E}}_{\mathcal{Q}_\varepsilon}(x) \subseteq \overrightarrow{\mathcal{E}}_\mathcal{F}(x)$. Then we will show that $\mathcal{E}_{\mathcal{Q}_\varepsilon}(x) \geq (1 - \varepsilon)\mathcal{E}_\mathcal{F}(x)$. If we use Algorithm 1 to compute a relaxed-$\varepsilon$-kernel of the dual $\mathcal{F}^*$, we know that each $q_i^* \in \mathcal{Q}_\varepsilon^*$ is

$$q_i^* = \alpha_1^i \Psi(a_1) + \cdots + \alpha_N^i \Psi(a_N)$$

where $\sum_{\ell=1}^N \alpha_\ell^i = 1$. Additionally, we will let $k = |\mathcal{Q}_\varepsilon^*|$. Then we know there is a $q_i \in \mathcal{Q}_\varepsilon$ where

$$q_i(x) = (\alpha_1^i \ldots \alpha_N^i) \begin{pmatrix} \psi_0(a_1) \\ \vdots \\ \psi_0(a_N) \end{pmatrix} + (\alpha_1^i \ldots \alpha_N^i) \begin{pmatrix} \psi_1(a_1) \\ \vdots \\ \psi_1(a_N) \end{pmatrix} \varphi_1(x) + \cdots + (\alpha_1^i \ldots \alpha_N^i) \begin{pmatrix} \psi_m(a_1) \\ \vdots \\ \psi_m(a_N) \end{pmatrix} \varphi_m(x)$$

To ease notation, for any $j \in \{0, \ldots, m\}$ and $i \in [k]$, we write

$$\tilde{\Psi}_j = \begin{pmatrix} \psi_j(a_1) \\ \vdots \\ \psi_j(a_N) \end{pmatrix} \in \mathbb{R}^N \quad \boldsymbol{\alpha}^i = \begin{pmatrix} \alpha_1^i \\ \vdots \\ \alpha_N^i \end{pmatrix} \in \mathbb{R}^N$$

so $q_i(x) = (\boldsymbol{\alpha}^i)^T \tilde{\Psi}_0 + (\boldsymbol{\alpha}^i)^T \tilde{\Psi}_1 \varphi_1(x) + \cdots + (\boldsymbol{\alpha}^i)^T \tilde{\Psi}_m \varphi_m(x)$. We can re-write $q_i$ as a convex combination of elements in $\mathcal{F}$:

$$q_i(x) = \alpha_1^i \Psi(a_1)^T \Phi(x) + \cdots + \alpha_N^i \Psi(a_N)^T \Phi(x)$$
$$= \alpha_1^i f_1(x) + \cdots + \alpha_N^i f_N(x)$$

Let $x \in \mathbb{R}^d$ and let $f^* = \operatorname{argmin}_{f \in \mathcal{F}} f(x)$. Then for any $q_i \in \mathcal{Q}$,

$$q_i(x) = \alpha_1^i f_1(x) + \cdots \alpha_N^i f_N(x) \geq \alpha_1^i f^*(x) + \cdots + \alpha_N^i f^*(x) = f^*(x)$$

Therefore, $L_{\mathcal{Q}_\varepsilon} = \min_{q \in \mathcal{Q}_\varepsilon} q(x) \geq L_\mathcal{F}(x) = \min_{f \in \mathcal{F}} f(x)$. A similar argument shows that $U_{\mathcal{Q}_\varepsilon}(x) \leq U_\mathcal{F}(x)$.

Table 3: **Dataset details for approximating relaxed-$\varepsilon$-kernels.** $|P_{\text{train}}|$ and $|P_{\text{test}}|$ refer to the size of the train and test point clouds, respectively. The first four columns describe synthetic datasets while the last two columns describe two real datasets. For the 'Uniform ball' dataset, we sample point clouds uniformly from a $d$-dimensional ball. Note that for the 'Uniform ball' and 'Uniform Ellipse' in $\mathbb{R}^{10}$, we use point clouds of size 700 to train and test. For 'Uniform Ball', we will sometimes call it 'Uniform Disk' for point clouds in $\mathbb{R}^2$. For 'Ellipse', we sample from a randomly scaled and rotated point cloud. For 'Single Gaussians', we sample point clouds from a Gaussian with a random standard deviation and 'Gaussian Mixture' refers to point clouds sampled from two to five randomly placed Gaussian clusters. Note that for ModelNet, we apply random scaling to the point cloud so that the bounds are between $[-5, 5]$. See [43] and [26] for more details regarding SQUID and ModelNet, respectively.

| | Synthetic | | | | Real | |
| --- | --- | --- | --- | --- | --- | --- |
| | Uniform Ball | Ellipse | Single Gaussian | Gaussian Mixture | ModelNet | SQUID |
| $|P_{\text{train}}|$ | 500, 700 (for $\mathbb{R}^{10}$) | 500, 700 (for $\mathbb{R}^{10}$) | 500 | 500 | 200 | 350 |
| $|P_{\text{test}}|$ | 500, 700 (for $\mathbb{R}^{10}$) | 500, 700 (for $\mathbb{R}^{10}$) | 500 | 500 | 200 | 350 |
| # train point clouds | 3000 | 3000 | 3000 | 3000 | 2048 | 774 |
| # test point clouds | 750 | 750 | 750 | 750 | 2048 | 332 |
| Input dim. | 2,3,5,10 | 2,3,5,10 | 2,3,5 | 2,3,5 | 3 | 2 |
| Bounds | $[-5, 5]$ | $[-5, 5]$ | $[-5, 5]$ | $[-5, 5]$ | $[-5, 5]$ | $[0, 450]$ |

Now we will show that $\mathcal{E}_{\mathcal{Q}_\varepsilon}(x) \geq (1 - \varepsilon)\mathcal{E}_{\mathcal{F}}(x)$ for any $x \in \mathbb{R}^d$. Given $x \in \mathbb{R}^d$, we know that $\Phi(x) = (1, \varphi_1(x), \ldots, \varphi_m(x)) \in \mathbb{R}^{m+1}$. Define $\mathcal{F}^{\text{Lin}}$ and $\mathcal{Q}_\varepsilon^{\text{Lin}}$ to be a set of linear functions

$$\mathcal{F}^{\text{Lin}} = \left\{ f_i^{\text{Lin}}(x_1, \ldots, x_m) = \Psi(a_j)^T \begin{pmatrix} 1 \\ x_1 \\ \vdots \\ x_m \end{pmatrix} : a_j \in A \right\}$$

$$\mathcal{Q}_\varepsilon^{\text{Lin}} = \left\{ ((\boldsymbol{\alpha}^i)^T \tilde{\Psi}_0 \cdots (\boldsymbol{\alpha}^i)^T \tilde{\Psi}_m) \begin{pmatrix} 1 \\ x_1 \\ \vdots \\ x_m \end{pmatrix} : i \in [m] \right\}.$$

First, notice that $(\mathcal{F}^{\text{Lin}})^* = \mathcal{F}^*$ and $(\mathcal{Q}_\varepsilon^{\text{Lin}})^* = \mathcal{Q}_\varepsilon^*$. We know that $\mathcal{Q}_\varepsilon^*$ is a relaxed-$\varepsilon$-kernel of $\mathcal{F}^*$ so by Theorem 4.3, $\mathcal{Q}_\varepsilon^{\text{Lin}}$ is a relaxed-$\varepsilon$-kernel for $\mathcal{F}^*$. Given any $x \in \mathbb{R}^d$, we know that $f_i(x) = f_i^{\text{Lin}}(\Phi(x))$ so

$$\mathcal{E}_{\mathcal{F}}(x) = \mathcal{E}_{\mathcal{F}^{\text{Lin}}}(\Phi(x)).$$

Similarly, we know that $q_j(x) = q_j^{\text{Lin}}(x)$ so

$$\mathcal{E}_{\mathcal{Q}_\varepsilon}(x) = \mathcal{E}_{\mathcal{Q}_\varepsilon^{\text{Lin}}}(\Phi(x)).$$

Since $\mathcal{Q}_\varepsilon^{\text{Lin}}$ is a relaxed-$\varepsilon$-kernel for $\mathcal{F}^*$,

$$\mathcal{E}_{\mathcal{Q}_\varepsilon}(x) \geq (1 - \varepsilon)\mathcal{E}_{\mathcal{F}}(x).$$

## A.4 Additional experimental details and results

All models are implemented in PyTorch and trained on 8 NVIDIA RTX A6000 GPUs. Our code is publically available[1]. Additionally, all models are trained using the ADAM optimizer [20] provided in PyTorch and trained using a learning rate of 0.001. Relaxed-$\varepsilon$-kernel networks are trained for 200 epochs while all extent-measure models are trained for 500 epochs. Additionally, for the processor for

---
[1] https://github.com/chens5/coreset-nn

Table 4: **Dataset details for extent measure approximation tasks.** $|P_{\text{train}}|$ and $|P_{\text{test}}|$ refer to the size of the train and test point clouds, respectively. The 'Uniform Ball' dataset is used for the minimum enclosing ball task and consists of point clouds randomly sampled from a $d$-dimensional ball. The 'Uniform Ellipse' dataset is used for the minimum enclosing ellipse task and consists of point clouds sampled from randomly scaled and rotated ellipses. The 'Uniform Annulus' dataset is used for the minimum enclosing annulus task and consists of point clouds sampled from annuli with widths from 0.1 to 3. See [43] and [26] for more details regarding SQUID and ModelNet, respectively.

|  | Synthetic | | | Real | |
|---|---|---|---|---|---|
|  | Uniform Ball | Uniform Ellipse | Uniform Annulus | ModelNet | SQUID |
| $|P_{\text{train}}|$ | 200 | 100 | 100 | 200 | 350 |
| $|P_{\text{test}}|$ | 200 | 100 | 100 | 200 | 350 |
| # train point clouds | 3000 | 3000 | 3000 | 2048 | 774 |
| # test point clouds | 3000 | 3000 | 3000 | 2048 | 332 |
| Input dim. | 2, 3 | 2 | 2 | 3 | 2 |
| Bounds | $[-5, 5]$ | $[-5, 5]$ | $[-5, 5]$ | $[-5, 5]$ | $[0, 450]$ |

the encode-decode-process models (used for extent measure tasks), we fix the processor to take input from $\mathbb{R}^5$, have three blocks (either SumFormer or Transformer) and output 150 points from $\mathcal{N}_{\phi_\varepsilon}$. We choose this configuration as these hyperparameters had the best performance on the relaxed-$\varepsilon$-coreset task in $\mathbb{R}^5$. Additionally, for the frozen training regime, the processor networks ($\mathcal{N}_{\phi_\varepsilon}$) take input of a point cloud $\widehat{P}$ in $\mathbb{R}^5$ (which could contain arbitrary number of points), output $k = 150$ points, and are pre-trained on the 'Mixed Synthetic' data in $\mathbb{R}^5$. Detailed dataset specifications for the relaxed-$\varepsilon$-kernel experiments are given in Table 3 and those for extent-measure tasks are given Table 4. All code (including models, hyperparameter settings, and synthetic dataset generation) is available.

**Size generalization for extent measures.** Here we described the train and test set details for the size generalization experiments on the extent measure tasks described in Section 5, Figure 6 as well in our motivating example in Section 2. Each model used for our size generalization experiment is trained on the synthetic $\mathbb{R}^2$ datasets detailed in Table 4. For the minimum enclosing ball (MEB) task, test point clouds have sizes $N \in 500, 1000, \ldots, 17000$, with the bounding box from which test points are sampled defined as $[-N/500, N/500]^2$. For the minimum enclosing ellipse (MEE) and minimum enclosing annulus (MEA) tasks, test point clouds have sizes $N \in 100, 200, \ldots, 3000$, and the bounding box scales as $[-N/100, N/100]^2$.

**Hyperparameter details.** For our hyperparameter search on the relaxed-$\varepsilon$-kernel, we chose the depth of the network from $\{2, 3, 4\}$, the hidden dimension from $\{128, 256, 512, 1024\}$, and the embedding dimension from $16, 32$. All results in our paper are reported on the best performing models from our hyperparameter search. For the extent-measure models, we fix the hyperparameters of the processor based on the best configuration from the $\varepsilon$-kernel task and search over the hyperparameters (width/depth) of the encoder and decoder networks. The depth of both the encoder and decoder MLPs are sampled from $\{2, 3, 4\}$ and the width from $\{64, 128, 256\}$. Finally, we also tune the number of epochs trained and chose to test the models trained for 200 epochs as they had the best validation loss. We note that we did not observe the transformer OOD performance to improve with longer training times.

**Loss functions.** Here, we list all loss functions used to train each task given an input point set $P$.

- **Relaxed-$\varepsilon$-kernel:** To train a model to approximate relaxed-$\varepsilon$-kernel, at each epoch $t$, we first sample a random set of 100 directions $\Omega_t$. We then compute the difference between the max projection of $P$ and $\mathcal{N}_{\phi_\varepsilon}(P)$ for $d \in \Omega_t$ as well as the difference between the min

projection of $P$ and $\mathcal{N}_{\phi_\varepsilon}(P)$ for $d \in D_t$:

$$\mathcal{L}_\varepsilon(P, \mathcal{N}_{\phi_\varepsilon}(P), D_t) = \frac{1}{|D_t|} \sum_{u \in \Omega_t} \left( |\max_{p \in P} \langle u, p \rangle - \max_{q \in \mathcal{N}_{\phi_\varepsilon}(P)} \langle u, q \rangle| + |\min_{p \in P} \langle u, p \rangle - \min_{q \in \mathcal{N}_{\phi_\varepsilon}(P)} \langle u, q \rangle| \right)$$

- **Minimum enclosing ball:** We aim to approximate the radius of the minimum enclosing ball. In order to validate that we have actually learned the ball, we also predict the center of ball. Therefore, given an input point set in $\mathbb{R}^d$, we configure $\mathcal{N}_{\text{extent}}$ such that $\mathcal{N}_{\text{extent}}(P) \in \mathbb{R}^{d+1}$ where the first $d$ coordinates, $\mathcal{N}_{\text{extent}}(P)_{1:d}$ represent the center and the last coordinate $\mathcal{N}_{\text{extent}}(P)_{d+1}$ represents the radius. Given the ground truth center $c \in \mathbb{R}^d$ and radius $r \in \mathbb{R}$, the loss function we use to train $\mathcal{N}_{\text{extent}}$ is

$$\mathcal{L}(c, r, \mathcal{N}_{\text{extent}}(P)) = \|c - \mathcal{N}_{\text{extent}}(P)_{1:d}\| + (r - \mathcal{N}_{\text{extent}}(P)_{d+1})^2$$

- **Minimum enclosing ellipse:** Recall that we aim to predict the minimum area enclosing ellipse. First, we note that we center all data at the origin for the minimum enclosing ellipse. Unlike the minimum enclosing ball, we do not predict the center for the ellipse. Given an input point set $P \subseteq \mathbb{R}^d$, we configure $\mathcal{N}_{\text{extent}}$ such that $\mathcal{N}_{\text{extent}}(P) \in \mathbb{R}^3$ where $\mathcal{N}_{\text{extent}}(P)_1$ represents the major radius, $\mathcal{N}_{\text{extent}}(P)_2$ represents the minor radius, and $\mathcal{N}_{\text{extent}}(P)_3$ represents the angle of rotation. Given the ground truth major radius $r_{\text{maj}} \in \mathbb{R}$, minor radius $r_{\text{min}} \in \mathbb{R}$ and angle of rotation $\theta \in \mathbb{R}$, the loss function we use to train $\mathcal{N}_{\text{extent}}$ is

$$\mathcal{L}(r_{\text{maj}}, r_{\text{min}}, \theta, \mathcal{N}_{\text{extent}}(P)) = (r_{\text{maj}} - \mathcal{N}_{\text{extent}}(P)_1)^2 + (r_{\text{min}} - \mathcal{N}_{\text{extent}}(P)_2)^2 \\ + (\sin(\theta) - \sin(\mathcal{N}_{\text{extent}}(P)_3))^2 + (\cos(\theta) - \cos(\mathcal{N}_{\text{extent}}(P)_3))^2$$

- **Minimum enclosing annulus:** We aim to predict the minimum width of the minimum enclosing annulus, which can be computed by predicting the inner and outer radii of the annulus and then taking their difference. Given an input point set $P \subseteq \mathbb{R}^d$, $\mathcal{N}_{\text{extent}}(P) \in \mathbb{R}^2$ where $\mathcal{N}_{\text{extent}}(P)_{d+1}$ represents the inner radius and $\mathcal{N}_{\text{extent}}(P)_{d+2}$ represents the outer radius. Then given the ground truth inner radius $r_{\text{inner}} \in \mathbb{R}$ and outer radius $r_{\text{outer}} \in \mathbb{R}$, the loss function we use to train $\mathcal{N}_{\text{extent}}$ is

$$\mathcal{L}(c, r_{\text{inner}}, r_{\text{outer}}, \mathcal{N}_{\text{extent}}(P)) = \frac{|r_{\text{inner}} - \mathcal{N}_{\text{extent}}(P)_1|}{r_{\text{inner}}} + \frac{|r_{\text{outer}} - \mathcal{N}_{\text{extent}}(P)_2|}{r_{\text{outer}}}$$

### A.4.1  Additional experiments for approximating relaxed-$\varepsilon$-kernels

First, we note that we include all complete tables from the main text with error bars in Table 5 and Table 6. We additionally include results on datasets in higher-dimensional synthetic datasets in $\mathbb{R}^5$ in Table 7 and $\mathbb{R}^{10}$ in Table 8. We see largely the same trends, where the $\mathcal{S}_\varepsilon$ performs comparably to $\mathcal{T}_\varepsilon$ on in-distribution data and then $\mathcal{S}_\varepsilon$ performs much better than $\mathcal{T}_\varepsilon$ on out of distribution data. Although the classical $\varepsilon$-kernel algorithm sometimes has lower directional error than $\mathcal{S}_\varepsilon$, we compare the runtime of the $\varepsilon$-kernel algorithm with all neural approaches in Figure 11 and show that our neural models are generally much more efficient at inference time than algorithmic approaches. We also provide more extensive results in Tables 9, 10, and 11, where we detail the performance of each neural approach on different types of synthetic and real datasets.

For the relaxed-$\varepsilon$-kernel tasks, we also provide several sensitivity analyses showing the effect of input dimension, fatness of point sets, and the size of relaxed-$\varepsilon$-kernel on the direction error ($\mathcal{E}_{\text{dir}}$). For each of these sensitivity analyses, we also provide a baseline comparison to the relaxed-$\varepsilon$-kernel algorithm described in Algorithm 1 given $\varepsilon = 0.1$ in order to analyze the behavior of the neural network with respect to its aligned algorithm. In all cases, the data-driven way of learning the relaxed-$\varepsilon$-kernel presents significantly more accurate (in terms of directional error) coresets than the baseline algorithmic implementation.

**Effect of input dimension.**  From Theorem 3.4, we know that the accuracy of the approximation of the relaxed-$\varepsilon$-kernel will depend on the dimension of the input point cloud. We verify this empirically in Figure 7 across different sizes of $\varepsilon$-kernels. For this experiment, we train and test each model on the point clouds with 500 points sampled uniformly from a ball in $\mathbb{R}^D$ where $D \in \{2, 3, 5\}$. We can see in Figure 7 that $\mathcal{E}_{\text{dir}}$ increases as the input dimension of the point cloud increases.

**Effect of $\alpha$-fatness.** Similar to the relationship between error and input point cloud dimension, we know from Theorem 3.4 that the accuracy of the approximation of the relaxed-$\varepsilon$-kernel will depend on the $\alpha$-fatness of the input point set. To verify this, we train both the SumFormer and Transformer model on point clouds with 500 points sampled uniformly from a balls in $\mathbb{R}^3$ and $\mathbb{R}^2$ (disks in the case of $\mathbb{R}^2$). We then test on point clouds of 100 points sampled from ellipsoids which have their minor axes scaled to simulate point clouds with a range of $\alpha$-fatness. The results are reported in Figure 8. Indeed, we see that as $\alpha$ increases (i.e. the point cloud becomes fatter) the accuracy of our approach increases.

**Effect of output size.** In order to examine the effect of the output relaxed-$\varepsilon$-kernel size for the quality of the approximation, we plot the normalized output size (i.e. number of relaxed-$\varepsilon$-kernel points/total number of input points) against the directional width error. See Figure 9. As expected, we see that the error decreases as the output size increases.

**Effect of input point set size.** We also examine the out-of-distribution capabilities of the models in terms of generalizing to larger point sets. For this experiment, we train each model on the point clouds with 500 points sampled uniformly from a ball in $\mathbb{R}^D$ where $D \in \{2, 3, 5\}$ and then test the models on much larger point clouds (up to 1000 points per cloud) sampled from the uniform ball. The results are reported in Figure 10 and we find that the SumFormer model generalizes especially well to out-of-distribution point set sizes (maintains low error).

### A.4.2 Additional experiments for approximation of extent measures

We include the full results from the main text (which include the percentage of points excluded from each covering object). The full results each extent measure task are included in Tables 12, 13, 14, and 15 for minimum enclosing ball in $\mathbb{R}^2$, minimum enclosing ball in $\mathbb{R}^3$, minimum enclosing ellipse in $\mathbb{R}^2$, and minimum enclosing annulus $\mathbb{R}^2$, respectively. We also include comparisons to classical algorithmic approximations in our full experimental tables. In particular, we evaluate the standard pipeline of first computing an $\varepsilon$-kernel and then applying the exact algorithm to the resulting coreset. For MEB and MEE, we use Welzl's algorithm as the exact solver; for MEA, we use the quadratic programming formulation implemented in the Computational Geometry Algorithms Library (CGAL) [35]. Additionally, Figure 12 reports the inference times of the neural models alongside the runtimes of the corresponding algorithmic approaches.

First, note that our $\varepsilon$-kernel–aligned model, the frozen $\mathcal{S}_{\text{extent}}$, misses a much smaller proportion of points than other models (even those with lower radius or width prediction error) indicating that its predicted shape more accurately captures the true enclosing object. Notice that while classical algorithms can achieve higher accuracy in some cases, our neural models outperform them on the minimum enclosing annulus task, which is more challenging than the other extent measure tasks as it is unfaithful and non LP-type. Moreover, as shown in Figure 12, our neural approaches are substantially faster than all algorithmic baselines (both the exact and the approximation), particularly for the minimum enclosing annulus. Finally, as emphasized in the main text, neural approximations offer a key conceptual advantage over classical algorithms beyond gains in speed: they do not require explicit knowledge of the underlying optimization objective. To train our neural models, only labeled data and the knowledge that the problem belongs to the general family of extent-measure tasks are needed. In contrast, classical algorithmic methods require full specification of the objective and often the design of a custom algorithm to solve each specific problem.

Finally, Figure 13 presents additional results for the size-generalization experiment shown in Figure 6, where we record the proportion of points excluded as the number of points per cloud increases. Not only does the frozen $\mathcal{S}_{\text{extent}}$ model outperform all other neural baselines in terms of size generalization and the error (as seen in Figure 6 but also the predicted output from the frozen $\mathcal{S}_{\text{extent}}$ model misses a low proportion of points. Notably, models that exclude fewer points than the frozen $\mathcal{S}$extent tend to exhibit higher width or radius error, indicating an overestimate of the enclosing object. This can also be seen in visualizations of the predicted output from each model in Figure 14, where we see that the frozen $\mathcal{S}_{\text{extent}}$ model can better adapt to out-of-distribution input data.

## A.5 Figures and Tables

Table 5: $\mathcal{E}_{\mathrm{dir}}$ on datasets in $\mathbb{R}^2$ for an output relaxed-$\varepsilon$-kernel of 16 points. We compare Sumformer and Transformer neural approaches across three different test sets for each train set. Note that 'Mixed Synthetic' has no 'OOD, Synthetic' because the train set contains all types of synthetic data. Boldface indicates the lower error per row.

| Train Set | Method | In-dist. | OOD Synthetic | OOD SQUID |
|---|---|---|---|---|
| Ellipse | $\mathcal{S}_\varepsilon$ | $0.060 \pm 0.026$ | $\mathbf{0.044 \pm 0.042}$ | $\mathbf{0.127 \pm 0.111}$ |
| | $\mathcal{T}_\varepsilon$ | $\mathbf{0.029 \pm 0.013}$ | $0.098 \pm 0.056$ | $0.275 \pm 0.112$ |
| | $\mathcal{S}_{\varepsilon,B}$ | $0.131 \pm 0.029$ | $0.354 \pm 0.172$ | $0.993 \pm 0.002$ |
| | $\mathcal{T}_{\varepsilon,B}$ | $0.117 \pm 0.045$ | $0.63 \pm 0.115$ | $0.976 \pm 0.007$ |
| Gaussian Mixture | $\mathcal{S}_\varepsilon$ | $\mathbf{0.014 \pm 0.013}$ | $\mathbf{0.043 \pm 0.040}$ | $\mathbf{0.028 \pm 0.043}$ |
| | $\mathcal{T}_\varepsilon$ | $0.027 \pm 0.024$ | $0.097 \pm 0.049$ | $0.218 \pm 0.101$ |
| | $\mathcal{S}_{\varepsilon,B}$ | $0.146 \pm 0.062$ | $0.146 \pm 0.057$ | $0.962 \pm 0.011$ |
| | $\mathcal{T}_{\varepsilon,B}$ | $0.19 \pm 0.087$ | $0.383 \pm 0.067$ | $0.94 \pm 0.015$ |
| SQUID | $\mathcal{S}_\varepsilon$ | $\mathbf{0.028 \pm 0.022}$ | $0.504 \pm 0.114$ | $\mathbf{0.028 \pm 0.022}$ |
| | $\mathcal{T}_\varepsilon$ | $0.061 \pm 0.052$ | $\mathbf{0.40 \pm 0.109}$ | $0.061 \pm 0.052$ |
| | $\mathcal{S}_{\varepsilon,B}$ | $0.421 \pm 0.511$ | $7.137 \pm 3.892$ | $0.421 \pm 0.511$ |
| | $\mathcal{T}_{\varepsilon,B}$ | $0.228 \pm 0.207$ | $14.561 \pm 6.632$ | $0.228 \pm 0.207$ |
| Mixed Synthetic | $\mathcal{S}_\varepsilon$ | $\mathbf{0.027 \pm 0.018}$ | – | $\mathbf{0.065 \pm 0.071}$ |
| | $\mathcal{T}_\varepsilon$ | $0.032 \pm 0.024$ | – | $0.296 \pm 0.113$ |
| | $\mathcal{S}_{\varepsilon,B}$ | $0.094 \pm 0.058$ | - | $0.971 \pm 0.008$ |
| | $\mathcal{T}_{\varepsilon,B}$ | $0.203 \pm 0.181$ | - | $0.939 \pm 0.017$ |
| Algorithmic Baselines | $\mathcal{B}_{\varepsilon,R}$ | - | $0.180 \pm 0.079$ | $0.228 \pm 0.139$ |
| | $\mathcal{B}_{\varepsilon,E}$ | - | $0.040 \pm 0.049$ | $0.062 \pm 0.095$ |

Table 6: $\mathcal{E}_{\mathrm{dir}}$ for predicted relaxed-$\varepsilon$-kernels of 64 points in $\mathbb{R}^3$. We compare Sumformer and Transformer neural approaches across three different test sets for each train set. Note that 'Mixed Synthetic' has no 'OOD, Synthetic' because the train set contains all types of synthetic data. Boldface indicates the lower error per row.

| | | | Test Sets | |
| | | | OOD | |
| Train Set | Method | In-Dist. | Synthetic | ModelNet |
|---|---|---|---|---|
| Ellipse | $\mathcal{S}_\varepsilon$ | $\mathbf{0.049 \pm 0.019}$ | $\mathbf{0.032 \pm 0.024}$ | $\mathbf{0.066 \pm 0.053}$ |
| | $\mathcal{T}_\varepsilon$ | $0.055 \pm 0.021$ | $0.106 \pm 0.076$ | $0.147 \pm 0.077$ |
| | $\mathcal{S}_{\varepsilon,\mathrm{B}}$ | $0.221 \pm 0.039$ | $0.462 \pm 0.098$ | $4.736 \pm 3.49$ |
| | $\mathcal{T}_{\varepsilon,\mathrm{B}}$ | $0.243 \pm 0.056$ | $0.555 \pm 0.063$ | $3.603 \pm 5.454$ |
| Gaussian Mixture | $\mathcal{S}_\varepsilon$ | $\mathbf{0.047 \pm 0.027}$ | $\mathbf{0.039 \pm 0.026}$ | $\mathbf{0.064 \pm 0.050}$ |
| | $\mathcal{T}_\varepsilon$ | $0.054 \pm 0.027$ | $0.066 \pm 0.033$ | $0.250 \pm 0.125$ |
| | $\mathcal{S}_{\varepsilon,\mathrm{B}}$ | $0.274 \pm 0.049$ | $0.289 \pm 0.029$ | $1.949 \pm 3.49$ |
| | $\mathcal{T}_{\varepsilon,\mathrm{B}}$ | $0.357 \pm 0.146$ | $0.851 \pm 0.104$ | $4.486 \pm 5.514$ |
| ModelNet | $\mathcal{S}_\varepsilon$ | $\mathbf{0.041 \pm 0.037}$ | $\mathbf{0.099 \pm 0.077}$ | $\mathbf{0.041 \pm 0.037}$ |
| | $\mathcal{T}_\varepsilon$ | $0.049 \pm 0.044$ | $0.265 \pm 0.118$ | $0.049 \pm 0.044$ |
| | $\mathcal{S}_{\varepsilon,\mathrm{B}}$ | $0.378 \pm 0.621$ | $0.542 \pm 0.069$ | $0.379 \pm 0.621$ |
| | $\mathcal{T}_{\varepsilon,\mathrm{B}}$ | $0.542 \pm 0.952$ | $0.658 \pm 0.142$ | $0.542 \pm 0.952$ |
| Mixed Synthetic | $\mathcal{S}_\varepsilon$ | $\mathbf{0.035 \pm 0.029}$ | – | $\mathbf{0.055 \pm 0.048}$ |
| | $\mathcal{T}_\varepsilon$ | $0.042 \pm 0.038$ | – | $0.117 \pm 0.071$ |
| | $\mathcal{S}_{\varepsilon,\mathrm{B}}$ | $0.174 \pm 0.054$ | - | $1.518 \pm 2.288$ |
| | $\mathcal{T}_{\varepsilon,\mathrm{B}}$ | $0.289 \pm 0.179$ | - | $4.356 \pm 5.257$ |
| Algorithmic Baselines | $\mathcal{B}_{\varepsilon,R}$ | - | $1.016 \pm 9.988$ | $2.176 \pm 2.849$ |
| | $\mathcal{B}_{\varepsilon,E}$ | - | $0.062 \pm 0.040$ | $0.050 \pm 0.047$ |

Table 7: $\mathcal{E}_{\mathrm{dir}}$ for predicted relaxed-$\varepsilon$-kernels of 200 points in $\mathbb{R}^5$. We compare Sumformer and Transformer neural approaches for in-distribution and out-of-distribution synthetic test sets. Note that there are no real datasets in $\mathbb{R}^5$ so the only test sets here are synthetic.

| Train Set | Method | In-Distribution | OOD, Synthetic |
|---|---|---|---|
| Uniform Ball | $\mathcal{S}_\varepsilon$ | $\mathbf{0.084 \pm 0.016}$ | $\mathbf{0.091 \pm 0.041}$ |
| | $\mathcal{T}_\varepsilon$ | $0.118 \pm 0.022$ | $0.29 \pm 0.117$ |
| | $\mathcal{S}_{\varepsilon,\mathrm{B}}$ | $0.238 \pm 0.029$ | $0.694 \pm 0.234$ |
| | $\mathcal{T}_{\varepsilon,\mathrm{B}}$ | $0.35 \pm 0.043$ | $0.575 \pm 0.249$ |
| Ellipse | $\mathcal{S}_\varepsilon$ | $0.085 \pm 0.025$ | $\mathbf{0.077 \pm 0.027}$ |
| | $\mathcal{T}_\varepsilon$ | $\mathbf{0.084 \pm 0.024}$ | $0.196 \pm 0.133$ |
| | $\mathcal{S}_{\varepsilon,\mathrm{B}}$ | $0.399 \pm 0.108$ | $0.589 \pm 0.256$ |
| | $\mathcal{T}_{\varepsilon,\mathrm{B}}$ | $0.343 \pm 0.096$ | $0.614 \pm 0.167$ |
| Single Gaussian | $\mathcal{S}_\varepsilon$ | $\mathbf{0.065 \pm 0.018}$ | $\mathbf{0.10 \pm 0.033}$ |
| | $\mathcal{T}_\varepsilon$ | $0.065 \pm 0.022$ | $0.183 \pm 0.053$ |
| | $\mathcal{S}_{\varepsilon,\mathrm{B}}$ | $0.352 \pm 0.048$ | $3.75 \pm 0.154$ |
| | $\mathcal{T}_{\varepsilon,\mathrm{B}}$ | $0.333 \pm 0.044$ | $3.941 \pm 0.112$ |
| Gaussian Mixture | $\mathcal{S}_\varepsilon$ | $\mathbf{0.084 \pm 0.028}$ | $\mathbf{0.087 \pm 0.027}$ |
| | $\mathcal{T}_\varepsilon$ | $0.091 \pm 0.025$ | $0.109 \pm 0.032$ |
| | $\mathcal{S}_{\varepsilon,\mathrm{B}}$ | $0.717 \pm 0.021$ | $0.465 \pm 0.184$ |
| | $\mathcal{T}_{\varepsilon,\mathrm{B}}$ | $0.757 \pm 0.278$ | $1.488 \pm 0.344$ |
| Mixed Synthetic | $\mathcal{S}_\varepsilon$ | $\mathbf{0.075 \pm 0.025}$ | - |
| | $\mathcal{T}_\varepsilon$ | $0.087 \pm 0.032$ | - |
| | $\mathcal{S}_{\varepsilon,\mathrm{B}}$ | $0.347 \pm 0.188$ | - |
| | $\mathcal{T}_{\varepsilon,\mathrm{B}}$ | $0.879 \pm 0.599$ | - |
| Algorithmic Baselines | $\mathcal{B}_{\varepsilon,R}$ | - | $0.653 \pm 0.366$ |
| | $\mathcal{B}_{\varepsilon,E}$ | - | $0.083 \pm 0.033$ |

Table 8: $\mathcal{E}_{\mathrm{dir}}$ for predicted relaxed-$\varepsilon$-kernels of 200 points in $\mathbb{R}^{10}$. We compare Sumformer and Transformer neural approaches for in-distribution and out-of-distribution synthetic test sets. Note that there are no real datasets in $\mathbb{R}^5$ so the only test sets here are synthetic.

| Train Set | Method | In-Distribution | OOD, Synthetic |
|---|---|---|---|
| Uniform Ball | $\mathcal{S}_\varepsilon$ | $0.367 \pm 0.0289$ | $0.301 \pm 0.074$ |
| | $\mathcal{T}_\varepsilon$ | $\mathbf{0.259 \pm 0.0227}$ | $\mathbf{0.292 \pm 0.044}$ |
| | $\mathcal{S}_{\varepsilon,\mathrm{B}}$ | $0.290 \pm 0.033$ | $0.459 \pm 0.214$ |
| | $\mathcal{T}_{\varepsilon,\mathrm{B}}$ | $0.266 \pm 0.029$ | $0.435 \pm 0.208$ |
| Ellipse | $\mathcal{S}_\varepsilon$ | $\mathbf{0.222 \pm 0.037}$ | $\mathbf{0.275 \pm 0.062}$ |
| | $\mathcal{T}_\varepsilon$ | $0.251 \pm 0.042$ | $0.276 \pm 0.041$ |
| | $\mathcal{S}_{\varepsilon,\mathrm{B}}$ | $0.693 \pm 0.166$ | $0.518 \pm 0.209$ |
| | $\mathcal{T}_{\varepsilon,\mathrm{B}}$ | $0.777 \pm 0.192$ | $0.582 \pm 0.281$ |
| Mixed Synthetic | $\mathcal{S}_\varepsilon$ | $\mathbf{0.241 \pm 0.036}$ | - |
| | $\mathcal{T}_\varepsilon$ | $0.258 \pm 0.034$ | - |
| | $\mathcal{S}_{\varepsilon,\mathrm{B}}$ | $0.530 \pm 0.281$ | - |
| | $\mathcal{T}_{\varepsilon,\mathrm{B}}$ | $0.535 \pm 0.287$ | - |
| Algorithmic Baselines | $\mathcal{B}_{\varepsilon,R}$ | - | $0.621 \pm 0.225$ |
| | $\mathcal{B}_{\varepsilon,E}$ | - | $0.199 \pm 0.034$ |

Table 9: Directional error across 2D datasets for various $\epsilon$-kernel sizes. Note that we already report performance for the mixed synthetic data in the tables in the main text so we do not report the performance here.

| Train Set | $\epsilon$ | Method | Uniform Disk | Ellipse | Single Gaussian | Gaussian Mixture | SQUID |
|---|---|---|---|---|---|---|---|
| Uniform Disk | 16 | $\mathcal{S}_\varepsilon$ | $0.040 \pm 0.013$ | $0.109 \pm 0.081$ | $0.125 \pm 0.057$ | $0.074 \pm 0.059$ | $0.101 \pm 0.065$ |
| | | $\mathcal{T}_\varepsilon$ | $0.040 \pm 0.013$ | $0.200 \pm 0.094$ | $0.370 \pm 0.078$ | $0.346 \pm 0.144$ | $0.297 \pm 0.110$ |
| | | $\mathcal{S}_{\varepsilon,B}$ | $0.237 \pm 0.009$ | $0.544 \pm 0.159$ | $0.767 \pm 0.017$ | $0.712 \pm 0.115$ | $0.995 \pm 0.002$ |
| | | $\mathcal{T}_{\varepsilon,B}$ | $0.061 \pm 0.007$ | $0.456 \pm 0.167$ | $0.720 \pm 0.019$ | $0.512 \pm 0.159$ | $0.973 \pm 0.008$ |
| | 64 | $\mathcal{S}_\varepsilon$ | $0.009 \pm 0.004$ | $0.065 \pm 0.051$ | $0.214 \pm 0.057$ | $0.061 \pm 0.053$ | $0.264 \pm 0.093$ |
| | | $\mathcal{T}_\varepsilon$ | $0.007 \pm 0.004$ | $0.158 \pm 0.079$ | $0.385 \pm 0.058$ | $0.287 \pm 0.123$ | $0.236 \pm 0.099$ |
| | | $\mathcal{S}_{\varepsilon,B}$ | $0.062 \pm 0.008$ | $0.443 \pm 0.165$ | $0.703 \pm 0.020$ | $0.561 \pm 0.123$ | $0.975 \pm 0.007$ |
| | | $\mathcal{T}_{\varepsilon,B}$ | $0.044 \pm 0.011$ | $0.456 \pm 0.186$ | $0.702 \pm 0.020$ | $0.534 \pm 0.134$ | $0.972 \pm 0.009$ |
| | 200 | $\mathcal{S}_\varepsilon$ | $0.006 \pm 0.003$ | $0.069 \pm 0.057$ | $0.128 \pm 0.062$ | $0.097 \pm 0.099$ | $0.174 \pm 0.085$ |
| | | $\mathcal{T}_\varepsilon$ | $0.005 \pm 0.003$ | $0.199 \pm 0.110$ | $0.465 \pm 0.041$ | $0.262 \pm 0.110$ | $0.427 \pm 0.109$ |
| | | $\mathcal{S}_{\varepsilon,B}$ | $0.111 \pm 0.008$ | $0.473 \pm 0.165$ | $0.721 \pm 0.019$ | $0.589 \pm 0.103$ | $0.982 \pm 0.005$ |
| | | $\mathcal{T}_{\varepsilon,B}$ | $0.123 \pm 0.010$ | $0.484 \pm 0.167$ | $0.739 \pm 0.018$ | $0.599 \pm 0.144$ | $0.975 \pm 0.008$ |
| Ellipse | 16 | $\mathcal{S}_\varepsilon$ | $0.045 \pm 0.013$ | $0.058 \pm 0.026$ | $0.042 \pm 0.039$ | $0.067 \pm 0.057$ | $0.130 \pm 0.109$ |
| | | $\mathcal{T}_\varepsilon$ | $0.031 \pm 0.010$ | $0.029 \pm 0.013$ | $0.093 \pm 0.056$ | $0.050 \pm 0.041$ | $0.272 \pm 0.108$ |
| | | $\mathcal{S}_{\varepsilon,B}$ | $0.110 \pm 0.012$ | $0.131 \pm 0.029$ | $0.458 \pm 0.039$ | $0.495 \pm 0.271$ | $0.993 \pm 0.002$ |
| | | $\mathcal{T}_{\varepsilon,B}$ | $0.715 \pm 0.061$ | $0.119 \pm 0.046$ | $0.601 \pm 0.035$ | $0.575 \pm 0.247$ | $0.975 \pm 0.007$ |
| | 64 | $\mathcal{S}_\varepsilon$ | $0.011 \pm 0.005$ | $0.015 \pm 0.008$ | $0.034 \pm 0.050$ | $0.078 \pm 0.086$ | $0.088 \pm 0.070$ |
| | | $\mathcal{T}_\varepsilon$ | $0.013 \pm 0.005$ | $0.012 \pm 0.006$ | $0.116 \pm 0.057$ | $0.033 \pm 0.032$ | $0.241 \pm 0.087$ |
| | | $\mathcal{S}_{\varepsilon,B}$ | $0.121 \pm 0.010$ | $0.146 \pm 0.029$ | $0.438 \pm 0.039$ | $0.456 \pm 0.210$ | $0.976 \pm 0.007$ |
| | | $\mathcal{T}_{\varepsilon,B}$ | $0.674 \pm 0.049$ | $0.101 \pm 0.040$ | $0.613 \pm 0.031$ | $0.593 \pm 0.231$ | $0.978 \pm 0.007$ |
| | 200 | $\mathcal{S}_\varepsilon$ | $0.004 \pm 0.002$ | $0.005 \pm 0.003$ | $0.033 \pm 0.046$ | $0.057 \pm 0.077$ | $0.118 \pm 0.073$ |
| | | $\mathcal{T}_\varepsilon$ | $0.010 \pm 0.004$ | $0.008 \pm 0.005$ | $0.030 \pm 0.027$ | $0.034 \pm 0.033$ | $0.197 \pm 0.079$ |
| | | $\mathcal{S}_{\varepsilon,B}$ | $0.098 \pm 0.011$ | $0.118 \pm 0.025$ | $0.445 \pm 0.038$ | $0.435 \pm 0.215$ | $0.989 \pm 0.003$ |
| | | $\mathcal{T}_{\varepsilon,B}$ | $0.217 \pm 0.028$ | $0.076 \pm 0.030$ | $0.436 \pm 0.038$ | $0.583 \pm 0.413$ | $0.961 \pm 0.011$ |
| Gaussian | 16 | $\mathcal{S}_\varepsilon$ | $0.056 \pm 0.015$ | $0.097 \pm 0.052$ | $0.032 \pm 0.020$ | $0.071 \pm 0.049$ | $0.091 \pm 0.069$ |
| | | $\mathcal{T}_\varepsilon$ | $0.062 \pm 0.017$ | $0.069 \pm 0.029$ | $0.019 \pm 0.014$ | $0.080 \pm 0.052$ | $0.206 \pm 0.095$ |
| | | $\mathcal{S}_{\varepsilon,B}$ | $0.924 \pm 0.031$ | $1.174 \pm 0.604$ | $0.108 \pm 0.030$ | $1.256 \pm 0.638$ | $0.946 \pm 0.015$ |
| | | $\mathcal{T}_{\varepsilon,B}$ | $2.417 \pm 0.109$ | $2.376 \pm 1.023$ | $0.246 \pm 0.053$ | $2.200 \pm 1.126$ | $0.922 \pm 0.024$ |
| | 64 | $\mathcal{S}_\varepsilon$ | $0.029 \pm 0.009$ | $0.050 \pm 0.029$ | $0.011 \pm 0.010$ | $0.039 \pm 0.032$ | $0.078 \pm 0.068$ |
| | | $\mathcal{T}_\varepsilon$ | $0.052 \pm 0.015$ | $0.056 \pm 0.027$ | $0.005 \pm 0.007$ | $0.083 \pm 0.058$ | $0.241 \pm 0.100$ |
| | | $\mathcal{S}_{\varepsilon,B}$ | $0.884 \pm 0.032$ | $1.102 \pm 0.612$ | $0.101 \pm 0.032$ | $1.314 \pm 0.741$ | $0.949 \pm 0.015$ |
| | | $\mathcal{T}_{\varepsilon,B}$ | $2.374 \pm 0.102$ | $2.209 \pm 0.941$ | $0.152 \pm 0.044$ | $2.147 \pm 1.089$ | $0.879 \pm 0.028$ |
| | 200 | $\mathcal{S}_\varepsilon$ | $0.017 \pm 0.007$ | $0.057 \pm 0.060$ | $0.003 \pm 0.004$ | $0.056 \pm 0.048$ | $0.067 \pm 0.058$ |
| | | $\mathcal{T}_\varepsilon$ | $0.047 \pm 0.015$ | $0.049 \pm 0.020$ | $0.004 \pm 0.006$ | $0.071 \pm 0.049$ | $0.220 \pm 0.099$ |
| | | $\mathcal{S}_{\varepsilon,B}$ | $0.927 \pm 0.031$ | $1.131 \pm 0.617$ | $0.114 \pm 0.030$ | $1.316 \pm 0.703$ | $0.944 \pm 0.019$ |
| | | $\mathcal{T}_{\varepsilon,B}$ | $2.456 \pm 0.125$ | $2.151 \pm 0.946$ | $0.144 \pm 0.041$ | $2.216 \pm 1.116$ | $0.910 \pm 0.024$ |
| Gaussian Mixture | 16 | $\mathcal{S}_\varepsilon$ | $0.024 \pm 0.008$ | $0.033 \pm 0.015$ | $0.071 \pm 0.059$ | $0.013 \pm 0.013$ | $0.028 \pm 0.037$ |
| | | $\mathcal{T}_\varepsilon$ | $0.040 \pm 0.011$ | $0.034 \pm 0.011$ | $0.063 \pm 0.050$ | $0.027 \pm 0.024$ | $0.214 \pm 0.100$ |
| | | $\mathcal{S}_{\varepsilon,B}$ | $0.067 \pm 0.010$ | $0.087 \pm 0.025$ | $0.188 \pm 0.043$ | $0.128 \pm 0.058$ | $0.967 \pm 0.011$ |
| | | $\mathcal{T}_{\varepsilon,B}$ | $0.686 \pm 0.051$ | $0.485 \pm 0.075$ | $0.272 \pm 0.047$ | $0.190 \pm 0.086$ | $0.944 \pm 0.015$ |
| | 64 | $\mathcal{S}_\varepsilon$ | $0.015 \pm 0.006$ | $0.018 \pm 0.010$ | $0.107 \pm 0.067$ | $0.006 \pm 0.006$ | $0.015 \pm 0.013$ |
| | | $\mathcal{T}_\varepsilon$ | $0.032 \pm 0.011$ | $0.017 \pm 0.008$ | $0.026 \pm 0.033$ | $0.009 \pm 0.010$ | $0.156 \pm 0.082$ |
| | | $\mathcal{S}_{\varepsilon,B}$ | $0.062 \pm 0.010$ | $0.084 \pm 0.024$ | $0.179 \pm 0.046$ | $0.121 \pm 0.052$ | $0.963 \pm 0.011$ |
| | | $\mathcal{T}_{\varepsilon,B}$ | $1.031 \pm 0.050$ | $0.609 \pm 0.105$ | $0.316 \pm 0.046$ | $0.164 \pm 0.075$ | $0.947 \pm 0.015$ |
| | 200 | $\mathcal{S}_\varepsilon$ | $0.005 \pm 0.003$ | $0.009 \pm 0.007$ | $0.100 \pm 0.047$ | $0.003 \pm 0.004$ | $0.012 \pm 0.018$ |
| | | $\mathcal{T}_\varepsilon$ | $0.030 \pm 0.012$ | $0.011 \pm 0.006$ | $0.032 \pm 0.033$ | $0.006 \pm 0.007$ | $0.149 \pm 0.084$ |
| | | $\mathcal{S}_{\varepsilon,B}$ | $0.086 \pm 0.010$ | $0.103 \pm 0.026$ | $0.193 \pm 0.046$ | $0.138 \pm 0.067$ | $0.966 \pm 0.011$ |
| | | $\mathcal{T}_{\varepsilon,B}$ | $1.706 \pm 0.082$ | $0.815 \pm 0.140$ | $0.356 \pm 0.052$ | $0.191 \pm 0.103$ | $0.940 \pm 0.014$ |
| SQUID | 16 | $\mathcal{S}_\varepsilon$ | $0.548 \pm 0.012$ | $0.528 \pm 0.027$ | $0.547 \pm 0.040$ | $0.395 \pm 0.183$ | $0.029 \pm 0.021$ |
| | | $\mathcal{T}_\varepsilon$ | $0.434 \pm 0.041$ | $0.403 \pm 0.060$ | $0.390 \pm 0.081$ | $0.368 \pm 0.163$ | $0.066 \pm 0.051$ |
| | | $\mathcal{S}_{\varepsilon,B}$ | $10.061 \pm 0.191$ | $9.954 \pm 3.629$ | $2.875 \pm 0.296$ | $10.191 \pm 4.073$ | $0.283 \pm 0.276$ |
| | | $\mathcal{T}_{\varepsilon,B}$ | $19.858 \pm 0.815$ | $18.238 \pm 6.542$ | $5.490 \pm 0.519$ | $16.164 \pm 6.500$ | $0.231 \pm 0.224$ |
| | 64 | $\mathcal{S}_\varepsilon$ | $0.287 \pm 0.014$ | $0.289 \pm 0.032$ | $0.411 \pm 0.047$ | $0.274 \pm 0.173$ | $0.009 \pm 0.008$ |
| | | $\mathcal{T}_\varepsilon$ | $0.373 \pm 0.025$ | $0.376 \pm 0.039$ | $0.409 \pm 0.057$ | $0.350 \pm 0.149$ | $0.062 \pm 0.042$ |
| | | $\mathcal{S}_{\varepsilon,B}$ | $16.621 \pm 0.499$ | $16.123 \pm 4.591$ | $6.463 \pm 0.608$ | $12.393 \pm 5.642$ | $0.224 \pm 0.202$ |
| | | $\mathcal{T}_{\varepsilon,B}$ | $22.050 \pm 1.050$ | $19.865 \pm 7.018$ | $5.916 \pm 0.611$ | $16.114 \pm 7.801$ | $0.234 \pm 0.241$ |
| | 200 | $\mathcal{S}_\varepsilon$ | $0.368 \pm 0.013$ | $0.339 \pm 0.034$ | $0.378 \pm 0.048$ | $0.255 \pm 0.190$ | $0.009 \pm 0.009$ |
| | | $\mathcal{T}_\varepsilon$ | $0.322 \pm 0.059$ | $0.346 \pm 0.069$ | $0.463 \pm 0.081$ | $0.379 \pm 0.140$ | $0.047 \pm 0.036$ |
| | | $\mathcal{S}_{\varepsilon,B}$ | $5.683 \pm 0.144$ | $5.374 \pm 2.142$ | $1.582 \pm 0.179$ | $6.337 \pm 4.633$ | $0.199 \pm 0.199$ |
| | | $\mathcal{T}_{\varepsilon,B}$ | $15.879 \pm 0.788$ | $15.438 \pm 4.420$ | $4.426 \pm 0.487$ | $13.109 \pm 5.951$ | $0.261 \pm 0.243$ |
| Mixed | 16 | $\mathcal{S}_\varepsilon$ | $0.038 \pm 0.012$ | $0.052 \pm 0.024$ | $0.019 \pm 0.015$ | $0.031 \pm 0.026$ | $0.064 \pm 0.073$ |
| | | $\mathcal{T}_\varepsilon$ | $0.035 \pm 0.011$ | $0.046 \pm 0.023$ | $0.017 \pm 0.014$ | $0.040 \pm 0.033$ | $0.307 \pm 0.112$ |
| | | $\mathcal{S}_{\varepsilon,B}$ | $0.046 \pm 0.008$ | $0.057 \pm 0.016$ | $0.069 \pm 0.017$ | $0.124 \pm 0.058$ | $0.971 \pm 0.008$ |
| | | $\mathcal{T}_{\varepsilon,B}$ | $0.577 \pm 0.0509$ | $0.646 \pm 0.267$ | $0.220 \pm 0.059$ | $0.512 \pm 0.329$ | $0.939 \pm 0.017$ |
| | 64 | $\mathcal{S}_\varepsilon$ | $0.010 \pm 0.004$ | $0.013 \pm 0.008$ | $0.002 \pm 0.004$ | $0.008 \pm 0.008$ | $0.023 \pm 0.030$ |
| | | $\mathcal{T}_\varepsilon$ | $0.016 \pm 0.007$ | $0.014 \pm 0.006$ | $0.001 \pm 0.003$ | $0.014 \pm 0.017$ | $0.183 \pm 0.080$ |
| | | $\mathcal{S}_{\varepsilon,B}$ | $0.109 \pm 0.010$ | $0.136 \pm 0.031$ | $0.108 \pm 0.029$ | $0.138 \pm 0.065$ | $0.967 \pm 0.010$ |
| | | $\mathcal{T}_{\varepsilon,B}$ | $1.339 \pm 0.094$ | $0.877 \pm 0.265$ | $0.320 \pm 0.073$ | $0.561 \pm 0.368$ | $0.922 \pm 0.024$ |
| | 200 | $\mathcal{S}_\varepsilon$ | $0.006 \pm 0.003$ | $0.008 \pm 0.005$ | $0.001 \pm 0.002$ | $0.005 \pm 0.006$ | $0.014 \pm 0.022$ |
| | | $\mathcal{T}_\varepsilon$ | $0.014 \pm 0.008$ | $0.012 \pm 0.008$ | $0.001 \pm 0.003$ | $0.013 \pm 0.014$ | $0.144 \pm 0.062$ |
| | | $\mathcal{S}_{\varepsilon,B}$ | $0.251 \pm 0.012$ | $0.285 \pm 0.121$ | $0.261 \pm 0.049$ | $0.391 \pm 0.222$ | $0.949 \pm 0.015$ |
| | | $\mathcal{T}_{\varepsilon,B}$ | $1.428 \pm 0.064$ | $1.133 \pm 0.366$ | $0.238 \pm 0.055$ | $0.468 \pm 0.289$ | $0.913 \pm 0.024$ |

Table 10: $\mathcal{E}_{\text{dir}}$ across all 3D datasets for various $\epsilon$-kernel sizes. Note that we already report performance for the mixed synthetic data in the tables in the main text so we do not report the performance here.

| Dataset | $\epsilon$ | Architecture | Uniform Ball | Ellipse | Single Gaussian | Gaussian Mixture | ModelNet |
|---|---|---|---|---|---|---|---|
| Uniform Ball | 16 | $\mathcal{S}_\epsilon$ | $0.180 \pm 0.030$ | $0.239 \pm 0.069$ | $0.159 \pm 0.045$ | $0.180 \pm 0.072$ | $0.161 \pm 0.094$ |
| | | $\mathcal{T}_\epsilon$ | $0.180 \pm 0.032$ | $0.358 \pm 0.085$ | $0.553 \pm 0.030$ | $0.421 \pm 0.123$ | $0.426 \pm 0.130$ |
| | | $\mathcal{S}_{\epsilon,B}$ | $0.251 \pm 0.031$ | $0.620 \pm 0.214$ | $0.774 \pm 0.014$ | $0.643 \pm 0.128$ | $2.887 \pm 4.045$ |
| | | $\mathcal{T}_{\epsilon,B}$ | $0.149 \pm 0.019$ | $0.569 \pm 0.2$ | $0.757 \pm 0.015$ | $0.623 \pm 0.133$ | $3.547 \pm 5.67$ |
| | 64 | $\mathcal{S}_\epsilon$ | $0.046 \pm 0.012$ | $0.051 \pm 0.018$ | $0.029 \pm 0.014$ | $0.050 \pm 0.028$ | $0.067 \pm 0.057$ |
| | | $\mathcal{T}_\epsilon$ | $0.041 \pm 0.010$ | $0.198 \pm 0.079$ | $0.389 \pm 0.047$ | $0.257 \pm 0.093$ | $0.361 \pm 0.111$ |
| | | $\mathcal{S}_{\epsilon,B}$ | $0.161 \pm 0.018$ | $0.582 \pm 0.204$ | $0.759 \pm 0.015$ | $0.673 \pm 0.086$ | $5.920 \pm 7.937$ |
| | | $\mathcal{T}_{\epsilon,B}$ | $0.149 \pm 0.019$ | $0.569 \pm 0.2$ | $0.757 \pm 0.015$ | $0.623 \pm 0.133$ | $3.547 \pm 5.67$ |
| | 200 | $\mathcal{S}_\epsilon$ | $0.021 \pm 0.007$ | $0.020 \pm 0.008$ | $0.007 \pm 0.007$ | $0.029 \pm 0.021$ | $0.053 \pm 0.047$ |
| | | $\mathcal{T}_\epsilon$ | $0.014 \pm 0.005$ | $0.158 \pm 0.065$ | $0.332 \pm 0.055$ | $0.237 \pm 0.097$ | $0.361 \pm 0.111$ |
| | | $\mathcal{S}_{\epsilon,B}$ | $0.152 \pm 0.019$ | $0.572 \pm 0.203$ | $0.755 \pm 0.015$ | $0.670 \pm 0.080$ | $6.110 \pm 7.866$ |
| | | $\mathcal{T}_{\epsilon,B}$ | $0.103 \pm 0.017$ | $0.543 \pm 0.195$ | $0.746 \pm 0.016$ | $0.603 \pm 0.141$ | $2.817 \pm 3.886$ |
| Ellipse | 16 | $\mathcal{S}_\epsilon$ | $0.178 \pm 0.029$ | $0.244 \pm 0.074$ | $0.159 \pm 0.045$ | $0.187 \pm 0.076$ | $0.149 \pm 0.093$ |
| | | $\mathcal{T}_\epsilon$ | $0.176 \pm 0.027$ | $0.244 \pm 0.078$ | $0.235 \pm 0.063$ | $0.235 \pm 0.111$ | $0.232 \pm 0.114$ |
| | | $\mathcal{S}_{\epsilon,B}$ | $0.188 \pm 0.014$ | $0.221 \pm 0.036$ | $0.562 \pm 0.027$ | $0.602 \pm 0.191$ | $3.180 \pm 3.799$ |
| | | $\mathcal{T}_{\epsilon,B}$ | $0.257 \pm 0.041$ | $0.232 \pm 0.048$ | $0.600 \pm 0.035$ | $0.898 \pm 0.466$ | $3.276 \pm 4.107$ |
| | 64 | $\mathcal{S}_\epsilon$ | $0.043 \pm 0.010$ | $0.049 \pm 0.019$ | $0.028 \pm 0.013$ | $0.051 \pm 0.029$ | $0.065 \pm 0.053$ |
| | | $\mathcal{T}_\epsilon$ | $0.047 \pm 0.011$ | $0.050 \pm 0.018$ | $0.196 \pm 0.068$ | $0.075 \pm 0.051$ | $0.163 \pm 0.089$ |
| | | $\mathcal{S}_{\epsilon,B}$ | $0.187 \pm 0.014$ | $0.221 \pm 0.039$ | $0.517 \pm 0.030$ | $0.659 \pm 0.262$ | $3.510 \pm 4.736$ |
| | | $\mathcal{T}_{\epsilon,B}$ | $0.425 \pm 0.046$ | $0.243 \pm 0.055$ | $0.625 \pm 0.027$ | $0.615 \pm 0.117$ | $3.603 \pm 5.454$ |
| | 200 | $\mathcal{S}_\epsilon$ | $0.021 \pm 0.007$ | $0.021 \pm 0.009$ | $0.008 \pm 0.007$ | $0.030 \pm 0.021$ | $0.049 \pm 0.045$ |
| | | $\mathcal{T}_\epsilon$ | $0.019 \pm 0.006$ | $0.019 \pm 0.009$ | $0.039 \pm 0.050$ | $0.087 \pm 0.074$ | $0.250 \pm 0.121$ |
| | | $\mathcal{S}_{\epsilon,B}$ | $0.177 \pm 0.022$ | $0.219 \pm 0.058$ | $0.531 \pm 0.030$ | $0.677 \pm 0.285$ | $3.848 \pm 4.900$ |
| | | $\mathcal{T}_{\epsilon,B}$ | $0.198 \pm 0.042$ | $0.219 \pm 0.070$ | $0.512 \pm 0.032$ | $0.907 \pm 0.523$ | $3.609 \pm 5.109$ |
| Single Gaussian | 16 | $\mathcal{S}_\epsilon$ | $0.180 \pm 0.030$ | $0.252 \pm 0.076$ | $0.156 \pm 0.043$ | $0.183 \pm 0.073$ | $0.200 \pm 0.087$ |
| | | $\mathcal{T}_\epsilon$ | $0.239 \pm 0.037$ | $0.232 \pm 0.051$ | $0.146 \pm 0.040$ | $0.227 \pm 0.082$ | $0.363 \pm 0.121$ |
| | | $\mathcal{S}_{\epsilon,B}$ | $1.85 \pm 0.068$ | $2.332 \pm 0.891$ | $0.248 \pm 0.04$ | $2.314 \pm 0.78$ | $7.204 \pm 8.48$ |
| | | $\mathcal{T}_{\epsilon,B}$ | $2.742 \pm 0.080$ | $3.300 \pm 1.244$ | $0.252 \pm 0.044$ | $3.480 \pm 1.063$ | $20.436 \pm 24.422$ |
| | 64 | $\mathcal{S}_\epsilon$ | $0.057 \pm 0.013$ | $0.080 \pm 0.034$ | $0.033 \pm 0.016$ | $0.063 \pm 0.032$ | $0.075 \pm 0.052$ |
| | | $\mathcal{T}_\epsilon$ | $0.078 \pm 0.016$ | $0.110 \pm 0.044$ | $0.037 \pm 0.015$ | $0.123 \pm 0.054$ | $0.257 \pm 0.124$ |
| | | $\mathcal{S}_{\epsilon,B}$ | $2.883 \pm 0.092$ | $3.565 \pm 1.341$ | $0.29 \pm 0.048$ | $2.6 \pm 1.09$ | $9.931 \pm 11.224$ |
| | | $\mathcal{T}_{\epsilon,B}$ | $2.762 \pm 0.083$ | $3.249 \pm 1.199$ | $0.261 \pm 0.041$ | $3.179 \pm 0.951$ | $15.886 \pm 17.536$ |
| | 200 | $\mathcal{S}_\epsilon$ | $0.034 \pm 0.009$ | $0.039 \pm 0.017$ | $0.013 \pm 0.010$ | $0.040 \pm 0.024$ | $0.059 \pm 0.047$ |
| | | $\mathcal{T}_\epsilon$ | $0.069 \pm 0.016$ | $0.075 \pm 0.026$ | $0.014 \pm 0.010$ | $0.088 \pm 0.044$ | $0.219 \pm 0.100$ |
| | | $\mathcal{S}_{\epsilon,B}$ | $1.179 \pm 0.052$ | $1.791 \pm 0.716$ | $0.17 \pm 0.032$ | $2.17 \pm 0.712$ | $10.473 \pm 14.457$ |
| | | $\mathcal{T}_{\epsilon,B}$ | $2.707 \pm 0.079$ | $3.286 \pm 1.236$ | $0.263 \pm 0.042$ | $2.997 \pm 0.903$ | $16.368 \pm 19.089$ |
| Gaussian Mixture | 16 | $\mathcal{S}_\epsilon$ | $0.183 \pm 0.031$ | $0.231 \pm 0.062$ | $0.163 \pm 0.046$ | $0.179 \pm 0.072$ | $0.154 \pm 0.092$ |
| | | $\mathcal{T}_\epsilon$ | $0.176 \pm 0.027$ | $0.204 \pm 0.047$ | $0.176 \pm 0.049$ | $0.178 \pm 0.069$ | $0.209 \pm 0.099$ |
| | | $\mathcal{S}_{\epsilon,B}$ | $0.215 \pm 0.015$ | $0.2145 \pm 0.039$ | $0.292 \pm 0.041$ | $0.259 \pm 0.069$ | $1.880 \pm 2.784$ |
| | | $\mathcal{T}_{\epsilon,B}$ | $2.175 \pm 0.103$ | $1.408 \pm 0.223$ | $0.392 \pm 0.049$ | $0.425 \pm 0.179$ | $5.125 \pm 6.170$ |
| | 64 | $\mathcal{S}_\epsilon$ | $0.054 \pm 0.013$ | $0.061 \pm 0.020$ | $0.032 \pm 0.019$ | $0.047 \pm 0.027$ | $0.065 \pm 0.053$ |
| | | $\mathcal{T}_\epsilon$ | $0.060 \pm 0.013$ | $0.059 \pm 0.016$ | $0.054 \pm 0.027$ | $0.054 \pm 0.027$ | $0.250 \pm 0.127$ |
| | | $\mathcal{S}_{\epsilon,B}$ | $0.261 \pm 0.015$ | $0.266 \pm 0.036$ | $0.343 \pm 0.035$ | $0.274 \pm 0.049$ | $1.941 \pm 3.493$ |
| | | $\mathcal{T}_{\epsilon,B}$ | $1.222 \pm 0.065$ | $0.952 \pm 0.198$ | $0.378 \pm 0.048$ | $0.357 \pm 0.146$ | $4.485 \pm 5.514$ |
| | 200 | $\mathcal{S}_\epsilon$ | $0.027 \pm 0.008$ | $0.030 \pm 0.013$ | $0.010 \pm 0.009$ | $0.023 \pm 0.014$ | $0.042 \pm 0.037$ |
| | | $\mathcal{T}_\epsilon$ | $0.032 \pm 0.010$ | $0.029 \pm 0.012$ | $0.021 \pm 0.019$ | $0.030 \pm 0.019$ | $0.232 \pm 0.098$ |
| | | $\mathcal{S}_{\epsilon,B}$ | $0.179 \pm 0.015$ | $0.215 \pm 0.047$ | $0.276 \pm 0.040$ | $0.284 \pm 0.100$ | $2.111 \pm 3.048$ |
| | | $\mathcal{T}_{\epsilon,B}$ | $1.550 \pm 0.086$ | $1.220 \pm 0.212$ | $0.352 \pm 0.045$ | $0.416 \pm 0.185$ | $4.501 \pm 7.400$ |
| ModelNet | 16 | $\mathcal{S}_\epsilon$ | $0.209 \pm 0.035$ | $0.268 \pm 0.075$ | $0.199 \pm 0.058$ | $0.206 \pm 0.082$ | $0.149 \pm 0.098$ |
| | | $\mathcal{T}_\epsilon$ | $0.448 \pm 0.029$ | $0.408 \pm 0.073$ | $0.683 \pm 0.053$ | $0.486 \pm 0.126$ | $0.150 \pm 0.096$ |
| | | $\mathcal{S}_{\epsilon,B}$ | $0.255 \pm 0.017$ | $0.384 \pm 0.088$ | $0.660 \pm 0.023$ | $0.627 \pm 0.124$ | $0.363 \pm 0.538$ |
| | | $\mathcal{T}_{\epsilon,B}$ | $0.589 \pm 0.048$ | $0.775 \pm 0.373$ | $0.793 \pm 0.019$ | $0.664 \pm 0.122$ | $0.533 \pm 0.814$ |
| | 64 | $\mathcal{S}_\epsilon$ | $0.113 \pm 0.022$ | $0.148 \pm 0.043$ | $0.182 \pm 0.052$ | $0.106 \pm 0.084$ | $0.041 \pm 0.038$ |
| | | $\mathcal{T}_\epsilon$ | $0.303 \pm 0.027$ | $0.217 \pm 0.051$ | $0.332 \pm 0.060$ | $0.274 \pm 0.124$ | $0.048 \pm 0.040$ |
| | | $\mathcal{S}_{\epsilon,B}$ | $0.322 \pm 0.019$ | $0.434 \pm 0.093$ | $0.693 \pm 0.023$ | $0.65 \pm 0.127$ | $0.379 \pm 1.459$ |
| | | $\mathcal{T}_{\epsilon,B}$ | $0.593 \pm 0.058$ | $0.703 \pm 0.229$ | $0.249 \pm 0.036$ | $0.476 \pm 0.235$ | $4.355 \pm 5.257$ |
| | 200 | $\mathcal{S}_\epsilon$ | $0.037 \pm 0.015$ | $0.047 \pm 0.029$ | $0.021 \pm 0.015$ | $0.043 \pm 0.033$ | $0.024 \pm 0.030$ |
| | | $\mathcal{T}_\epsilon$ | $0.169 \pm 0.022$ | $0.183 \pm 0.040$ | $0.466 \pm 0.046$ | $0.286 \pm 0.105$ | $0.030 \pm 0.025$ |
| | | $\mathcal{S}_{\epsilon,B}$ | $0.388 \pm 0.021$ | $0.451 \pm 0.062$ | $0.701 \pm 0.024$ | $0.608 \pm 0.118$ | $0.379 \pm 0.988$ |
| | | $\mathcal{T}_{\epsilon,B}$ | $0.494 \pm 0.047$ | $0.708 \pm 0.312$ | $0.791 \pm 0.019$ | $0.668 \pm 0.114$ | $0.512 \pm 0.833$ |
| Mixed | 16 | $\mathcal{S}_\epsilon$ | $0.179 \pm 0.030$ | $0.238 \pm 0.069$ | $0.157 \pm 0.043$ | $0.180 \pm 0.075$ | $0.150 \pm 0.093$ |
| | | $\mathcal{T}_\epsilon$ | $0.189 \pm 0.031$ | $0.245 \pm 0.071$ | $0.162 \pm 0.043$ | $0.189 \pm 0.075$ | $0.235 \pm 0.106$ |
| | | $\mathcal{S}_{\epsilon,B}$ | $0.157 \pm 0.017$ | $0.259 \pm 0.089$ | $0.207 \pm 0.029$ | $0.345 \pm 0.135$ | $1.611 \pm 2.038$ |
| | | $\mathcal{T}_{\epsilon,B}$ | $0.416 \pm 0.050$ | $0.521 \pm 0.187$ | $0.217 \pm 0.037$ | $0.477 \pm 0.224$ | $5.080 \pm 18.203$ |
| | 64 | $\mathcal{S}_\epsilon$ | $0.040 \pm 0.010$ | $0.047 \pm 0.019$ | $0.020 \pm 0.012$ | $0.042 \pm 0.025$ | $0.055 \pm 0.049$ |
| | | $\mathcal{T}_\epsilon$ | $0.044 \pm 0.010$ | $0.055 \pm 0.024$ | $0.025 \pm 0.014$ | $0.066 \pm 0.042$ | $0.117 \pm 0.068$ |
| | | $\mathcal{S}_{\epsilon,B}$ | $0.148 \pm 0.013$ | $0.183 \pm 0.062$ | $0.16 \pm 0.024$ | $0.278 \pm 0.102$ | $1.518 \pm 2.288$ |
| | | $\mathcal{T}_{\epsilon,B}$ | $0.593 \pm 0.058$ | $0.703 \pm 0.229$ | $0.249 \pm 0.036$ | $0.476 \pm 0.235$ | $4.355 \pm 5.257$ |
| | 200 | $\mathcal{S}_\epsilon$ | $0.017 \pm 0.006$ | $0.017 \pm 0.009$ | $0.004 \pm 0.005$ | $0.022 \pm 0.016$ | $0.032 \pm 0.032$ |
| | | $\mathcal{T}_\epsilon$ | $0.023 \pm 0.007$ | $0.023 \pm 0.012$ | $0.005 \pm 0.006$ | $0.035 \pm 0.029$ | $0.112 \pm 0.063$ |
| | | $\mathcal{S}_{\epsilon,B}$ | $0.154 \pm 0.012$ | $0.178 \pm 0.035$ | $0.159 \pm 0.023$ | $0.244 \pm 0.067$ | $1.631 \pm 2.385$ |
| | | $\mathcal{T}_{\epsilon,B}$ | $0.625 \pm 0.082$ | $0.726 \pm 0.219$ | $0.216 \pm 0.037$ | $0.451 \pm 0.212$ | $5.681 \pm 7.941$ |

Table 11: $\mathcal{E}_{\mathrm{dir}}$ across all 5D datasets for various $\epsilon$-kernel sizes. Note that we already report performance for the mixed synthetic data in the tables in the main text so we do not report the performance here.

| Train dataset | $\epsilon$ | Method | Uniform Ball | Ellipse | Single Gaussian | Gaussian Mixture |
|---|---|---|---|---|---|---|
| Uniform Ball | 16 | $\mathcal{S}_\varepsilon$ | $0.431 \pm 0.043$ | $0.562 \pm 0.086$ | $0.426 \pm 0.067$ | $0.524 \pm 0.102$ |
| | | $\mathcal{T}_\varepsilon$ | $0.464 \pm 0.048$ | $0.514 \pm 0.069$ | $0.676 \pm 0.038$ | $0.599 \pm 0.092$ |
| | | $\mathcal{S}_{\varepsilon,B}$ | $0.440 \pm 0.047$ | $0.742 \pm 0.248$ | $0.803 \pm 0.012$ | $0.725 \pm 0.111$ |
| | | $\mathcal{T}_{\varepsilon,B}$ | $0.352 \pm 0.044$ | $0.684 \pm 0.202$ | $0.816 \pm 0.011$ | $0.741 \pm 0.099$ |
| | 64 | $\mathcal{S}_\varepsilon$ | $0.188 \pm 0.025$ | $0.247 \pm 0.057$ | $0.159 \pm 0.033$ | $0.232 \pm 0.058$ |
| | | $\mathcal{T}_\varepsilon$ | $0.197 \pm 0.027$ | $0.366 \pm 0.066$ | $0.538 \pm 0.033$ | $0.362 \pm 0.092$ |
| | | $\mathcal{S}_{\varepsilon,B}$ | $0.242 \pm 0.027$ | $0.621 \pm 0.177$ | $0.796 \pm 0.011$ | $0.695 \pm 0.133$ |
| | | $\mathcal{T}_{\varepsilon,B}$ | $0.248 \pm 0.033$ | $0.633 \pm 0.199$ | $0.791 \pm 0.011$ | $0.702 \pm 0.113$ |
| | 200 | $\mathcal{S}_\varepsilon$ | $0.084 \pm 0.016$ | $0.112 \pm 0.038$ | $0.058 \pm 0.023$ | $0.103 \pm 0.038$ |
| | | $\mathcal{T}_\varepsilon$ | $0.118 \pm 0.023$ | $0.202 \pm 0.052$ | $0.369 \pm 0.047$ | $0.356 \pm 0.113$ |
| | | $\mathcal{S}_{\varepsilon,B}$ | $0.238 \pm 0.03$ | $0.63 \pm 0.198$ | $0.79 \pm 0.011$ | $0.669 \pm 0.105$ |
| | | $\mathcal{T}_{\varepsilon,B}$ | $0.23 \pm 0.029$ | $0.625 \pm 0.193$ | $0.786 \pm 0.012$ | $0.704 \pm 0.118$ |
| Ellipse | 16 | $\mathcal{S}_\varepsilon$ | $0.547 \pm 0.066$ | $0.549 \pm 0.072$ | $0.554 \pm 0.089$ | $0.609 \pm 0.107$ |
| | | $\mathcal{T}_\varepsilon$ | $0.534 \pm 0.066$ | $0.534 \pm 0.072$ | $0.577 \pm 0.054$ | $0.553 \pm 0.102$ |
| | | $\mathcal{S}_{\varepsilon,B}$ | $0.351 \pm 0.032$ | $0.394 \pm 0.070$ | $0.844 \pm 0.017$ | $1.035 \pm 0.295$ |
| | | $\mathcal{T}_{\varepsilon,B}$ | $0.370 \pm 0.047$ | $0.514 \pm 0.158$ | $0.758 \pm 0.029$ | $0.872 \pm 0.216$ |
| | 64 | $\mathcal{S}_\varepsilon$ | $0.193 \pm 0.024$ | $0.213 \pm 0.043$ | $0.168 \pm 0.033$ | $0.187 \pm 0.048$ |
| | | $\mathcal{T}_\varepsilon$ | $0.313 \pm 0.051$ | $0.302 \pm 0.063$ | $0.543 \pm 0.034$ | $0.417 \pm 0.095$ |
| | | $\mathcal{S}_{\varepsilon,B}$ | $0.356 \pm 0.043$ | $0.498 \pm 0.125$ | $0.894 \pm 0.011$ | $0.825 \pm 0.157$ |
| | | $\mathcal{T}_{\varepsilon,B}$ | $0.351 \pm 0.054$ | $0.548 \pm 0.174$ | $0.702 \pm 0.026$ | $0.902 \pm 0.280$ |
| | 200 | $\mathcal{S}_\varepsilon$ | $0.091 \pm 0.017$ | $0.085 \pm 0.024$ | $0.069 \pm 0.017$ | $0.094 \pm 0.031$ |
| | | $\mathcal{T}_\varepsilon$ | $0.092 \pm 0.018$ | $0.084 \pm 0.024$ | $0.369 \pm 0.042$ | $0.239 \pm 0.091$ |
| | | $\mathcal{S}_{\varepsilon,B}$ | $0.257 \pm 0.031$ | $0.399 \pm 0.108$ | $0.695 \pm 0.02$ | $0.814 \pm 0.161$ |
| | | $\mathcal{T}_{\varepsilon,B}$ | $0.35 \pm 0.043$ | $0.343 \pm 0.096$ | $0.721 \pm 0.024$ | $0.77 \pm 0.188$ |
| Gaussian | 16 | $\mathcal{S}_\varepsilon$ | $0.488 \pm 0.066$ | $0.529 \pm 0.072$ | $0.448 \pm 0.075$ | $0.565 \pm 0.142$ |
| | | $\mathcal{T}_\varepsilon$ | $0.728 \pm 0.061$ | $0.722 \pm 0.068$ | $0.467 \pm 0.073$ | $0.631 \pm 0.095$ |
| | | $\mathcal{S}_{\varepsilon,B}$ | $3.627 \pm 0.169$ | $4.249 \pm 1.076$ | $0.436 \pm 0.064$ | $2.446 \pm 0.675$ |
| | | $\mathcal{T}_{\varepsilon,B}$ | $3.601 \pm 0.171$ | $4.290 \pm 1.114$ | $0.410 \pm 0.052$ | $3.043 \pm 0.677$ |
| | 64 | $\mathcal{S}_\varepsilon$ | $0.204 \pm 0.027$ | $0.250 \pm 0.050$ | $0.163 \pm 0.033$ | $0.196 \pm 0.049$ |
| | | $\mathcal{T}_\varepsilon$ | $0.471 \pm 0.055$ | $0.452 \pm 0.064$ | $0.205 \pm 0.045$ | $0.411 \pm 0.099$ |
| | | $\mathcal{S}_{\varepsilon,B}$ | $3.122 \pm 0.109$ | $4.051 \pm 1.045$ | $0.291 \pm 0.035$ | $2.106 \pm 0.666$ |
| | | $\mathcal{T}_{\varepsilon,B}$ | $3.434 \pm 0.127$ | $4.332 \pm 1.130$ | $0.354 \pm 0.050$ | $3.225 \pm 0.734$ |
| | 200 | $\mathcal{S}_\varepsilon$ | $0.095 \pm 0.017$ | $0.116 \pm 0.038$ | $0.065 \pm 0.019$ | $0.102 \pm 0.032$ |
| | | $\mathcal{T}_\varepsilon$ | $0.177 \pm 0.026$ | $0.170 \pm 0.041$ | $0.065 \pm 0.022$ | $0.165 \pm 0.056$ |
| | | $\mathcal{S}_{\varepsilon,B}$ | $3.46 \pm 0.12$ | $4.421 \pm 1.108$ | $0.352 \pm 0.048$ | $2.069 \pm 0.718$ |
| | | $\mathcal{T}_{\varepsilon,B}$ | $3.297 \pm 0.112$ | $4.206 \pm 1.096$ | $0.333 \pm 0.044$ | $3.018 \pm 0.59$ |
| Gaussian Mixture | 16 | $\mathcal{S}_\varepsilon$ | $0.546 \pm 0.066$ | $0.539 \pm 0.074$ | $0.460 \pm 0.079$ | $0.501 \pm 0.098$ |
| | | $\mathcal{T}_\varepsilon$ | $0.622 \pm 0.060$ | $0.571 \pm 0.071$ | $0.516 \pm 0.081$ | $0.504 \pm 0.098$ |
| | | $\mathcal{S}_{\varepsilon,B}$ | $0.342 \pm 0.030$ | $0.413 \pm 0.078$ | $0.438 \pm 0.039$ | $0.573 \pm 0.194$ |
| | | $\mathcal{T}_{\varepsilon,B}$ | $1.097 \pm 0.141$ | $1.314 \pm 0.329$ | $0.434 \pm 0.040$ | $1.002 \pm 0.433$ |
| | 64 | $\mathcal{S}_\varepsilon$ | $0.193 \pm 0.025$ | $0.213 \pm 0.041$ | $0.164 \pm 0.032$ | $0.170 \pm 0.045$ |
| | | $\mathcal{T}_\varepsilon$ | $0.197 \pm 0.025$ | $0.205 \pm 0.039$ | $0.182 \pm 0.036$ | $0.174 \pm 0.046$ |
| | | $\mathcal{S}_{\varepsilon,B}$ | $0.427 \pm 0.025$ | $0.409 \pm 0.044$ | $0.465 \pm 0.035$ | $0.444 \pm 0.113$ |
| | | $\mathcal{T}_{\varepsilon,B}$ | $1.731 \pm 0.143$ | $1.925 \pm 0.391$ | $0.447 \pm 0.035$ | $0.860 \pm 0.395$ |
| | 200 | $\mathcal{S}_\varepsilon$ | $0.093 \pm 0.017$ | $0.103 \pm 0.030$ | $0.073 \pm 0.019$ | $0.084 \pm 0.028$ |
| | | $\mathcal{T}_\varepsilon$ | $0.097 \pm 0.018$ | $0.093 \pm 0.025$ | $0.084 \pm 0.025$ | $0.091 \pm 0.032$ |
| | | $\mathcal{S}_{\varepsilon,B}$ | $0.487 \pm 0.045$ | $0.541 \pm 0.121$ | $0.366 \pm 0.046$ | $0.717 \pm 0.207$ |
| | | $\mathcal{T}_{\varepsilon,B}$ | $2.154 \pm 0.122$ | $1.953 \pm 0.287$ | $0.356 \pm 0.042$ | $0.757 \pm 0.278$ |
| Mixed | 16 | $\mathcal{S}_\varepsilon$ | $0.562 \pm 0.065$ | $0.544 \pm 0.073$ | $0.485 \pm 0.085$ | $0.512 \pm 0.101$ |
| | | $\mathcal{T}_\varepsilon$ | $0.602 \pm 0.063$ | $0.559 \pm 0.071$ | $0.477 \pm 0.078$ | $0.524 \pm 0.095$ |
| | | $\mathcal{S}_{\varepsilon,B}$ | $0.349 \pm 0.031$ | $0.412 \pm 0.097$ | $0.360 \pm 0.037$ | $0.675 \pm 0.217$ |
| | | $\mathcal{T}_{\varepsilon,B}$ | $1.883 \pm 0.188$ | $2.458 \pm 0.684$ | $0.391 \pm 0.049$ | $1.525 \pm 0.572$ |
| | 64 | $\mathcal{S}_\varepsilon$ | $0.194 \pm 0.026$ | $0.220 \pm 0.045$ | $0.162 \pm 0.032$ | $0.177 \pm 0.047$ |
| | | $\mathcal{T}_\varepsilon$ | $0.202 \pm 0.025$ | $0.228 \pm 0.046$ | $0.168 \pm 0.034$ | $0.188 \pm 0.051$ |
| | | $\mathcal{S}_{\varepsilon,B}$ | $0.265 \pm 0.032$ | $0.351 \pm 0.087$ | $0.285 \pm 0.04$ | $0.592 \pm 0.224$ |
| | | $\mathcal{T}_{\varepsilon,B}$ | $0.977 \pm 0.102$ | $1.632 \pm 0.448$ | $0.336 \pm 0.042$ | $1.276 \pm 0.455$ |
| | 200 | $\mathcal{S}_\varepsilon$ | $0.085 \pm 0.017$ | $0.076 \pm 0.022$ | $0.060 \pm 0.020$ | $0.079 \pm 0.031$ |
| | | $\mathcal{T}_\varepsilon$ | $0.094 \pm 0.018$ | $0.094 \pm 0.028$ | $0.064 \pm 0.020$ | $0.096 \pm 0.047$ |
| | | $\mathcal{S}_{\varepsilon,B}$ | $0.195 \pm 0.017$ | $0.294 \pm 0.079$ | $0.257 \pm 0.034$ | $0.551 \pm 0.216$ |
| | | $\mathcal{T}_{\varepsilon,B}$ | $3.092 \pm 0.164$ | $2.999 \pm 0.699$ | $0.314 \pm 0.041$ | $1.575 \pm 0.424$ |

Table 12: Results for minimum enclosing ball (2D). We report the relative error for the predicted radius (w.r.t to ground truth radius) ($\mathcal{E}_\mathrm{r}$) and percentage of points excluded from the predicted enclosing ball ($\mathcal{E}_\mathrm{p}$). All processors are configured to output a coarsened set of 16 points and frozen processors are trained on mixed synthetic data. Boldface indicates the best performing method while underlines denote best performing *neural* method.

| Train Set | Architecture | $\mathcal{E}_\mathbf{r}$ | $\mathcal{E}_\mathbf{p}(\%)$ |
|---|---|---|---|
| Uniform Disk | $\mathcal{S}_\mathrm{extent}$ (Frozen) | $\underline{0.009 \pm 0.008}$ | $1.9 \pm 1.3$ |
| | $\mathcal{S}_\mathrm{extent}$ (E2E) | $0.023 \pm 0.019$ | $2.3 \pm 1.9$ |
| | $\mathcal{T}_\mathrm{extent}$ (Frozen) | $0.099 \pm 0.050$ | $\underline{0.1 \pm 0.3}$ |
| | $\mathcal{T}_\mathrm{extent}$ (E2E) | $0.024 \pm 0.020$ | $1.4 \pm 1.8$ |
| | $\mathcal{S}_\mathrm{Baseline}$ (E2E) | $0.048 \pm 0.028$ | $0.5 \pm 1.0$ |
| | $\mathcal{T}_\mathrm{Baseline}$ (E2E) | $0.030 \pm 0.024$ | $0.3 \pm 3.0$ |
| | $\mathcal{S}_\mathrm{Direct}$ | $0.012 \pm 0.006$ | $3.2 \pm 1.3$ |
| | $\mathcal{T}_\mathrm{Direct}$ | $0.022 \pm 0.018$ | $7.1 \pm 3.8$ |
| | $\varepsilon$-kernel + Exact | $\mathbf{0.000002 \pm 0.00009}$ | $\mathbf{0 \pm 0}$ |
| SQUID | $\mathcal{S}_\mathrm{extent}$ (Frozen) | $0.020 \pm 0.200$ | $2.0 \pm 3.0$ |
| | $\mathcal{S}_\mathrm{extent}$ (E2E) | $\underline{0.010 \pm 0.010}$ | $2.0 \pm 3.0$ |
| | $\mathcal{T}_\mathrm{extent}$ (Frozen) | $0.068 \pm 0.045$ | $\underline{1.6 \pm 3.2}$ |
| | $\mathcal{T}_\mathrm{extent}$ (E2E) | $0.190 \pm 0.090$ | $4.0 \pm 5.0$ |
| | $\mathcal{S}_\mathrm{Baseline}$ (E2E) | $0.027 \pm 0.025$ | $2.8 \pm 2.5$ |
| | $\mathcal{T}_\mathrm{Baseline}$ (E2E) | $0.039 \pm 0.034$ | $4.5 \pm 3.8$ |
| | $\mathcal{S}_\mathrm{Direct}$ | $0.058 \pm 0.046$ | $3.3 \pm 4.4$ |
| | $\mathcal{T}_\mathrm{Direct}$ | $0.685 \pm 0.065$ | $83.3 \pm 10.7$ |
| | $\varepsilon$-kernel + Exact | $\mathbf{0.00004 \pm 0.0006}$ | $\mathbf{0 \pm 0}$ |

Table 13: Results for minimum enclosing ball (3D). We report the relative error for the predicted radius ($\mathcal{E}_\mathrm{r}$) and the percentage of points which excluded from the estimated ball ($\mathcal{E}_\mathrm{p}(\%)$). All frozen processors are trained on mixed synthetic data. Boldface indicates the best performing method while underlines denote best performing *neural* method.

| Train Set | Architecture | $\mathcal{E}_\mathbf{r}$ | $\mathcal{E}_\mathbf{p}(\%)$ |
|---|---|---|---|
| Uniform Ball | $\mathcal{S}_\mathrm{extent}$ (Frozen) | $0.030 \pm 0.020$ | $\underline{0.7 \pm 1.0}$ |
| | $\mathcal{S}_\mathrm{extent}$ (E2E) | $0.035 \pm 0.028$ | $2.8 \pm 3.4$ |
| | $\mathcal{T}_\mathrm{extent}$ (Frozen) | $0.020 \pm 0.018$ | $4.6 \pm 3.8$ |
| | $\mathcal{T}_\mathrm{extent}$ (E2E) | $0.076 \pm 0.051$ | $2.0 \pm 4.0$ |
| | $\mathcal{S}_\mathrm{Baseline}$ (E2E) | $0.056 \pm 0.031$ | $1.8 \pm 2.5$ |
| | $\mathcal{T}_\mathrm{Baseline}$ (E2E) | $0.035 \pm 0.032$ | $8.6 \pm 6.5$ |
| | $\mathcal{S}_\mathrm{Direct}$ | $\underline{0.014 \pm 0.011}$ | $1.6 \pm 1.5$ |
| | $\mathcal{T}_\mathrm{Direct}$ | $0.028 \pm 0.017$ | $1.7 \pm 5.9$ |
| | $\varepsilon$-kernel + Exact | $\mathbf{0.001 \pm 0.003}$ | $\mathbf{0 \pm 0}$ |
| ModelNet | $\mathcal{S}_\mathrm{extent}$ (Frozen) | $0.100 \pm 0.070$ | $\underline{0.2 \pm 0.7}$ |
| | $\mathcal{S}_\mathrm{extent}$ (E2E) | $0.060 \pm 0.050$ | $0.2 \pm 0.9$ |
| | $\mathcal{T}_\mathrm{extent}$ (Frozen) | $0.077 \pm 0.072$ | $0.6 \pm 1.3$ |
| | $\mathcal{T}_\mathrm{extent}$ (E2E) | $\underline{0.029 \pm 0.032}$ | $3.3 \pm 4.4$ |
| | $\mathcal{S}_\mathrm{Baseline}$ (E2E) | $0.053 \pm 0.058$ | $1.6 \pm 2.5$ |
| | $\mathcal{T}_\mathrm{Baseline}$ (E2E) | $0.858 \pm 0.289$ | $57.3 \pm 39.6$ |
| | $\mathcal{S}_\mathrm{Direct}$ | $0.214 \pm 0.179$ | $8.1 \pm 9.7$ |
| | $\mathcal{T}_\mathrm{Direct}$ | $0.033 \pm 0.02$ | $3.6 \pm 3.8$ |
| | $\varepsilon$-kernel + Exact | $\mathbf{0.0005 \pm 0.0023}$ | $\mathbf{0.27 \pm 2.67}$ |

Table 14: Minimum enclosing ellipse error for point clouds in $\mathbb{R}^2$ on in-distribution test data, comparing frozen and end-to-end (E2E) training procedures. All frozen processors are trained on mixed synthetic data. We report the relative error in major ($\mathcal{E}_{\mathrm{r,maj}}$) and minor radii ($\mathcal{E}_{\mathrm{r,min}}$) and the percentage of points excluded ($\mathcal{E}_{\mathrm{p}}(\%)$). Boldface indicates the best performing method while underlines denote best performing *neural* method.

| Train Set | Architecture | $\mathcal{E}_{\mathrm{r,min}}$ | $\mathcal{E}_{\mathrm{r,maj}}$ | $\mathcal{E}_{\mathrm{p}}$ (%) |
|---|---|---|---|---|
| Synthetic Ellipses | $\mathcal{S}_{\mathrm{extent}}$ (Frozen) | $0.037 \pm 0.037$ | $\underline{0.022 \pm 0.018}$ | $\underline{5.1 \pm 5.2}$ |
| | $\mathcal{S}_{\mathrm{extent}}$ (E2E) | $0.056 \pm 0.047$ | $0.047 \pm 0.035$ | $9.1 \pm 5.4$ |
| | $\mathcal{T}_{\mathrm{extent}}$ (Frozen) | $0.041 \pm 0.040$ | $0.027 \pm 0.020$ | $7.5 \pm 5.4$ |
| | $\mathcal{T}_{\mathrm{extent}}$ (E2E) | $0.038 \pm 0.033$ | $0.022 \pm 0.019$ | $6.5 \pm 5.3$ |
| | $\mathcal{S}_{\mathrm{Baseline}}$ (E2E) | $0.378 \pm 0.234$ | $0.426 \pm 0.387$ | $27.7 \pm 25.1$ |
| | $\mathcal{T}_{\mathrm{Baseline}}$ (E2E) | $0.071 \pm 0.660$ | $0.047 \pm 0.038$ | $8.8 \pm 7.2$ |
| | $\mathcal{S}_{\mathrm{Direct}}$ | $\underline{0.033 \pm 0.025}$ | $0.039 \pm 0.022$ | $9.8 \pm 4.4$ |
| | $\mathcal{T}_{\mathrm{Direct}}$ | $0.035 \pm 0.03$ | $0.025 \pm 0.019$ | $6.6 \pm 4.9$ |
| | $\varepsilon$-kernel + Exact | $\mathbf{0.0004 \pm 0.003}$ | $\mathbf{0.0001 \pm 0.0009}$ | $\mathbf{3.3 \pm 1.3}$ |
| SQUID | $\mathcal{S}_{\mathrm{extent}}$ (Frozen) | $\underline{0.056 \pm 0.047}$ | $0.047 \pm 0.035$ | $6.6 \pm 5.4$ |
| | $\mathcal{S}_{\mathrm{extent}}$ (E2E) | $0.078 \pm 0.065$ | $0.050 \pm 0.038$ | $13.9 \pm 7.2$ |
| | $\mathcal{T}_{\mathrm{extent}}$ (Frozen) | $0.058 \pm 0.051$ | $\underline{0.032 \pm 0.026}$ | $12.5 \pm 6.6$ |
| | $\mathcal{T}_{\mathrm{extent}}$ (E2E) | $0.093 \pm 0.074$ | $0.045 \pm 0.034$ | $14.4 \pm 7.7$ |
| | $\mathcal{S}_{\mathrm{Baseline}}$ (E2E) | $0.322 \pm 0.365$ | $0.250 \pm 0.262$ | $32.0 \pm 16.6$ |
| | $\mathcal{T}_{\mathrm{Baseline}}$ (E2E) | $0.357 \pm 0.550$ | $0.049 \pm 0.041$ | $16.8 \pm 18.4$ |
| | $\mathcal{S}_{\mathrm{Direct}}$ | $0.684 \pm 0.729$ | $0.049 \pm 0.039$ | $0 \pm 0$ |
| | $\mathcal{T}_{\mathrm{Direct}}$ | $1.483 \pm 1.287$ | $0.128 \pm 0.064$ | $1.8 \pm 0.7$ |
| | $\varepsilon$-kernel + Exact | $\mathbf{0.004 \pm 0.028}$ | $\mathbf{0.002 \pm 0.014}$ | $\mathbf{7.7 \pm 5.34}$ |

Table 15: Minimum enclosing annulus error on in-distribution test data, comparing frozen and end-to-end (E2E) training procedures. All frozen processors are trained on mixed synthetic data. We report the relative error in the width of the annuli and the proportion of points excluded

| Train Set | Architecture | $\mathcal{E}_{\mathrm{w}}$ | $\mathcal{E}_{\mathrm{p}}$ (%) |
|---|---|---|---|
| Synthetic Annuli | $\mathcal{S}_{\mathrm{extent}}$ (Frozen) | $0.050 \pm 0.070$ | $\mathbf{4.26 \pm 4.94}$ |
| | $\mathcal{S}_{\mathrm{extent}}$ (E2E) | $\underline{0.038 \pm 0.042}$ | $5.55 \pm 4.64$ |
| | $\mathcal{T}_{\mathrm{extent}}$ (Frozen) | $0.077 \pm 0.104$ | $11.39 \pm 8.02$ |
| | $\mathcal{T}_{\mathrm{extent}}$ (E2E) | $0.083 \pm 0.087$ | $12.53 \pm 4.64$ |
| | $\mathcal{S}_{\mathrm{Baseline}}$ | $0.128 \pm 0.263$ | $5.27 \pm 5.27$ |
| | $\mathcal{T}_{\mathrm{Baseline}}$ | $0.417 \pm 0.574$ | $4.39 \pm 8.22$ |
| | $\mathcal{S}_{\mathrm{Direct}}$ | $0.141 \pm 0.117$ | $4.99 \pm 6.95$ |
| | $\mathcal{T}_{\mathrm{Direct}}$ | $0.076 \pm 0.120$ | $4.43 \pm 6.25$ |
| | $\varepsilon$-kernel + Exact | $\mathbf{0.022 \pm 0.07}$ | $4.32 \pm 5.9$ |
| SQUID | $\mathcal{S}_{\mathrm{extent}}$ (Frozen) | $0.073 \pm 0.110$ | $\underline{6.31 \pm 4.15}$ |
| | $\mathcal{S}_{\mathrm{extent}}$ (E2E) | $0.087 \pm 0.103$ | $8.42 \pm 4.64$ |
| | $\mathcal{T}_{\mathrm{extent}}$ (Frozen) | $0.064 \pm 0.084$ | $9.60 \pm 4.75$ |
| | $\mathcal{T}_{\mathrm{extent}}$ (E2E) | $0.104 \pm 0.133$ | $6.43 \pm 4.21$ |
| | $\mathcal{S}_{\mathrm{Baseline}}$ (E2E) | $0.191 \pm 0.219$ | $7.23 \pm 5.04$ |
| | $\mathcal{T}_{\mathrm{Baseline}}$ (E2E) | $0.118 \pm 0.120$ | $11.9 \pm 6.86$ |
| | $\mathcal{S}_{\mathrm{Direct}}$ | $\mathbf{0.041 \pm 0.079}$ | $9.87 \pm 6.05$ |
| | $\mathcal{T}_{\mathrm{Direct}}$ | $\underline{0.065 \pm 0.102}$ | $10.2 \pm 4.66$ |
| | $\varepsilon$-kernel + Exact | $0.111 \pm 0.095$ | $\mathbf{0.213 \pm 0.122}$ |

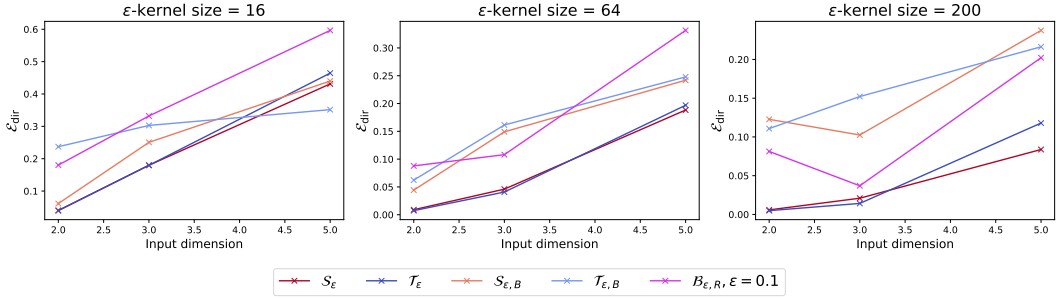

Figure 7: Directional width error ($\mathcal{E}_{\mathrm{dir}}$) vs. the input dimension of the point cloud. We train each model on 'Uniform Ball' datasets in $\mathbb{R}^2$, $\mathbb{R}^3$ and $\mathbb{R}^5$ and test on in-distribution dataset. We notice that all models, as well as the baseline algorithm have increasing directional width error as the input point cloud dimension increases.

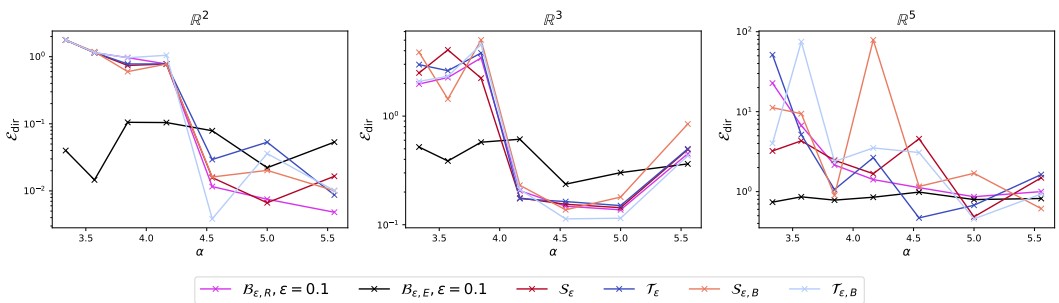

Figure 8: Directional width error ($\mathcal{E}_{\mathrm{dir}}$) vs $\alpha$-fatness on relaxed-$\varepsilon$-kernel approximation task for $\mathbb{R}^2$ and $\mathbb{R}^3$. We vary the $\alpha$-fatness of the input data by varying the scale of the minor axes of ellipsoids in $\mathbb{R}^2$ and $\mathbb{R}^3$ and then sampling point clouds from such ellipsoids. Each input point cloud has 100 points and the output $\varepsilon$-kernels are fixed at 64.

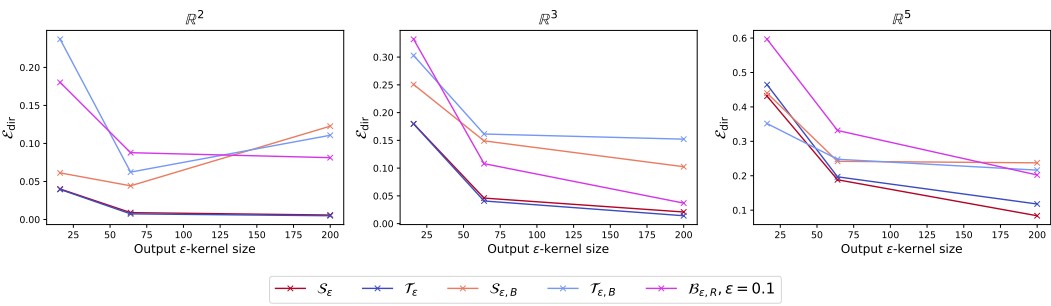

Figure 9: Direction width error $\mathcal{E}_{\mathrm{dir}}$ vs output size of relaxed-$\varepsilon$-kernel size across dimensions. All models are trained and evaluated on point clouds sampled from uniform balls/disks with 500 points per set. Notice that, similar to the baseline algorithm, each model will have better results as we increase output point set size (with SumFormer implementation of $\mathcal{N}_{\phi_\varepsilon}$ still outperforming Transformer).

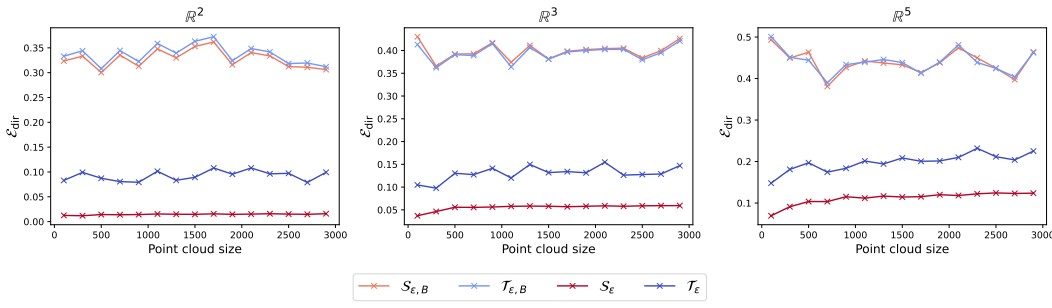

Figure 10: Directional width error $\mathcal{E}_{\text{dir}}$ vs. input point cloud size. All models are trained on the uniform ball/disk dataset and evaluated on uniform ellipses/ellipsoids of varying sizes.

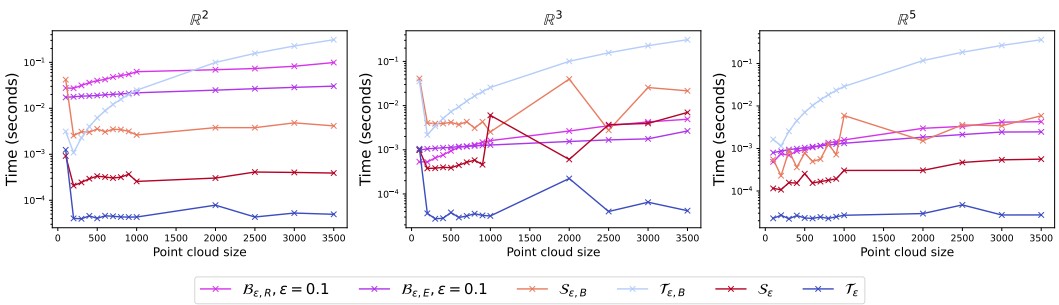

Figure 11: Comparison of inference time for each model along with the runtime of the exact and approximate algorithms for coreset tasks.

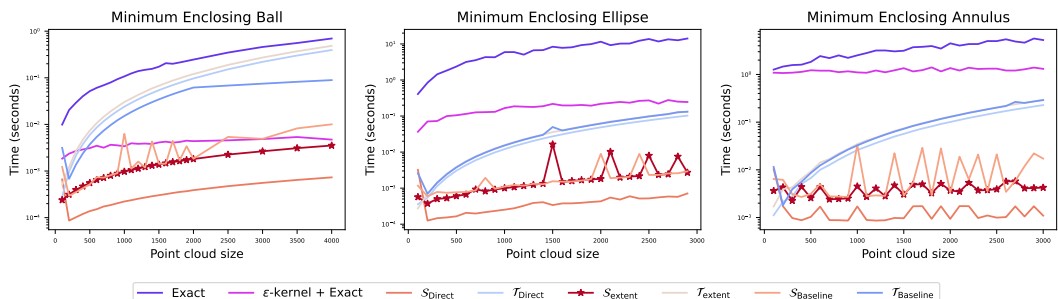

Figure 12: Comparison of inference time for each model along with runtimes of the exact and approximate algorithms for extent measure tasks. Note that our model, $\mathcal{S}_{\text{extent}}$, is significantly faster than all classical algorithmic approaches.

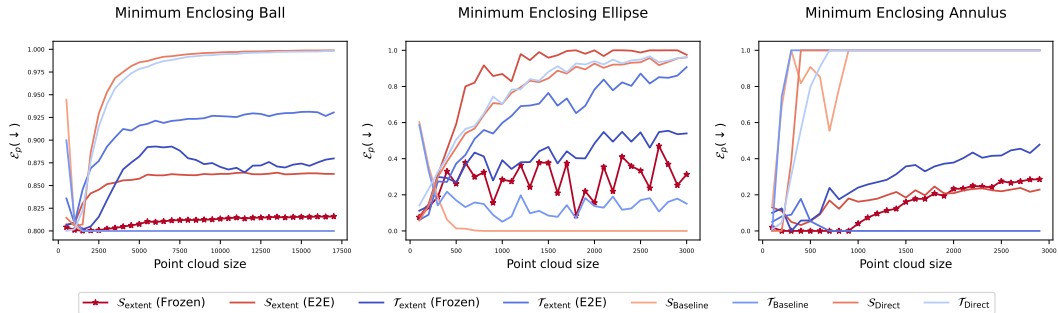

Figure 13: Size generalization across all extent measure tasks. We record the proportion of points missed by the predicted shapes produced by each model on the extent-measure tasks as both the size and bounding box of the input point clouds increase.

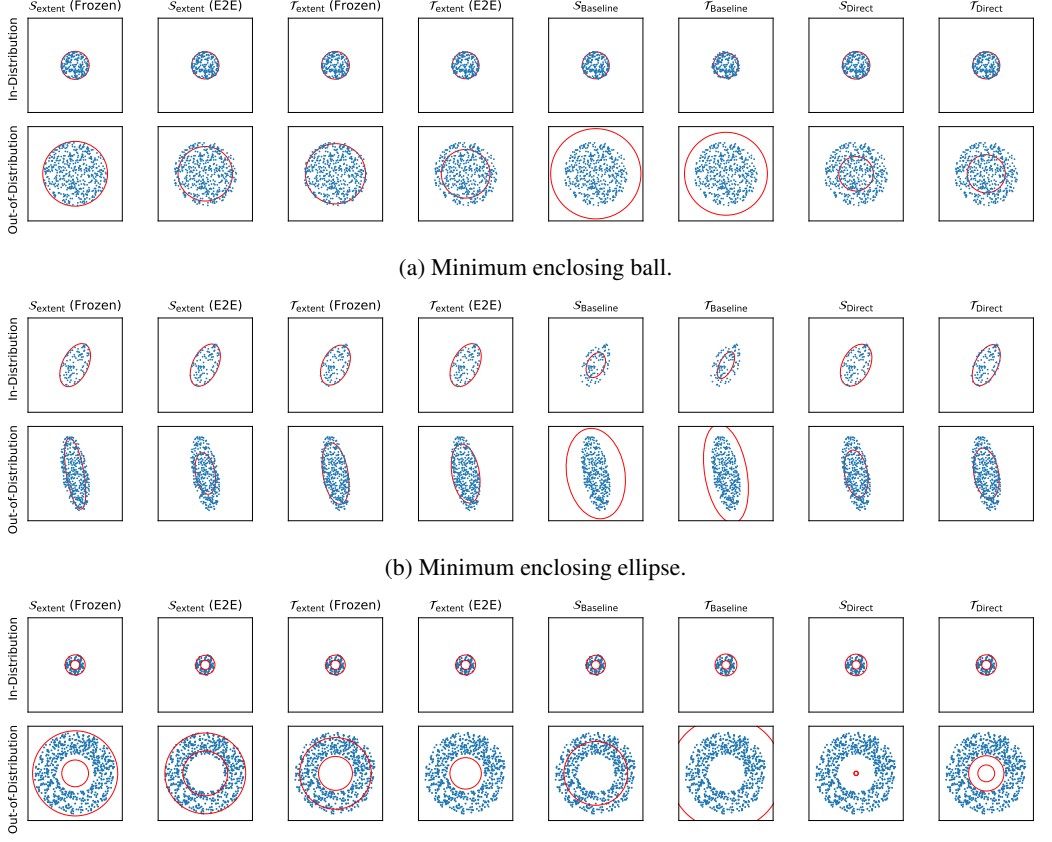

(a) Minimum enclosing ball.

(b) Minimum enclosing ellipse.

(c) Minimum enclosing annulus.

Figure 14: Visualizations for each extent measure problem. Notice that our frozen model performs much better out-of-distribution than other comparable neural models.

