# OpenReview forum: "Effective Neural Approximations for Geometric Optimization Problems"
_NeurIPS.cc/2025/Conference — NeurIPS 2025 poster_

### Official Review · Reviewer_r5xy · 2025-06-23

**Clarity:** 4
**Significance:** 3
**Originality:** 3
**Rating:** 5
**Confidence:** 2

**Summary:**

This paper introduces a novel approach for approximating "extent measures" in computational geometry with neural networks. It develops a "relaxed $\epsilon$ theory" which can be implemented using a neural network (SumFormer) with a bound on the number of parameters. The framework encodes input points into a latent space where they are processed using a SumFormer-based model. The model size is only dependent on the input point dimension, not on the number of points in the set, and a bound on the model parameters, depending on a user-specified approximation error, is derived.

This latent space model is then tested on both synthetic data for validation and real datasets (SQUID [a fish boundary contour dataset] and ModelNet). The experiments test multiple latent processing models and training schemes (transformer, sumformer, end-to-end, frozen processing model) with mixed results and performance on both in and out of distribution points. The appendix provides further results on varying sampling schemas and set sizes.

**Questions:**

I'm not an expert on point cloud geometry. Reading your paper, I have the following questions:

- How strong is the assumption of "fat" point clouds in practice? Are there important cases where this assumption doesn't hold?

- Can you provide a comparison to a classical baseline (algorithms using $\epsilon$-kernels$?), both in terms of performance and compute requirements/scaling?

- Are there any deep learning baselines to compare to?

- The SQUID dataset seems a bit odd (fish boundary contours?). Can this method also be applied to noisy real-world data, such as lidar scans?

**Ethical Concerns:**

["NO or VERY MINOR ethics concerns only"]

**Final Justification:**

I think a novel application with strong theoretical contributions should be included at NeurIPS. Yeah they have high errors, but e.g. PDE-surrogates also were not competitve with traditional PDE solvers in the first paper. Geometry processing is always in need of more speed, so for me the motivation is clear (I disagree with reviewer VDgZ here).

Main problem with the initial submission was insufficient comparison to classical baselines, but this was provided in the rebuttal.

**Limitations:**

yes

**Quality:**

3

**Strengths And Weaknesses:**

## Strengths

- Good structure

- Clear theory development and implementation thereof. Theory from the initial definition to theorems can be followed by someone not familiar with the topic.

- Robost framework, which can handle point clouds of varying sizes

- The latent space processing model can be retrained on new data/task by simply retraining the encoder/decoder.

## Weaknesses

- Derivations require "fatness" of the point cloud, not clear how much of a restriction this is in practice.

- Missing Baselines: No deep-learning baselines and "none-neural" baselines.

---

> ### Author Rebuttal · Authors · 2025-07-31
>
> Thank you for your comments! We are glad you appreciate the robustness of our framework -- especially with regards to its out-of-distribution performance. We note that the performance of our algorithmic processor module on the $\varepsilon$-kernel task is shows much better performance OOD. Furthermore, using this $\varepsilon$-kernel processor in the encoder-decoder-processor on more complex geometric optimization problem allows for accurate approximations even with only re-training the encoder/decoder.
>
> **Regarding Q1, about the fatness of the point set:** First, we note that $\alpha$-fatness is a standard assumption in computational geometry. Intuitively, a larger $\alpha$ indicates that the point cloud is not excessively “skinny,” while smaller values of $\alpha$ correspond to increasingly elongated or degenerate point sets. In the extreme case, $\alpha$ can shrink inversely with the number of points in the input—this occurs when the point cloud is effectively degenerate, exhibiting no variation in at least one direction.
> Our theoretical guarantees hold even for point sets with low $\alpha$ values; the approximation factor in our analysis scales with $\alpha$. Consequently, as $\alpha$ decreases (i.e., the point sets become “skinnier”), the theoretical approximation factor worsens, and more parameters are required. Despite this, we observe strong empirical performance across a wide range of $\alpha$ values (cf. Figure 6 in the appendix). Additionally we find that real-world datasets tend to have relatively large $\alpha$ values. For example, in the widely used ModelNet dataset, where each point cloud lies within the bounding box $[-1, 1]^d$, the average $\alpha$-fatness is 0.81 (with the maximum possible value being 1).
>
>
> **Regarding Q2, baselines:** Thank you for your suggestion regarding comparing to more neural and algorithmic baselines. We will reiterate our response to Reviewer VDgZ.
> To the best of our knowledge, ours is the first work considering using neural networks as solvers for geometric extent measure problems. Therefore, to understand the power of leveraging algorithmic frameworks in our $\varepsilon$-kernel approximating NN, we compare with natural neural baselines. In our paper, we propose baselines built on the well-known encoder-processor-decoder paradigm (see [1]). We use a simple embedding MLP as the encoder network, the processor network is some suitable architecture (e.g. a transformer or sumformer) which processes embedded point cloud as input and aggregates the resulting representations into a global feature vector, and the a final MLP as the decoder network.
> The Sumformer and transformer versions of this baseline, $\mathcal{S}\_{\mathrm{Baseline}}$ and $\mathcal{T}\_\mathrm{Baseline}$ respectively, are exactly the neural baselines used for comparison in our paper.
>
> To give a case study as to why this is the neural baseline we compare against, we consider the minimum enclosing ball problem. The most straightforward neural approach would be to directly aggregate the point cloud input into a global feature vector using some architecture (sumformer or transformer) and then directly compute a final estimate for the radius of the minimum enclosing ball with a final MLP. We will refer to the Sumformer versions of this baseline as $\mathcal{S}_\mathrm{Direct}$. Our included encode-process-decode baseline is a slightly more complicated version of this as we do an initial encoding step with the encoder network. We show in Tables 2 and 13 in our paper that our encode-process-decode extent measure model using the $\varepsilon$-kernel-approximating Sumformer network as the processor network outperforms both encode-process-decode baselines ($\mathcal{S}\_\mathrm{Baseline}$ and $\mathcal{T}\_\mathrm{Baseline}$) on the minimum enclosing ball problem.
> To further illustrate the utility of our algorithmically aligned extent measure model $\mathcal{S}\_\mathrm{extent}$, we include additional results comparing our original baseline $\mathcal{S}\_\mathrm{Baseline}$ against $\mathcal{S}\_\mathrm{Direct}$ on the minimum enclosing ball problem for the SQUID dataset in the table below (Table A).
> | **Model** | $\mathcal{E}\_{r}$ | $\mathcal{E}\_\mathrm{p}$ |
> |-|-|-|
> | $\mathcal{S}\_\mathrm{Direct}$ | 0.986 ± 0.004  | 1.000 ± 0.000  |
> | $\mathcal{S}\_\mathrm{extent}$ (frozen) | **0.065 ± 0.055** | **0.052 ± 0.056**  |
> |$\mathcal{S}\_\mathrm{extent}$ (E2E) | 0.428 ± 0.057 | 0.596 ± 0.106 |
> | $\mathcal{S}\_\mathrm{Baseline}$  | 0.687 ± 0.080| 0.881 ± 0.089|
>
> Additionally, we thank the reviewer for the suggestion of comparing against additional baselines. In Table B below, we provide a further comparison of our extent measure approximating model, $\mathcal{S}\_\mathrm{extent}$, against the classical algorithms for the minimum enclosing ball task with different input point cloud sizes. In particular, we compare our neural model's runtime at inference time with Welzl's algorithm. Since Welzl's algorithm is an exact algorithm, we include a comparison to the classical exact $\varepsilon$-kernel algorithm followed by Welzl's algorithm. While our network is less accurate than using a $\varepsilon$-kernel followed by Welzl's algorithm, we observe that our network is significantly faster than all baseline algorithms and more accurate than neural baselines.
>
> **Table B**. Comparison of extent measure model against algorithmic baselines
> | **Method** | **200**| | **600**| |
> |-|-|-|-|-|
> |   | Runtime | Error| Runtime| Error|
> | $\mathcal{S}\_\mathrm{extent}$ (frozen) | **0.0004** | 0.020    | **0.0007**     | 0.028       |
> | Welzl's  algorithm | 0.0203 | **0** | 0.0607  | **0** |
> | ε-kernel + Welzl's algorithm   | 0.0022  | 0.0002 | 0.0031| 0.0002 |
> | $\mathcal{S}\_\mathrm{Baseline}$| 0.0011| 0.020| 0.0100| 1.900 |
>
> Additionally, we provide a comparison of our $\varepsilon$-kernel-approximating network, $\mathcal{S}\_\varepsilon$, against the additional direct neural baseline described above as well as the algorithmic $\varepsilon$-kernel baselines for input point clouds of varying sizes in $\mathbb{R}^5$. Note that some comparison already exists in our paper as we compare $\mathcal{S}\_\varepsilon$ against the relaxed-$\varepsilon$-kernel computed in Algorithm 1 (cf. Appendix D, Figures 5-8).
>
> **Table C**. Comparison of $\mathcal{S}\_\varepsilon$ against baselines
> |Method|200| |400| |800| |
> |--|--|--|--|--|--|--|
> | |Runtime|Error|Runtime|Error|Runtime|Error|
> |$\mathcal{S}\_\varepsilon$|**0.0002**|**0.081**|**0.0005**|**0.088**|**0.0003**|**0.105**|
> |Relaxed ε-kernel|0.0007|0.145|0.0007|0.148|0.0013|0.162|
> |ε-kernel|0.0009|0.085|0.0010|0.099|0.0013|0.111|
> |$\mathcal{S}\_\mathrm{Direct}$|0.0016|0.509|0.0007|0.451|0.0006|0.373|
>
> We will include more comparisons with both classical algorithmic and neural baselines for extent measure problems in our final manuscript.
>
> [1] "Relational Inductive Biases, Deep Learning, and Graph Networks" (Battaglia, et al. 2018)
>
> **Regarding Q4, applicability to real-world datasets:** First, we note that the SQUID dataset contains boundary contours of approximately 1100 marine animals. We use this dataset because boundaries are a natural setting for $\varepsilon$‑kernels and it provides a real‑world 2D point cloud benchmark.
> Additionally, we see that both the relaxed-$\varepsilon$-kernel and the encode-process-decode extent measure model are effective given the real-world 3D point clouds in ModelNet.

---

> ### Comment · Reviewer_r5xy · 2025-08-04
>
> Thanks for providing these baselines. So you gain around ~8x-10x speed-ups compared to (ε-kernel + Welzl's algorithm) but the error increases by ~100x?
>
> This should be in your paper as a major point of discussion / limitation

---

> ### Author Response · Authors · 2025-08-05
>
> Thank you for the observation. We will acknowledge the limitation you point out. However, we would like to clarify that the main contribution of our work is proposing a general framework for solving extent-measures problems, especially those for which an efficient approximation algorithm does not exist. We use minimum enclosing ball as an example of a faithful extent measure problem which can be efficiently solved with our architecture. Because minimum enclosing ball is an LP type problem, it has a relatively efficient solution (although our neural method is still faster). However, we emphasize that our architectures can also work effectively for non LP-type problems where there are no efficient solutions. In fact, we have done additional experiments comparing our architecture with the baseline $\varepsilon$-kernel followed by exact algorithm for minimum-width spherical shell (annulus), which is a non LP-type problem. Here, we find a speedup of approximately 305x with N=200, while still reporting a relatively low error. Note that we include the percentage of points excluded in addition to the error in the computed widths and that our method outperforms the baseline in terms of proportion of points excluded from the approximated spherical shell.
>
> Furthermore, we emphasize that our theoretical results show that our neural architectures can approximate extent measure problems without requiring parameters scaling with input size as compared to previous point cloud processing architectures, which required the parameters to scale at least linearly with the size of the input point cloud.
>
> | Method                 | Input Size | Runtime (s) | Error (↓)  | % Excluded (↓) |
> |------------------------|------------|-------------|------------|----------------|
> | Frozen Sumformer       | 200        | 0.000459    | 0.021643   | 1.61           |
> | ε-kernel + exact       | 200        | 0.141551    | 0.014939   | 2.37           |
> | Frozen Sumformer       | 600        | 0.006520    | 0.023028   | 0.92           |
> | ε-kernel + exact       | 600        | 0.144270    | 0.018481   | 1.60           |

---

> > ### Comment · Reviewer_r5xy · 2025-08-05
> >
> > I agree that your theoretical contributions have their own merit and that it's a novel problem to approach with neural surrogates. You should however be fully transparent in the main paper to show the limitations and fair comparisons to classical/non-deep learning algorithms, not just "naive" neural approaches. The tradeoffs and (and possibilities for future improvements) should be clear to the reader.
> >
> > Since the discussion period was extended a bit, can you provide some simple runtime scaling experiments wrt input size? So for what point cloud input sizes are you very fast or does the $N^2$ transformer scaling prevent you from scaling to larger point cloud sizes efficiently? (not just $N \sim O(100)$)

---

> > > ### Author Response · Authors · 2025-08-06
> > >
> > > Yes, thank you very much for your feedback. We will definitely include further discussion regarding the limitations you pointed out. First, we would like to clarify that in our empirical results, our encode-process-decode model uses a Sumformer as the processor module. In our experiments, we also provide comparisons using a transformer as the processor module processor. The SumFormer in fact requires only an $O(n)$ forward pass where $n$ is the number of input elements. Furthermore, we are happy to provide further experiments regarding the runtime scaling for both an LP-type extent measure problem, minimum enclosing ball, and a non-LP-type extent measure problem, minimum enclosing spherical shell (annulus). See Tables 1 and 2 below. We note that our model consistently performs faster than all algorithmic baselines with significant speedups especially for the non LP-type problem (minimum enclosing spherical shell). As mentioned in our response to VDgZ, while minimum enclosing ball is an LP-type problem with a reasonably efficient algorithm, this is not true for minimum-width spherical shell (takes $O(n^d)$ times where $d$ is the dimension of linearization) and indeed in this case, we see our method offers even greater increases in efficiency.
> > >
> > > Table 1. Runtime scaling for minimum enclosing ball in $\mathbb{R}^2$ (measured in seconds).
> > > |                                | 500      | 900      | 1300     | 1700     | 2000     | 3000     | 4000     |
> > > |:------------------------------:|:--------:|:--------:|:--------:|:--------:|:--------:|:--------:|:--------:|
> > > | $\mathcal{S}_\mathrm{extent}$ | 0.000575 | 0.000906 | 0.001073 | 0.001558 | 0.001840 | 0.002677 | 0.003561 |
> > > | Welzl                          | 0.05182  | 0.098125 | 0.15666  | 0.227613 | 0.279176 | 0.460060 | 0.749621 |
> > > | $\varepsilon$-kernel + Welzl   | 0.003216 | 0.003704 | 0.004193 | 0.004361 | 0.004525 | 0.004842 | 0.005180 |
> > >
> > > Table 2. Runtime scaling for minimum enclosing spherical shell $\mathbb{R}^2$ (measured in seconds).
> > > |                                | 500      | 900      | 1300     | 1700     | 2000     | 3000     | 4000     |
> > > |:------------------------------:|:--------:|:--------:|:--------:|:--------:|:--------:|:--------:|:--------:|
> > > | $\mathcal{S}_\mathrm{extent}$ | 0.007821 | 0.001067 | 0.001394 | 0.008612 | 0.010599 | 0.012609 | 0.009303 |
> > > | Exact                          | 0.193887 | 0.216636 | 0.241404 | 0.268823 | 0.282620 | 0.356186 | 0.383274 |
> > > | $\varepsilon$-kernel + Welzl   | 0.145017 | 0.146992 | 0.150136 | 0.152979 | 0.155251 | 0.165896 | 0.162305 |

---

> > > > ### Comment · Reviewer_r5xy · 2025-08-07
> > > >
> > > > Thanks for the fast results, I would suggest you highlight these and also that Sumformer is only $O(N)$, I missed this. I'm satisfied with the rebuttal of the authors and will recommend acceptance. It is novel work in a new direction for neural approximations, even if the errors are sitll quite high currently. With the additions the authors have provided I will increase my score to 5.

---

### Official Review · Reviewer_VDgZ · 2025-06-28

**Clarity:** 1
**Significance:** 2
**Originality:** 2
**Rating:** 2
**Confidence:** 4

**Summary:**

This paper proposes a neural approach to address geometric "extent measure" problems. A key contribution is the introduction of the relaxed-ϵ-kernel, with a neural network employed as the fitting function. The effectiveness of the approach is demonstrated through experiments conducted on both synthetic and real datasets.

**Questions:**

* Could the author provide a more detailed discussion of the motivation?
* The related work section is essential, but it is currently missing.
* It is necessary to include comparisons with neural baselines and extensive classic baselines.
* Adding more visual comparisons would be helpful to verify that the proposed method achieves reasonable performance.

**Ethical Concerns:**

["NO or VERY MINOR ethics concerns only"]

**Final Justification:**

I appreciate the effort made by the authors to provide additional results. However, I would like to emphasize that my major concern remains unresolved:

- Very Limited "Related Work" in Paper: Although the authors included a "related work" paragraph focusing on the neural methods, they did not provide a discussion about the differences between the proposed algorithm and traditional algorithms.

- Unclear Motivation: The motivation for using neural networks to solve the geometric "extent measure" problem remains unclear. One possible advantage should be faster inference; however, the authors have not provided sufficient comparisons with existing methods to demonstrate this. In fact, the results in the provided table (Table B in the rebuttal) suggest that the original exact algorithm is already fast enough to support real applications. More importantly, the authors should compare the proposed method with widely-used approximate algorithms, but this important comparison is currently missing.

- Insufficient Comparisons: Providing comparisons with traditional methods is an important aspect of the experimental evaluation. Instead of comparing only with self-implemented neural variants, the authors should have included comparisons with traditional methods. Initially, no such comparisons were provided. During the rebuttal, the authors only included comparisons with one baseline on a single task, which is insufficient.


As a result, I believe the current draft is not yet ready. The motivation and contribution remain unclear, and the effectiveness is inadequately demonstrated.

**Limitations:**

Yes.

**Paper Formatting Concerns:**

No.

**Quality:**

1

**Strengths And Weaknesses:**

Strengths:

* This work presents a neural solution for addressing geometric "extent measure" problems, which is an interesting attempt.
* The paper includes some theoretical analysis.

Weaknesses:

* The motivation behind this work is unclear. The geometric "extent measure" problem is a well-studied area. It is not evident what specific advantages a neural solver brings to this problem. Furthermore, there is no discussion about the potential benefits of the proposed approach.
* The lack of a related works section makes it difficult for readers to evaluate the distinct innovations of the proposed method compared to existing approaches.
* There is a lack of comprehensive comparisons. The experimental section only includes results from some proposed neural variants, which does not sufficiently verify the effectiveness of the proposed method over existing works.
* Important experimental results are missing. Specifically, there is no time analysis or visual results, which limits the readers’ ability to fully understand the effectiveness and utility of the proposed method.

---

> ### Author Rebuttal · Authors · 2025-07-31
>
> Thank you for your time and careful review. In what follows, we will address your main questions. We think that there might be some misunderstanding of our paper. In particular, as we will detail below, our paper does have a related work section and we also have several baselines methods for comparison.
>
> **Regarding Q1, motivation:** We would like to clarify our motivation here. As we explain in the introduction of our paper, the key motivation behind our work is to design efficient and practical neural models to solve hard problems in a data-driven manner.
> In particular, we consider problems which take point cloud data as input. We are interested in models that both have the expressivity to approximately solve the problem using bounded size (i.e. fixed number of parameters) and produce high quality and efficient approximations.
>
> First, we observe that popular models for handling sets or sequences such as DeepSets, Sumformers, and Transformers usually require model complexity scaling with the size of input point sets in order to $\varepsilon$-approximate an arbitrary function.
> Our work shows that by *aligning neural design with algorithmic principles*, specifically by leveraging the frameworks of $\varepsilon$-kernels and linearization, we can construct two neural architectures which solve geometric extent measure problems with model complexity \textit{independent} of the input point set size. We note that finding accurate and efficient approximation algorithms for extent measure problems in the case of large point clouds remains of interest to the computational geometry community (see [1]).
>
> Importantly, our paper goes beyond just an expressivity argument. We demonstrate through extensive empirical experiments that our models are more efficient and accurate than both natural neural baselines and pure algorithmic approaches in terms of both approximating $\varepsilon$-kernels as well as solving extent measure problems. While our neural models are inspired by algorithmic approaches, those algorithmic ideas only provide high level scaffolding: we demonstrate through extensive empirical experiments that their data-driven design makes them more efficient and accurate than both natural neural baselines (see Tables 2, 13, 14, and 15 in our paper) and  pure algorithmic approaches (see Figures 5-8 in our paper). Below, we also include new experiments comparing the runtime of our neural models at inference time with classical algorithms and new natural baselines (Tables A and B below).
>
> In short, we believe that our work provides a compelling example that using algorithmic ideas can lead to more practical neural models (than both classical algorithmic approaches and baseline neural networks) for solving a family of geometric optimization problems, and contribute to the general research direction of neural algorithmic design.
> We will be happy to revise our Introduction to make these points more clear.
>
> [1] "Computing Instance-Optimal Kernels in Two Dimensions" (Agarwal and Har Peled, Discrete and Computational Geometry 2024)
>
> **Regarding Q2, our related work section:** In fact, we do have a related work section on page 2. We reiterate the main points in our related work section here.
> First, there are many architectures focused on point cloud processing -- the oldest of which are DeepSets and PointNet and more recently, transformers. Most, if not all, applications to date have come from computer vision. The utility of neural networks to act as solvers for computational geometry is previously unexplored. Our work is the first to apply neural networks for approximating solutions to common geometric problems from computational geometry. Therefore, in lieu of comparisons to previous work, we propose our own neural baselines for comparison in the paper (which we will further elaborate on in our response to your question about baselines). In our related works section, we acknowledge several previous works on studying the expressivity of point cloud networks. In particular, we highlight that for general functions over point clouds, the best known lower bounds for expressivity still depends linearly on the size of the input point set (See [1]). We emphasize that for the specific case of extent measure problems, our work demonstrates that one can achieve $\epsilon$-accuracy with parameters \textit{independent} of the input point set size using our specific sumformer architecture.
>
> [1]"Neural Injective Functions for Multisets, Measures and Graphs via a Finite Witness Theorem" (Amir, et al. NeurIPS 2023)
>
> **Regarding Q3, baselines:** Thank you for your suggestion regarding comparing to more neural and algorithmic baselines.
> To the best of our knowledge, ours is the first work considering using neural networks as solvers for geometric extent measure problems. Therefore, to understand the power of leveraging algorithmic frameworks in our $\varepsilon$-kernel approximating NN, we compare with natural neural baselines. In our paper, we propose baselines built on the well-known encoder-processor-decoder paradigm (see [1]). We use a simple embedding MLP as the encoder network, the processor network is some suitable architecture (e.g. a transformer or sumformer) which processes embedded point cloud as input and aggregates the resulting representations into a global feature vector, and the a final MLP as the decoder network.
> The Sumformer and transformer versions of this baseline, $\mathcal{S}\_{\mathrm{Baseline}}$ and $\mathcal{T}\_\mathrm{Baseline}$ respectively, are exactly the neural baselines used for comparison in our paper.
>
> To give a case study as to why this is the neural baseline we compare against, we consider the minimum enclosing ball problem. The most straightforward neural approach would be to directly aggregate the point cloud input into a global feature vector using some architecture (sumformer or transformer) and then directly compute a final estimate for the radius of the minimum enclosing ball with a final MLP. We will refer to the Sumformer versions of this baseline as $\mathcal{S}_\mathrm{Direct}$. Our included encode-process-decode baseline is a slightly more complicated version of this as we do an initial encoding step with the encoder network. We show in Tables 2 and 13 in our paper that our encode-process-decode extent measure model using the $\varepsilon$-kernel-approximating Sumformer network as the processor network outperforms both encode-process-decode baselines ($\mathcal{S}\_\mathrm{Baseline}$ and $\mathcal{T}\_\mathrm{Baseline}$) on the minimum enclosing ball problem.
> To further illustrate the utility of our algorithmically aligned extent measure model $\mathcal{S}\_\mathrm{extent}$, we include additional results comparing our original baseline $\mathcal{S}\_\mathrm{Baseline}$ against $\mathcal{S}\_\mathrm{Direct}$ on the minimum enclosing ball problem for the SQUID dataset in the table below (Table A).
> | **Model** | $\mathcal{E}\_{r}$ | $\mathcal{E}\_\mathrm{p}$ |
> |-|-|-|
> | $\mathcal{S}\_\mathrm{Direct}$ | 0.986 ± 0.004  | 1.000 ± 0.000  |
> | $\mathcal{S}\_\mathrm{extent}$ (frozen) | **0.065 ± 0.055** | **0.052 ± 0.056**  |
> |$\mathcal{S}\_\mathrm{extent}$ (E2E) | 0.428 ± 0.057 | 0.596 ± 0.106 |
> | $\mathcal{S}\_\mathrm{Baseline}$  | 0.687 ± 0.080| 0.881 ± 0.089|
>
> Additionally, we thank the reviewer for the suggestion of comparing against additional baselines. In Table B below, we provide a further comparison of our extent measure approximating model, $\mathcal{S}\_\mathrm{extent}$, against the classical algorithms for the minimum enclosing ball task with different input point cloud sizes. In particular, we compare our neural model's runtime at inference time with Welzl's algorithm. Since Welzl's algorithm is an exact algorithm so we include a comparison to the classical exact $\varepsilon$-kernel algorithm followed by Welzl's algorithm. While our network is less accurate than using a $\varepsilon$-kernel followed by Welzl's algorithm, we observe that our network is significantly faster than all baseline algorithms and more accurate than neural baselines.
>
> **Table B**. Comparison of extent measure model against algorithmic baselines
> | **Method** | **200**| | **600**| |
> |-|-|-|-|-|
> |   | Runtime | Error| Runtime| Error|
> | $\mathcal{S}\_\mathrm{extent}$ (frozen) | **0.0004** | 0.020    | **0.0007**     | 0.028       |
> | Welzl's  algorithm | 0.0203 | **0** | 0.0607  | **0** |
> | ε-kernel + Welzl's algorithm   | 0.0022  | 0.0002 | 0.0031| 0.0002 |
> | $\mathcal{S}\_\mathrm{Baseline}$| 0.0011| 0.020| 0.0100| 1.900 |
>
> Additionally, we provide a comparison of our $\varepsilon$-kernel-approximating network, $\mathcal{S}\_\varepsilon$, against the additional direct neural baseline described above as well as the algorithmic $\varepsilon$-kernel baselines for input point clouds of varying sizes in $\mathbb{R}^5$. Note that some comparison already exists in our paper as we compare $\mathcal{S}\_\varepsilon$ against the relaxed-$\varepsilon$-kernel computed in Algorithm 1 (cf. Appendix D, Figures 5-8).
>
> **Table C**. Comparison of $\mathcal{S}\_\varepsilon$ against baselines
> |Method|200| |400| |800| |
> |--|--|--|--|--|--|--|
> | |Runtime|Error|Runtime|Error|Runtime|Error|
> |$\mathcal{S}\_\varepsilon$|**0.0002**|**0.081**|**0.0005**|**0.088**|**0.0003**|**0.105**|
> |Relaxed ε-kernel|0.0007|0.145|0.0007|0.148|0.0013|0.162|
> |ε-kernel|0.0009|0.085|0.0010|0.099|0.0013|0.111|
> |$\mathcal{S}\_\mathrm{Direct}$|0.0016|0.509|0.0007|0.451|0.0006|0.373|
>
> We will include more comparisons with both classical algorithmic and neural baselines for extent measure problems in our final manuscript.
>
> [1] "Relational Inductive Biases, Deep Learning, and Graph Networks" (Battaglia, et al. 2018)
>
> **Regarding Q4, visualizations:** Thank you for your suggestion, we will include many more visualizations in our final manuscript. We would like to note that there are further experiments and visualizations present in our Appendix D.

---

> ### Comment · Reviewer_VDgZ · 2025-08-02
> **The limited analysis and discussion of traditional methods.**
>
> After reading the authors' rebuttal, I find that the following concerns remain unaddressed:
>
> - Unclear Motivation: The motivation for using neural networks to solve the geometric "extent measure" problem remains unclear. One possible advantage should be faster inference; however, the authors have not provided sufficient comparisons with existing methods to demonstrate this. In fact, the results in the provided table (Table B in the rebuttal) suggest that the original exact algorithm is already fast enough to support real applications. More importantly, the authors should compare the proposed method with widely-used approximate algorithms, but this important comparison is currently missing.
>
> - Insufficient Comparisons: Providing comparisons with traditional methods is an important aspect of the experimental evaluation. Instead of comparing only with self-implemented neural variants, the authors should have included comparisons with traditional methods. Initially, no such comparisons were provided. During the rebuttal, the authors only included comparisons with one baseline on a single task, which is insufficient.
>
> - Very Limited "Related Work" in Paper: Although the authors included a "related work" paragraph, they did not provide a discussion about the differences between the proposed algorithm and traditional algorithms.
>
> In my opinion, the current version is not ready for publication. I will maintain my original score.

---

> > ### Author Response · Authors · 2025-08-05
> > **response to reviewer**
> >
> > Thank you for your time and detailed follow-up. We would like to clarify two points. First, we use minimum enclosing ball as an example to show how fast the neural approach is compared to algorithmic approaches. Table B does show a significant difference in inference speed between our model and comparable algorithms (N=600, our method displays an 85x speedup vs the exact Welzl's algorithm). This speedup increases with the number of input points (for N=2000, 0.26 seconds for Welzl's vs 0.0019 for ours). We note that minimum enclosing ball is an LP type problem and therefore has a reasonably efficient algorithm. However, with problems such as minimum-width spherical shell, they are not LP-type, non-convex, and are harder.
> > In particular, for minimum-width spherical shell, one can use linearization solve it exactly in time $O(n^d)$ (see ``Efficient Approximation and Optimization Algorithms for Computational Metrology" (Duncan, et al. 1997)) where $d$ is the dimension of linearization. In this case, our method offers even greater increases in speed while also outperforming the baseline $\varepsilon$-kernel followed by exact algorithm technique in terms of proportion of points included in the minimum width spherical shell. See the table below.
> >
> > | Method                 | Input Size | Runtime (s) | Error (↓)  | % Excluded (↓) |
> > |------------------------|------------|-------------|------------|----------------|
> > | Frozen Sumformer       | 200        | 0.000459    | 0.021643   | 1.61           |
> > | ε-kernel + exact       | 200        | 0.141551    | 0.014939   | 2.37           |
> > | Frozen Sumformer       | 600        | 0.006520    | 0.023028   | 0.92           |
> > | ε-kernel + exact       | 600        | 0.144270    | 0.018481   | 1.60           |
> >
> > Secondly, we would again like to point out that in figures 5-8 of our original paper, we did compare our models with the relaxed $\varepsilon$-kernel algorithm, and we have included more comparisons for relaxed-$\varepsilon$-kernels here (Table C). Due to limited time and word count in the rebuttals, we were unable to provide more thorough results, but would be happy to include them in a future revision.

---

> ### Comment · Reviewer_VDgZ · 2025-08-05
>
> I appreciate the effort made by the authors to provide additional results. However, I would like to emphasize that my major concern remains unresolved: the original main paper omits several critical components, including necessary discussions (e.g., detailed reviews of traditional methods) and sufficient comparisons with existing state-of-the-art traditional methods.
>
> Additionally, the experimental results provided during the rebuttal involve only one traditional baseline on a single task, without a detailed description of the experimental setup. Furthermore, Figures 5–8 in the original paper are included in the supplementary material, but they seem to be based on self-implemented baselines, as no references are provided.
>
> As a result, I believe the current draft is not yet ready. The motivation and contribution remain unclear, and the effectiveness is inadequately demonstrated.

---

> > ### Author Response · Authors · 2025-08-06
> >
> > Thank you for your comments and suggestions during the discussion phase, they will be included in our revised manuscript. We'd like to clarify that the algorithmic baselines we've compared with in the rebuttal and discussion period are the state-of-the-art algorithmic baselines for the minimum enclosing ball and minimum-width spherical shell (annulus) tasks, Welzl's algorithm and furthest/nearest neighbor Voronoi diagrams based algorithms respectively. Table C in our previous comment contains results on the minimum width spherical shell problem and Table B in our original reply contained results on the minimum enclosing ball problem. Our method provides significant increases in inference speed without a significant loss in accuracy on both tasks, providing evidence of our method's viability on LP-type (min. enclosing ball) and non-LP-type (min-width spherical shell) problems. Please also see our most recently reply to Reviewer r5xy for a more extensive runtime analysis on both minimum enclosing ball and minimum width spherical shell. If there are additional algorithmic baselines that you believe we have missed, please let us know so that we may improve our manuscript for future revisions.

---

### Official Review · Reviewer_QaAt · 2025-06-30

**Clarity:** 2
**Significance:** 3
**Originality:** 4
**Rating:** 4
**Confidence:** 2

**Summary:**

This manuscript proposes a way of approximating certain quantities about point clouds with bounded model complexity (and linear runtime complexity). It does so by introducing the concept of “relaxed epsilon-kernel”, which unlike the traditional epsilon-kernel can be built out of points in the entire convex hull, not a subset of the point cloud. This approach is evaluated in a thorough way theoretically and in practical examples.

**Questions:**

What theoretical effects does changing the L1 normalization for a softmax have? Which results remain valid, and which are not? As a reader, I was a bit perplexed by this choice, and while I am convinced by the results that it is in practice better, I was left wondering how much of the prior 3 sections are actually theoretically justified when one performs this change.

What are some examples of unfaithful extent measures that one can keep in mind when reading the text? What are some examples of linearizable and non-linearizable ones? Adding such examples in the text will make Sec. 4 feel much more grounded.

To clarify: once trained, the same trained network can be used to process point clouds of different sizes, correct?

What does the “hyperparameter search” entail?

What is the (input) noise sensitivity of this entire pipeline?

**Ethical Concerns:**

["NO or VERY MINOR ethics concerns only"]

**Final Justification:**

The authors have resolved all my concerns, and I remain generally positive about this work. At the same time, other reviewers (with potentially more expertise in this topic than me) have raised what sound like reasonable concerns, so I look forward to hearing their opinion on the rebuttal conversation

**Limitations:**

In my opinion, the main limitation of the work is its applicability: it is relevant only for a small number of “simple” point cloud operations (as opposed to, e.g., reconstruction or triangulation), and likely has diminishing returns as the dimension of the data approaches the number of data points. That being said, these limitations are discussed accurately and openly, and I do not penalize this work for it.

**Paper Formatting Concerns:**

The title in Openreview and the pdf are different.

**Quality:**

3

**Strengths And Weaknesses:**

The main strength of the work is the theoretical contribution of introducing relaxed epsilon kernels and their relationship to commonly-used neural architectures. These seem like important contributions to the study of point clouds from the computational geometric perspective, and I can imagine they will inspire future work in this direction. Another major strength is the evaluation: all theoretical results are conclusively proven, including careful tracking of bounds and infinitesimal constants. The practical evaluation of the method in the supplemental also goes well beyond what is expected of such a theoretically heavy manuscript.

I will preface this by saying that I am not an expert on computational geometric problems like this one, so there is definitely a chance I missed a mistake in a proof or some other major limitation. I will carefully read other reviewers’ comments as well as the authors’ response.

Nonetheless, I could not find any major issue in this work. If anything, the sole reason I would not rank this manuscript higher is the didactic exposition of the method. For something as visual as point cloud processing, there is a single didactic figure (Fig. 3), and I cannot understand exactly what it is showing. What is u? Where is the origin, so I can understand p as a vector? Also, the font is way too small. I would really much prefer if all of sections 2 and 3 were followed by a study of the same didactic 2D point cloud: it would make an intuitive understanding of the concepts significantly easier.

I would also encourage the work to be double-checked for typos (e.g., “computatoin”) and missing articles.

---

> ### Author Rebuttal · Authors · 2025-07-31
>
> Thank you for time and careful review. We are glad the reviewer appreciates both our theoretical results relating the $\varepsilon$-kernel framework to neural networks as well as the extensive practical evaluation for both approximating $\varepsilon$-kernels as well as three different extent measure problems. Thank you for your suggestion regarding visual aids, we will definitely add more to our final manuscript.
>
> **Regarding Q1 about softmax vs. $L_1$ normalization:** We thank the reviewer for raising this question, which is a very valid one. In fact, it turns out that using softmax can serve just as well as $L_1$ normalization for our theoretical framework. The key requirement for our results to hold is simply a mechanism to transform the vector $\rho_\varepsilon(\mathbf{a})$ (which intuitively records how far each point is from the maximum point in a given direction) into a valid probability distribution over the input point set, assigning zero weight to any points located more than $\varepsilon w(A)$ from the maximum. While we adopt $L_1$ normalization, one could equivalently use a softmax variant e.g. $\frac{e^{z_i} - 1}{\sum_j \left(e^{z_j} - 1\right)}.$ Any normalization scheme satisfying the aforementioned requirement will work as we only need the normalization in order give final output points which are convex combinations of the input points lying within $\varepsilon w(A)$ of the true maximum in a given direction. We will add this softmax guarantee to our revised manuscript. Note that in Appendix A.5, Table 5, we compare softmax against $L_1$-normalization and find that softmax works much better in practice.
>
> **Regarding Q2, about example extent measure problems:** Thank you for your suggestions. A linearizable unfaithful extent measure that one can keep mind while reading Section 4 is the minimum enclosing annulus. In fact, the appendix has an example of using relaxed-$\varepsilon$-kernels with the minimum enclosing annulus problem.
>
> **Regarding Q3, input point set size:** Yes, our network is easily able to handle input point sets of different sizes as point sets of any size can be used as inputs to sumformers. Additionally, our theoretical results provide a guarantee that the network can still $\varepsilon$-approximate solutions to extent measure problems without scaling the number of parameters in the network.
>
> **Regarding Q4, about hyperparameter searching:** For our hyperparameter search on the $\varepsilon$-kernel task, we chose the sumformer/transformer and MLP depth from $\{2, 3, 4\}$, the hidden dimension from $\{128, 256, 512, 1024\}$, and embedding dimension from $\{16, 32\}$. For the extent-measures models, we fix the hyperparameters of the processor based on the best configuration from the $\varepsilon$-kernel task and vary those of the encoder and decoder, randomly selecting combinations and manually adjusting promising configurations. The depth of both the encoder and decoder MLPs are sampled from $\{2, 3, 4\}$, and the width from $\{64, 128, 256\}$.
>
> **Regarding Q5, robustness to input noise:**  We believe that our models are robust to unseen noisy data as they generalize well to out-of-distribution data at inference time. In Table 9 and 10, we have detailed results showing the performance of our relaxed-$\varepsilon$-kernel approximating network trained on simple synthetic datasets but tested of a variety of synthetic and real datasets. In these tables, we observe that the relaxed-$\varepsilon$-kernel approximating network, which is aligned with Algorithm 1 in our manuscript, displays similar test error across all datasets (including those which are out-of-distribution).
> Additionally, we show in Table A (in our response to Reviewer VDgZ) for our extent measure approximating network, a model trained on point clouds sampled from uniform disk still performs well on real-world SQUID dataset.

---

> > ### Comment · Reviewer_QaAt · 2025-08-02
> >
> > Thank you very much for your response. My concerns are addressed, and I am reassured about the applicability of the method. I would encourage you to include these clarifications in the final text if accepted.

---

### Official Review · Reviewer_WDZw · 2025-07-03

**Clarity:** 3
**Significance:** 3
**Originality:** 3
**Rating:** 5
**Confidence:** 2

**Summary:**

This paper defines the relaxed-$\varepsilon$-kernel as a kind of universal coreset for approximating faithful extent measures. The paper then generalizes this approach to unfaithful extent measures via a polynomial lifting/linearization trick. It is shown that a sumformer can approximate an algorithm to compute a relaxed kernel. This motivates the design of an architecture to approximate kernels, which, when composed with a learned encoder and decoder, can be used to approximate extent measrues. The experimental results compare a few architectures and training protocols on the tasks of learning to approximate kernels and extent measures.

**Questions:**

- If I understand correctly, the complexity of Algorithm 1 is bounded in the number of input points, but scales exponentially with the dimension $d$. Isn't this the more pernicious scaling? Standard algorithms for convex hull etc. work just fine on a large number of points in low dimension (2 or 3). In what context do you envision your method or model being used. In other words, in what range of dimensions, approximation quality $\varepsilon$, etc. is the approximation/time tradeoff reasonable?
- The fatness of the point set is also an important scaling parameter. Does the fatness not scale with the number of points in the worst case?
- What exactly do you mean by the "alignment" or "resemblance" of a model with an algorithm? I can see how you can design very specific model weights to enact the algorithm, as you do to prove Theorem 3.4. But this is just rewriting the algorithm in a different language. Does the fact that you can do this by hand somehow imply that the model class should generically learn to enact the approximation algorithm you want? Or at least that it will learn to approximate the algorithmic result better than any other model architecture would?
- Does the alignment happen in practice? I.e., can you see if the learned weights agree with the ones you would impose by hand to encode the approximation algorithm?
- In Algorithm 1, the required $\delta$-net size still scales exponentially with $d$. Doesn't this pose a problem for the bounded model complexity?
- Did you try any experiments in dimension higher than 5? In high dimension is presumably where the scaling performance would really start to matter.

**Ethical Concerns:**

["NO or VERY MINOR ethics concerns only"]

**Final Justification:**

I appreciate the authors' responses and new experiments. I understand better the value of the learned model over the classical approximation algorithm.

**Paper Formatting Concerns:**

The paper title does not match submission title. The paper refers to proofs and other results in an appendix, but they are actually in the supplementary material.

**Quality:**

2

**Strengths And Weaknesses:**

It seems the paper's most important contribution is in defining and proving results about relaxed $\varepsilon$-kernels. However, this is outside my usual domain of expertise, and I am not sure how much of a contribution this is relative to previous work in computational geometry. This paper might receive more careful and incisive peer review at a computational geometry conference rather than NeurIPS.

I am skeptical of the learning aspect of this work, and in particular of the whole "algorithmic alignment" idea. It seems the argument is that because a particular architecture X (in this case based on the sumformer) is capable of enacting a particular algorithm Y, that therefore X will learn to approximate Y during training. I don't see why this follows. Since no theoretical justification for this claim is provided (nor would I expect it would be possible to prove something like this), it seems like more rigorous empirical evaluation of the learning claim is merited.

If it were the case that the kernel approximation network were really learning to approximate Algorithm 1 to compute relaxed-$\varepsilon$-kernels, I'd expect it to have nearly perfect generalization to arbitrary input point clouds. However, in Figures 9 and 10 in the appendix, test error appears to get worse during training for the two OOD datasets. This suggests that the model is not actually learning a universal algorithm for computing relaxed $\varepsilon$-kernels, but rather is learning some heuristics that work on a particular dataset.

On a more philosophical note, given that Algorithm 1 is simple and efficient and that it comes with hard theoretical guarantees, what is the advantage of learning to approximate Algorithm 1? Even if it were guaranteed that the kernel approximation network would learn to approximately enact Algorithm 1, it would still be more efficient and reliable just to use Algorithm 1 directly.

---

> ### Author Rebuttal · Authors · 2025-07-31
>
> Thank you for your time and careful review! We are glad the reviewer appreciates our introduction of the relaxed-$\varepsilon$-kernel.
>
> **Regarding algorithmic alignment:** Before we respond to your questions, we would first like to elaborate on the motivation and benefit of algorithmic alignment. Overall, the framework of algorithmic alignment allows us to use algorithmic structure as a scaffold for the design of some task-specific network. By designing the model to imitate some algorithmic flow, we bias the model towards learning some task-specific subroutine which will help it generalize better to out-of-distribution data. Notably, the data-driven model may learn something different and better than the original algorithm used to design the model. See [1], [2], [3] for previous examples of how algorithmic alignment enhances model performance.
>
> In our work, we use two types of algorithmic alignment. First, we observe the alignment between the $\varepsilon$-kernel algorithm and the SumFormer to define a relaxed-$\varepsilon$-kernel approximating neural network. Second, we use the relaxed-$\varepsilon$-kernel approximating neural network as a processor in the *encode-process-decode* (EPD) framework. This entire encode-process-decode framework can be aligned with solving a geometric extent measure problem via linearization (encode), generating a coreset for the problem with a $\varepsilon$-kernel (process), and finally solving the extent measure problem with a downstream algorithm (decode). Theoretically, this alignment allows us to show that geometric extent measure problems can be solved with neural networks with model complexity *independent* of the size of the input point set.
>
> Similar to previous work on algorithmic alignment, we empirically observe our aligned sumformer model, denoted as $\mathcal{S}\_\varepsilon$, exhibits better performance in approximating $\varepsilon$-kernels as compared to a baseline transformer architecture (cf. Table 1) and our aligned EPD model is more accurate than natural neural baselines in approximating solutions to extent measure tasks (cf. Table 2). In fact, when compared to Algorithm 1 which guides the design of our model, we see that $\mathcal{S}\_\varepsilon$  outperforms the relaxed-$\varepsilon$-kernel algorithm in terms of accuracy (cf. Figures 5-8 in the Appendix). Below, we include additional comparisons for $\mathcal{S}\_\varepsilon$ against both the relaxed-$\varepsilon$-kernel and the classical $\varepsilon$-kernel algorithm in $\mathbb{R}^5$ at different input point set sizes. We see that $\mathcal{S}_\varepsilon$ has a much lower runtime and error, where error is measured by the directional width error, than both algorithmic baselines.
> | **Method**           | **200**        |               | **400**        |               | **800**        |               |
> |----------------------|----------------|---------------|----------------|---------------|----------------|---------------|
> |                      | Runtime        | Error         | Runtime        | Error         | Runtime        | Error         |
> | $\mathcal{S}\_\varepsilon$ (ours) | **0.0002**     | **0.081**     | **0.0005**     | **0.088**     | **0.0003**     | **0.105**     |
> | Relaxed ε-kernel     | 0.0007         | 0.145         | 0.0007         | 0.148         | 0.0013         | 0.162         |
> | ε-kernel             | 0.0009         | 0.085         | 0.0010         | 0.099         | 0.0013         | 0.111         |
>
> Additionally, for the EPD framework, we see more evidence of algorithmic alignment improving the performance of the model in Table 2: the extent measure networks with frozen processor modules which were *explicitly* trained on approximating $\varepsilon$-kernels outperform the end-to-end trained model.
>
> Finally, we appreciate your observation regarding the behavior of the test error on out-of-distribution datasets in Figures 9 and 10. However, we would like to clarify that these figures are intended to support our decision to train the models for only 200 epochs where performance on out-of-distribution data is strong. The degradation in out-of-distribution performance over longer training durations is consistent with this interpretation and does not contradict our core claims. In sum, algorithmic alignment provides high-level guidance for designing networks which can (1) generalize well on the specific algorithmic task which it is aligned to and (2) provide boosts in performance for downstream tasks in which the network can be used as a module.
>
> [1] *The CLRS Algorithmic Reasoning Benchmark* (Veličković et al., ICML 2022).
> [2] *Neural Execution of Graph Algorithms* (Veličković et al. ICLR 2020).
> [3] *Learning Graph Algorithms with Recurrent Graph Neural Networks* (Grötschla et al. GCLR Workshop @ AAAI 2023).
>
> **Regarding Q1 and Q5, about the complexity of Algorithm 1 and scaling with dimension $d$:** Yes, the complexity of Algorithm 1 is bounded in the number of input points but scales exponentially with the dimension $d$. The *alignment* with Algorithm 1 makes it so that our model has bounded model complexity *in terms of the size of the input point cloud*.
> Scaling with the size of the input point set is the far more pernicious scaling for most real-world point cloud processing applications as for most applications use low-dimensional but large scale point clouds (potentially millions of points per input point set).  Therefore, scaling with the dimension is fine (as most applications use low dimensions) but the required model parameters scaling with the number of points in order to keep some level of expressivity is less ideal.  Our universality guarantee which avoids dependence on the size of the input point cloud is particularly significant, as prior universality results for standard point cloud models required the number of parameters to scale with the size of the input point set in order to achieve universality.
>
> **Regarding Q2, about the fatness of the point set:** First, we note that $\alpha$-fatness is a standard assumption in computational geometry. Intuitively, a larger $\alpha$ indicates that the point cloud is not excessively “skinny,” while smaller values of $\alpha$ correspond to increasingly elongated or degenerate point sets. In the extreme case, $\alpha$ can shrink inversely with the number of points in the input—this occurs when the point cloud is effectively degenerate, exhibiting no variation in at least one direction. Our theoretical guarantees hold even for point sets with low $\alpha$ values; the approximation factor in our analysis scales with $\alpha$. Consequently, as $\alpha$ decreases (i.e., the point sets become “skinnier”), the theoretical approximation factor worsens, and more parameters are required. Despite this, we observe strong empirical performance across a wide range of $\alpha$ values (cf. Figure 6 in the appendix). Additionally we find that real-world datasets tend to have relatively large $\alpha$ values. For example, in the widely used ModelNet dataset, where each point cloud lies within the bounding box $[-1, 1]^d$, the average $\alpha$-fatness is 0.81 (with the maximum possible value being 1).
>
> **Regarding Q3, about the “alignment” of the model with an algorithm:** We touched on this point previously but will reiterate it in brief again. The alignment to Algorithm 1 as a high-level scaffold for our models. However, the data-driven model may learn something different and better than the original algorithm. Therefore, we are unable to directly observe if the learned weights implement Algorithm 1. We do observe that our aligned sumformer model learns to approximate the relaxed-$\varepsilon$-kernel better than a baseline transformer architecture (cf. Table 1) and to our point that the data-driven model learns a more efficient and accurate algorithm than the original, we observe in Figures 5–8 in our appendix that our trained models outperform the baseline algorithm in terms of error. In our revised manuscript, we will include more results comparing the runtime and accuracy of our model against the baseline algorithm as well as another natural baseline neural architecture. Preliminary results of this nature are included in our response to Reviewer VDgZ in Table C. For example, we find that for a point cloud in $\mathbb{R}^5$ with 2000 points, our relaxed-$\varepsilon$-kernel approximating neural network computes an output $\varepsilon$-kernel with 150 points in 0.0003 seconds while Algorithm 1 requires 0.002 seconds to output a $\varepsilon$-kernel with 150 points.
>
> **Regarding experiments on higher-dimensional point clouds:** We conducted additional experiments on synthetic point clouds in 10 dimensions and summarize some preliminary results. For a point cloud with 2000 points sampled uniformly from a randomly stretched and shifted ellipse, our SumFormer model (outputting a $\varepsilon$-kernel with 200 points) has a directional width error of 0.252 while the output point cloud from Algorithm 1 (also outputting a $\varepsilon$-kernel with 200 points) has a directional error of 0.269. Additionally, our SumFormer model has runtime $2 \times 10^4$ seconds while Algorithm 1 has runtime $3.3 \times 10^3$ seconds. In short, even for high-dimensional datasets, the directional width error for our model is better than the baseline algorithm while being significantly more efficient. We will include more results on high-dimensional data in our revision.

---

> > ### Comment · Area_Chair_Pnkt · 2025-08-05
> >
> > Dear reviewer WDZw,
> >
> > The authors have responded to the original reviews. Could you read the rebuttal and share your thoughts? Does it address your original concerns? Are there any remaining questions for the authors?
> >
> > Best,
> > AC

---

> > ### Comment · Reviewer_WDZw · 2025-08-07
> >
> > I appreciate the authors' detailed responses and the additional experiments, comparing against the algorithmic baseline, and evaluating the method in higher dimension. I did not previously realize that the learned model outperfoms the algorithm. I agree that this is a point to highlight. I am still a little bit confused about the nature of "alignment" here. I am naturally curious why the algorithmically-aligned architecture outperforms baselines, if not because it actually learns the approximation algorithm.
> >
> > I am puzzled by the authors' comment that Figures 9 and 10 "are intended to support our decision to train the models for only 200 epochs where performance on out-of-distribution data is strong. The degradation in out-of-distribution performance over longer training durations is consistent with this interpretation and does not contradict our core claims." If the model is in fact learning some (possibly improved) version of the relaxed-$\epsilon$-kernel approximation algorithm, why would generalization performance degrade after 200 epochs?

---

> > > ### Author Response · Authors · 2025-08-08
> > >
> > > Thank you for your comments. We will be sure to highlight the fact that the model outperforms the baseline algorithms in our final revision.
> > > Regarding your question about alignment, we would like to clarify that the model does not necessarily learn to implement the algorithm. By incorporating the principles of the $\varepsilon$-kernel algorithm into our Sumformer architecture, we bias the models to learn some task-specific subroutine (which may be better than the original $\varepsilon$-kernel). We hypothesize that the Sumformer architecture may be better able to learn relevant directions as opposed to just taking a $\delta$-net (as in the baseline $\varepsilon$-kernel algorithm). The exact nature of this subroutine is unknown and we plan to extend this work in the future to understand the exact behaviour of our model using explainability methods (e.g. linear probing). However, we can empirically see the effects of this alignment as Sumformers are able to generalize well OOD (see Table 1 and Tables 6-8 in our paper) whereas the non-aligned Transformers fail on OOD data.
> > >
> > > Regarding your question about figures 9-10, we note that while our models are specifically biased towards some algorithmic behavior, they are not immune to overfitting (which is happening as we continue training beyond 200 epochs). Additionally, figures 9-10 are also meant to further support the usage of Sumformers vs. Transformers as they show the Transformer's OOD performance does not improve even with longer training times.

---

### Note · Authors · 2025-08-14

We again thank the reviewers for an active and productive discussion period. All additional results presented in our replies will be added to the final version of our paper. As a final remark, we would like to reiterate the motivation and contributions of our paper. The key motivation behind our work is developing data-driven techniques for solving difficult geometric optimization problems. Our proposed neural model leverages algorithmic alignment to show that one can solve extent measure problems with bounded model complexity. At the same time, we show empirically that our models outperform natural neural baselines and are more efficient than classical algorithmic baselines (see our replies to Reviewers VDgZ and r5xy). We also note that an effective neural model for algorithmic tasks has the additional advantage that it is differentiable, which allows our models to be integrated into other ML pipelines. This is generally not possible with classical algorithms. Overall, our results open the door to further explorations of how classical algorithms inform model design for efficiently solving classical geometric optimization problems as well as downstream point cloud processing tasks.

---

### Decision · Program_Chairs · 2025-09-17

**Decision:**

Accept (poster)

**Comment:**

This paper proposes an approach for approximating "extent measures" in computational geometry using neural networks. Initially, the paper received one Reject and three Borderline Accept ratings. The reviewers appreciated the theoretical contribution coupled with the implementation, but raised concerns about the relatively limited quantitative evaluation and the limited discussion of related work. The rebuttal initiated further discussion, and in general the reviewers responded positively, with the paper eventually receiving one Reject, one Borderline Accept, and two Accept ratings. The reviewer advocating for rejection felt that the paper is not yet ready for publication and requires more careful evaluation. However, the other three reviewers valued the theoretical contribution, the practical demonstration, and the well-motivated approach. Based on this balance of opinions, the AC sides with the majority and recommends acceptance. Authors should still consider any additional comments or feedback from the reviewers while preparing their final version and of course update the manuscript to include any additional promised changes, analysis, results and/or discussion they provided in their rebuttal, particularly the extended quantitative evaluation results.